# The Central African Soil Spectral Library: A new soil infrared repository and a geographical prediction analysis

Laura Summerauer[1], Philipp Baumann[1], Leonardo Ramirez-Lopez[2], Matti Barthel[1], Marijn Bauters[3,4], Benjamin Bukombe[5], Mario Reichenbach[5], Pascal Boeckx[3], Elizabeth Kearsley[4], Kristof Van Oost[6], Bernard Vanlauwe[7], Dieudonné Chiragaga[7], Aimé Bisimwa Heri-Kazi[6], Pieter Moonen[8], Andrew Sila[9], Keith Shepherd[9], Basile Bazirake Mujinya[10], Eric Van Ranst[11], Geert Baert[3], Sebastian Doetterl[1,5], and Johan Six[1]

[1]Department of Environmental Systems Science, ETH Zurich, Switzerland
[2]Data Science Department, BUCHI Labortechnik AG, Flawil, Switzerland
[3]Department of Green Chemistry and Technology, Ghent University, Ghent, Belgium
[4]Department of Environment, Ghent University, Ghent, Belgium
[5]Institute of Geography, University of Augsburg, Germany
[6]Earth and Life Institute, UCLouvain, Louvain-la-Neuve, Belgium
[7]International Institute of Tropical Agriculture, Nairobi, Kenya and Bukavu, Democratic Republic of Congo
[8]Department of Earth and Environmental Sciences, KU Leuven, Leuven, Belgium
[9]World Agroforestry Centre, Nairobi, Kenya
[10]Department of General Agricultural Sciences, University of Lubumbashi, Lubumbashi, Democratic Republic of Congo
[11]Department of Geology, Ghent University, Gent, Belgium

**Correspondence:** Laura Summerauer, Universitaetstrasse 16, 8092 Zurich, Switzerland (laura.summerauer@usys.ethz.ch)

**Abstract.** Information on soil properties is crucial for soil preservation, improving food security, and the provision of ecosystem services. Especially, for the African continent, spatially explicit information on soils and their ability to sustain these services is still scarce. To address data gaps, infrared spectroscopy has gained great success as a cost-effective solution to quantify soil properties in recent decades. Here, we present a mid-infrared soil spectral library (SSL) for central Africa (CSSL) that can predict key soil properties allowing for future soil estimates with a minimal need for expensive and time-consuming wet chemistry. Currently, our CSSL contains over 1,800 soil samples from ten distinct geo-climatic regions throughout the Congo Basin and along the Albertine Rift. For the analysis, we selected six regions from the CSSL, for which we built predictive models for carbon (TC) and total nitrogen (TN) using an existing continental SSL (African Soil Information Service, AfSIS SSL; n = 1902) that does not include central African soils. Using memory-based learning (MBL), we explored three different strategies at decreasing degree of geographic extrapolation, using models built with (1) the AfSIS SSL only, (2) AfSIS SSL combined with the five remaining central African regions, and (3) a combination of AfSIS SSL, the remaining five regions, and selected samples from the target region (spiking). For this last strategy we introduce a method for spiking MBL models. We found that when using the AfSIS SSL only to predict the six central African regions, the Root Mean Square Error of the predictions ($RMSE_{pred}$) was between 3.85–8.74 $g\,kg^{-1}$ and 0.40–1.66 $g\,kg^{-1}$ for TC and TN, respectively. The Ratio of Performance to the InterQuartile distance ($RPIQ_{pred}$) ranged between 0.96–3.95 for TC and 0.59–2.86 for TN. While the effect of the second strategy compared to the first strategy was mixed, the third strategy, spiking with samples from the target regions,

could clearly reduce the $RMSE_{pred}$ to 3.19–7.32 g kg$^{-1}$ for TC and 0.24–0.89 g kg$^{-1}$ for TN. $RPIQ_{pred}$ values were increased to ranges of 1.43–5.48 and 1.62–4.45 for TC and TN, respectively. In general, predicted TC and TN for soils of each of the six regions were accurate; the effect of spiking and avoiding geographical extrapolation was noticeably large. We conclude that our CSSL adds valuable soil diversity that can improve predictions for the Congo Basin region compared to using the continental AfSIS SSL alone; thus, analyses of other soils in central Africa will be able to profit from a more diverse spectral feature space. Given these promising results, the library comprises an important tool to facilitate economical soil analyses and predict soil properties in an understudied yet critical region of Africa. Our SSL is openly available for application and for enlargement with more spectral and reference data to further improve soil diagnostic accuracy and cost-effectiveness.

## 1 Introduction

Soil health is critical to crop nutrition, agricultural production, food security, erosion prevention, and climate change mitigation via carbon storage. Global climate change and soil degradation by deforestation and soil mismanagement critically threaten these ecosystem services (Birgé et al., 2016). In particular, the humid tropics are a front line for these anthropogenic impacts. For example, increasing temperatures and accelerating deforestation in the humid tropics are estimated to enhance greenhouse gas emissions (Don et al., 2011; Cox et al., 2013), but also to significantly reduce soil functions and ecosystem services such as soil fertility, water storage and filtration capabilities and erosion protection (Veldkamp et al., 2020). Despite the expected severity of these impacts, our understanding of the effects on soils in the humid tropics of Africa are limited by sparse data and uneven distribution of low-latitude research. Within the tropics, both the future impacts and data gaps are most severe in the Congo Basin, which contains the second largest tropical forest ecosystem on Earth, represents a considerable reservoir of soil carbon and is critically endangered by fast deforestation (Hansen et al., 2013). Thereby, forest loss in central Africa is mainly driven by smallholder farmers practicing shifting cultivation (Tyukavina et al., 2018; Curtis et al., 2018) and cropland expansion to feed a fast growing population. For example, human population in Uganda, Rwanda and DRC are projected to more than double in the coming 80 years (Vollset et al., 2020). Such dramatic growth will likely contribute to further agricultural conversion. In the wake of these current and future impacts, more spatially explicit soil information is urgently needed in many research fields ranging from agricultural, to soil biogeochemistry and climate sciences. In recent decades, improvements have been made carrying out soil surveys and creating soil databases and maps for central Africa (Goyens et al., 2007), for Rwanda (Imerzoukene and Van Ranst, 2002) and for the DRC (Baert et al., 2013). Unfortunately, accessibility to such data is limited and gaps are still large in central Africa (Van Ranst et al., 2010), in parts due to the high cost of specialized equipment and chemicals for analyses, limited accessibility to sampling areas, and lack of infrastructure.

Diffuse Reflectance Infrared Fourier Transform (DRIFT) spectroscopy has gained attention as a cost effective and fast method for soil analyses (e.g., Nocita et al., 2015). Many soil minerals, as well as functional groups of soil organic matter, show distinct energy absorption features in the infrared (IR) region of the electromagnetic spectrum. These relationships can be empirically modelled to quantify soil properties relevant for soil quality, such as C, nitrogen (N) and other crop nutrients (e.g., Janik et al., 1998; Soriano-Disla et al., 2014). Due to its simple handling, quick measurements, low costs, and minimal need

for chemical consumables, IR spectroscopy is an important tool for soil analyses that further allows high reproducibility and coverage of spatial soil heterogeneity. Especially in developing countries, where practices are often hampered by the prohibitive costs of conventional soil analyses, IR spectroscopy has great potential (Shepherd and Walsh, 2007; Ramirez-Lopez et al., 2019).

Partial Least Squares (PLS) is a projection-based regression method which can be considered as the most widely used tool to calibrate models that translate spectral data into meaningful chemical and or physical information. The method is especially useful in non-complex contexts, where the relationships between spectra and response variables are essentially linear (e.g., spectral models developed for a small field where soil forming factors are relatively constant). One of the main aims of establishing large-scale soil spectral libraries (SSLs) is to minimize the need for future wet chemical analyses (e.g., Stevens et al., 2013; Shi et al., 2014; Viscarra Rossel et al., 2016; Demattê et al., 2019). However, these libraries often span vast geographical areas that include different soil types and climate zones, which comprise complex soil organic carbon forms and mineral compositions. Due to this heterogeneity, predictions rendered by global linear regression models are often unfeasible for new local soil property assessments at a regional, field or plot-scale, especially when the new set covers another geographical domain than the library. Despite the abundance of literature on the calibration of quantitative models of soil properties using both mid-infrared (MIR) and near-infrared (NIR) data, there is still a lack of simple and efficient modeling strategies that could bring SSLs to an operational level. Padarian et al. (2019) could considerably improve prediction accuracies for a new local set when using a compositionally related subset from a large-scale SSL together with a small number of local reference analyses. Thus, cost-accuracy trade-off can be met when the accuracy of the library-based prediction is similar to the one made when applying a local but more costly calibration strategy. Several data-driven methods have proven to be successful to overcome this issue, for example RS-LOCAL (Lobsey et al., 2017) and memory-based learning (a.k.a local learning (e.g., Naes et al., 1990; Shenk et al., 1997; Ramirez-Lopez et al., 2013a)). In addition, other promising approaches have also been proposed, although they require more research (e.g. deep learning (Ng et al., 2019), fuzzy rule-based systems (Tsakiridis et al., 2019)). Memory-based learning (MBL), for example, searches for each new spectral observation, a subset of similar observations in a spectral library, which are then used to fit a custom predictive model for every new observation. This method has shown promising results when applied to extremely complex SSLs such as the MIR library of the United States (Dangal et al., 2019) and in one developed for the European continent (Tsakiridis et al., 2019). Spiking of libraries with samples from the target site has also shown to improve prediction accuracy (e.g., Guerrero et al., 2010; Wetterlind and Stenberg, 2010; Seidel et al., 2019; Barthès et al., 2020). So far, SSLs have mainly been used for predictions of soil samples originating from the same geographical domain. Studies have shown that subsetting large-scale libraries for new spectra by their geographical zones can result in good prediction accuracy (Nocita et al., 2014; Shi et al., 2015). These geographical restrictions could allow for extrapolation to new areas that contain similar soils.

The aim of the present work was to propose three strategies that leverage the use of a large soil infrared spectral library to accurately predict soil properties in regions which are poorly covered by it. Furthermore, here we describe a convenient method for spiking MBL or local models. Here, we also present the first SSL for central Africa (CSSL) which can be used to enlarge the existing continental library of African soils (a.k.a AfSIS). This effort represents an important first step towards fulfilling the

 need for spatially explicit and high-resolution soil data in an important yet understudied region in the humid tropics of Africa, promoting vital soil information that is critical to the future of the region.

## 2 Methods

### 2.1 Site descriptions

Soil samples were collected from past projects in the Congo Basin and along the Albertine Rift, the western branch of the East African Rift System. Table 1 gives an overview of corresponding data sources and data contributors to the different sample sets and denotes the origin, the number of samples and sampling layers used for our CSSL. The sample locations of the entire library are clustered over a large geographical area of central Africa, from a latitude of 2.8 °N to -11.6 °N and a longitude of 12.9 °E to 30.4 °E. From our entire library, six clustered regions were identified which contained at least 80 samples to allow for reliable analysis. Therefore, this subset will be further presented in the manuscript (see Table A1 and Table A2 for information on the entire library). Four of the selected regions are located in the Democratic Republic of the Congo (Haut-Katanga, South Kivu, Tshopo, Tshuapa) while the other two are located in Rwanda (Iburengerazuba) and in Uganda (Kabarole), respectively (Figure 1 and Figure A1). Site specific characteristics, coordinate ranges, altitudes, average climate, and dominant soil types are summarized in Table 2. Annual precipitation ranges from about 1200 mm in Haut-Katanga to over 2000 mm in the tropical forest of Tshuapa. Mean annual temperature varies from 17.6 °C in the high altitudes of Ibrengerazuba and South Kivu to 24.9 °C in Tshopo (Fick and Hijmans, 2017). The study elevations range from 380 m.a.s.l. in Tshuapa and Tshopo to high altitudes of 2300 m.a.s.l. in South Kivu along the rift valley (Jarvis et al., 2008). Soil types are primarily Ferralsols, Acrisols, or Nitisols (Jones et al., 2013; IUSS Working Group WRB, 2015). The different regions contain multiple Köppen-Geiger climatic zones: The three regions located close to the equator (Tshuapa, Tshopo, Kabarole) are classified as Af (tropical rainforest), while eastern DRC and western Rwanda are classified as a mixture of climate zones Cfb (temperate, without dry season, warm summer), Csb (temperate, dry summer, warm summer), Aw (tropical savannah) and Cwb (temperate, dry winter, warm summer). The regions along the rift valley (South Kivu, Iburengerazuba, Kabarole) are partially also classified as Am (tropical monsoon). Finally, the south-east of the DRC is classified as Cwa (temperate, dry winter, hot summer) (Beck et al., 2018).

### 2.2 Laboratory soil analyses

In preparation for total carbon (TC) and total nitrogen (TN) analyses, all soil samples were sieved through a 2 mm mesh and either air-dried or oven-dried at temperatures of 50 °C or 60 °C. After sieving and drying, soil samples were ground to a powder ($< 50$ $\mu$m) using a ball mill. TC and TN were analyzed via dry combustion using either a LECO 628 Elemental Analyzer (LECO Corporation, USA), an ANCA-SL Automated Nitrogen Carbon Analyzer (SerCon, UK), or a Vario EL Cube CNS Element Analyzer (Elementar, Germany). In order to ensure data quality and facilitate the harmonisation of all TC and TN data, a subset of these samples were re-measured on the LECO. This performance comparison demonstrated high comparability

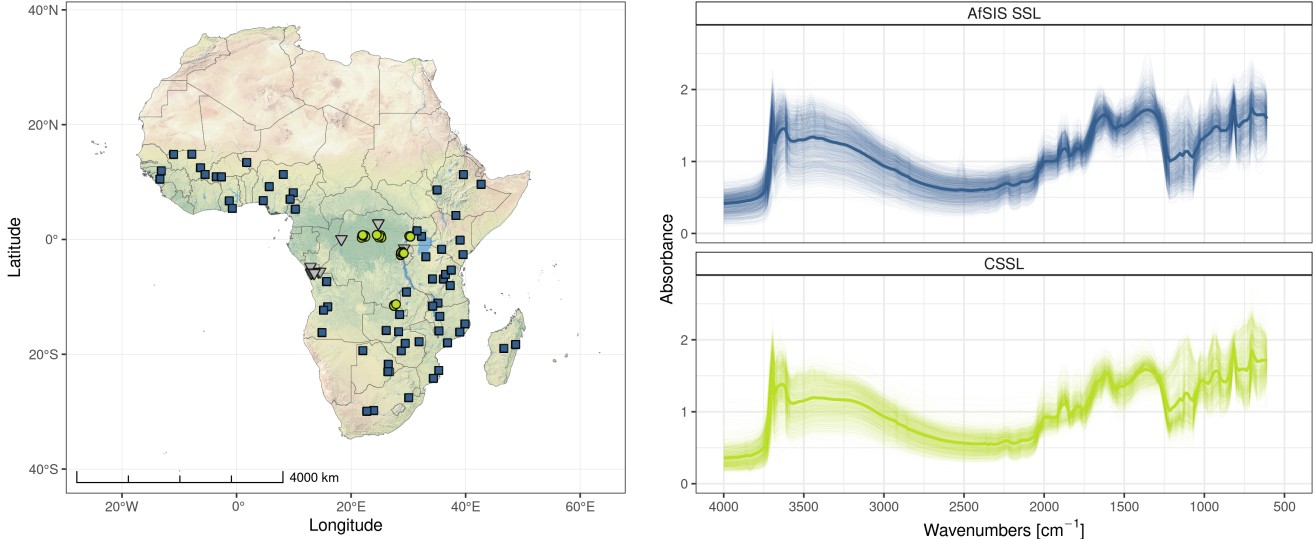

**Figure 1.** Location and spectra of soil samples from the central African spectral library (CSSL) and the continental soil spectral library from the African Soil Information Service (AfSIS SSL; left). The samples from the six regions from the CSSL further analyzed in this study are presented with a ○ symbol, the remaining samples of CSSL with a ▽ symbol and the AfSIS SSL with □ symbol. The average resampled spectrum of each library are shown (bold line) along with the individual resampled sample spectra (transparent lines; right).

**Table 1.** Soil sample archive used for the central African soil spectral library. The references show the publications from which the corresponding data was sourced. For previously unpublished data, the contributor institution is listed. The listed regions are provinces of the Democratic Republic of Congo (DRC) and Rwanda (RWA) and a district of Uganda (UGA).

| Data Source or Contributor | Region | $n$ | Soil Depth (cm) |
|---|---|---|---|
| Bauters et al. (2015, 2017) | Tshopo (DRC) | 33 | 5–10 |
| Gallarotti et al. (2021), Baumgartner et al. (2020) | Tshopo, South Kivu (DRC) | 38 | 0–5, 5–20, 0–15, 15–30 |
| Swiss Federal Institute of Technology Zurich | North Kivu, South Kivu, Équateur (DRC) | 40 | 0–5, 5–20, 0–15, 15–30, 0–100 soil pit |
| Kearsley et al. (2013, 2017) | Tshopo (DRC) | 40 | 0–10, 10–20, 20–30, 30–50, 50–100 |
| Bauters et al. (2019a) | Tshopo, South Kivu (DRC) | 12 | 0–5 |
| Bauters et al. (2021), Moonen et al. (2019) | Tshopo (DRC) | 208 | 0–5, 5–10, 10–20 |
| Bauters et al. (2019b) | Tshuapa (DRC) | 75 | 0–10, 10–20, 20–30, 30–50, 50–100 |
| Minten (2017) | Tshuapa (DRC) | 103 | 0–20 |
| Summerauer (2017) | Tshuapa (DRC) | 560 | 0–20, 20–50 |
| Heri-Kazi (2020) | South Kivu (DRC) | 51 | 0–20 |
| Université Catholique de Louvain | South Kivu, Haut-Katanga (DRC) | 46 | 0–20, 20–30 |
| Mujinya (2012); Mujinya et al. (2010, 2011, 2013, 2014) | Haut-Katanga (DRC) | 94 | Termite mound profiles |
| International institute of Tropical Agriculture / World Agroforestry Centre | Bas-Uélé, South Kivu (DRC) | 207 | 0–20, 20–40, 20–50 |
| Baert et al. (2009); Baert (1995) | Lower Congo and Central (DRC) | 40 | 0–123 soil pit |
| Doetterl et al. (2021) | South Kivu (DRC), Iburengerazuba (RWA), Kabarole (UGA) | 307 | 0–10, 30–40, 60–70, 90–100 |

**Table 2.** Number of samples, GPS coordinates, elevation, annual precipitation (AP), mean annual temperature (MAT), Koeppen-Geiger climate classifications and soil types for the sampled regions of the Democratic Republic of Congo, Rwanda and Uganda. Data were extracted for all coordinates from raster files: Climate data is sourced from Fick and Hijmans (2017), elevation from SRTM (90m resolution; Jarvis et al. (2008)), Köppen-Geiger climate classifications from Beck et al. (2018) and soil types from the *Soil Atlas of Africa* (Jones et al., 2013; IUSS Working Group WRB, 2015)

| Region | $n$ | Longitude (° E) | Latitude (° N) | Elevation (m) | MAT (°C) | AP (mm) | Köppen-Geiger | Soil types |
|---|---|---|---|---|---|---|---|---|
| Haut-Katanga | 119 | 27.48–27.85 | -11.61– -11.29 | 1197–1323 | 20.6 | 1223 | Cwa | Rhodic/Haplic Ferralsols |
| South Kivu | 369 | 28.64–28.91 | -2.79– -2.1 | 1487–2310 | 17.6 | 1627 | Cfb, Csb, Aw, Cwb | Umbric Ferralsols, Haplic Acrisols |
| Tshopo | 315 | 24.48–25.32 | 0.29–0.83 | 380–506 | 24.9 | 1789 | Af | Xanthic/Haplic Ferralsols |
| Tshuapa | 738 | 21.84–22.53 | 0.28–0.8 | 385–578 | 24.7 | 2090 | Af | Xanthic/Haplic Ferralsols |
| Iburengerazuba | 107 | 29.05–29.22 | -2.47– -2.34 | 1565–1939 | 17.6 | 1496 | Csb, Aw, Cwb | Haplic/Umbric Acrisols |
| Kabarole | 101 | 30.13–30.37 | 0.46–0.63 | 1271–1824 | 19.7 | 1360 | Af, Cfb, Am | Haplic Phaeozems, Rhodic Nitisols, Albic Luvisols |

of CN and TN data across all three instruments ($R^2 = 0.99$ for TC and TN, results not shown). The large majority of the soil samples originate from highly weathered and acidic soils and do not contain any carbonates and therefore, TC contents correspond to total organic carbon contents. Only in a few samples from termite mounds in the subtropical Haut-Katanga province calcium carbonate has been detected where pH values are > 8 (Mujinya, 2012). Moreover, the widely used slash-and-burn practices could additionally have influenced soil TC contents, even when visible charcoal pieces were removed prior to any measurement. Additionally, soil pH (either in $H_2O$, KCl or $CaCl_2$, depending on the study), texture (laser diffraction particle size analyser), and aqua regia extractable macro/micro nutrients (Al, Fe, Ca, Mg, Mn, Na, P and K; inductively coupled plasma-optical emission spectroscopy) were analyzed for a subset of samples. The chemical and MIR prediction results for these soil characteristics are not presented in this manuscript but were carried out using the same methods and are available on our GitHub repository (https://doi.org/10.5281/zenodo.4351254).

## 2.3 MIR spectral libraries

*Central African spectral library*

In order to determine the MIR reflectance, all ground soil samples were measured with a VERTEX70 Fourier Transform-IR (FT-IR) spectrometer with a High Throughput Screening Extension (HTS-XT) (Bruker Optics GmbH, Germany). Spectra were acquired at a resolution of $2\,cm^{-1}$ within a range of $7500\,cm^{-1}$ to $600\,cm^{-1}$, which corresponds to a wavelength range of $1333\,nm$ to $16667\,nm$. A gold coated reflectance standard (Infragold NIR-MIR Reflectance Coating, Labsphere) was used as a background material for all measured soils in order to normalize the sample spectra. Reflectance was transformed into absorbance using log(1/reflectance) prior to further processing and subsequent modeling. Two replicates per sample were filled into the cups of a 24-well plate and the surface was flattened without compression using a spatula. For each replicate, 32 co-added internal measurements were averaged and corrected for $CO_2$ and $H_2O$ using the OPUS spectrometer software (Bruker

Optics GmbH, Germany). This library is denoted as $C = \{Yc, Xc\}_1^m$ throughout the rest of the manuscript, being $Yc$ the matrix containing the two response variables (TC and TN), $Xc$ the matrix of spectra and $m$ the total number of samples in the library.

*AfSIS spectral library*

We used a MIR SSL created by the World Agroforestry (ICRAF) centre to predict soil samples of the six selected regions of central Africa for their TC and TN contents. This SSL was created as part of the Africa Soil Information Service (AfSIS) in order to improve soil information and land management on the continental scale of Sub-Saharan Africa (Vågen et al., 2020). For this continental library (see Figure 1), reference values for TC and TN were measured by using a ThermoQuest EA 1112

Elemental Analyzer. The MIR spectra of the samples were obtained by scanning them on a Tensor27 FT-IR spectrometer (Bruker Optics GmbH, Germany) with a high throughput screening extension. Soil samples were measured in a wavenumber range of $4000 \, \text{cm}^{-1}$ to $600 \, \text{cm}^{-1}$ ($2500 \, \text{nm}$ to $16666 \, \text{nm}$) with a spectral resolution of $2 \, \text{cm}^{-1}$. Four replicates per sample were measured and an average of 32 co-added scans were used for each sample (Sila et al., 2016). Here we denote this library as $A = \{Ya, Xa\}_1^n$ throughout the rest of the manuscript, where for all its samples ($n$), $Ya$ represents the matrix containing the

two response variables (TC and TN) and $Xa$ represents the matrix of spectra.

## 2.4 Spectral resampling and pre-processing

All CSSL and AfSIS spectra were processed using the R packages 'prospectr' (Stevens and Ramirez-Lopez, 2020), 'simpler-spec' (Baumann, 2020), and 'resemble' (Ramirez-Lopez, 2020) in the R statistical computing environment (R Core Team, 2020). Replicates of spectral measurements were aggregated to one average spectrum per sample. The spectra were then re-

155 sampled to a resolution of $16 \, \text{cm}^{-1}$ and trimmed to the $4000$–$600 \, \text{cm}^{-1}$ spectral range. Both spectral libraries were scanned on two FT-IR Bruker spectrometers (Bruker Optics GmbH, Germany), which use the same settings and the same internal standards. The scanning methods of the CSSL were adapted to the standard operating procedures of Soil Plant Spectral Diagnostics Laboratory at ICRAF. For these reasons, no instrument standardization was necessary.

As spectral pre-treatments have a marked impact on the performance of quantitative infrared models (Rinnan, 2014; Seybold

et al., 2019), the pre-processing procedure was specifically optimized for the MIR spectra of the central African samples. This procedure was based on the PLS method (Wold et al., 1984), which was also known as projection to latent structures and is widely used for regression analysis in infrared spectroscopy. However, it is also useful for projecting the spectral data onto a low-dimensional (and therefore less complex) subspace containing all the meaningful information of the original data. The projection model can be expressed as:

$$X = SP' + E \tag{1}$$

where $X$ is the original spectral matrix of $n \times d$ dimensions, $S$ is the PLS score matrix of $n \times l$ dimensions (where $l \leq \min(n, d)$) which contains the extracted variables, $P$ is the matrix of loadings of $d \times l$ dimensions which captures the spectral

variability across observations. E is an error term. For spectral data with high collinearity, the optimal $l$ (or the number of PLS factors) is usually small, which means that only a few PLS factors or latent variables are enough to properly represent the original variability of X. An important aspect of this type of projection is that it is obtained in such a way that the covariance between S and an external set of one or more variables is maximized. For a detailed description on PLS we refer the reader to Wold et al. (2001). In PLS, P can be used on new spectral observations to project them onto the lower dimensional subspace:

$$S_{new} = X_{new}P^{-1} \tag{2}$$

The spectral reconstruction residuals of the projection model can be then computed by back-transforming the matrix of scores to a spectral matrix and comparing it against the original spectral matrix as follows:

$$E_{new} = X_{new} - S_{new}P' \tag{3}$$

Finally, the spectral reconstruction error (also known as the Q-statistic) is computed as the sum of squares of $E_{new}$:

$$Q_{new} = E_{new}E'_{new} \tag{4}$$

The Q-statistic indicates how well a given new sample is represented by the PLS model (Wise and Gallagher, 1996; Ballabio and Consonni, 2013). This statistic is widely used in chemometrics for outlier identification and uncertainty assessment (Wise and Roginski, 2015).

In summary, our approach offers a data-driven solution to the selection of the spectral pre-processing steps which are optimized for the target/prediction set. The optimal set of steps is defined as the one that minimizes the Q-statistic. This approach does not require a prior knowledge of the response values of the target set and therefore is well suited for pre-processing optimization. It assumes that PLS models that cannot account for the spectral variability in the target set, may also fail at producing accurate predictions of the response variable. In other words, as suggested by Wise and Roginski (2015), large Q values can be used as proxies for large prediction errors, and therefore Q values can be used to judge the suitability of a set of pre-processing steps. To find an optimal combination of spectral pre-treatments, we defined a set of different pre-treatments $\{h_1, h_2, ..., h_z\}$ where $h_i$ represents one pre-treatment or a sequence of pre-treatments (with unique parameter values) to be applied on the spectral data. For this purpose, a projection model was built with the AfSIS spectra (using TC and TN as external variables) for each combination of spectral pre-treatments:

$$h_i(Xa) = Sa(i)Pa'(i) \tag{5}$$

this model was then used on the CSSL pre-treated spectra with reconstruction residuals (Ec) computed as follows:

$$Ec = h_i(Xc) - [h_i(Xc)Pa^{-1}ScPa'(i)] \tag{6}$$

where Pa are the loadings corresponding to the PLS model built with the AfSIS library, while Sc are the projected scores of the Central African Library.

For this analysis we fixed the number of PLS factors to 20 because projected variables beyond this dimension did not capture a sufficient amount of the original spectral variance. For example, PLS variable 21 amounted for less than 0.01 % of the original variance in all the cases. The mean Q value ($\bar{Q}$) for the $i$th set of pre-treatments was obtained by:

$$\bar{Q}(i) = \frac{1}{m\,d} \sum_{j=1}^{m} Ec_j Ec_j' \tag{7}$$

where $m$ and $d$ are the number of samples and the number of spectral variables in the CSSL respectively. To allow for comparisons across the reconstruction errors obtained for the different pre-treatments, $\bar{Q}$ was standardized as follows:

$$s\bar{Q}(i) = \frac{\bar{Q}(i)}{max(h_i(\mathrm{Xc})) - min(h_i(\mathrm{Xc}))} \tag{8}$$

Tested pre-treatments included different combinations of standard normal variate, multiplicative scatter correction, spectral detrending, first and second derivatives and window sizes from 3 to 35 points in increments of 2. Minimal spectral reconstruction error was achieved with a Savitzky-Golay filter with a second-order derivative using a second order polynomial approximation with a window size of 17 cm$^{-1}$ (Savitzky and Golay, 1964), and a subsequent multiplicative scatter correction. This pre-treatment was then applied to the spectra prior MBL.

### 2.5 Principal component analysis data visualization

To analyze the difference between the two spectral libraries and to visualize the similarities between soil samples, a principal component analysis (PCA) was conducted on the pre-processed spectra of both libraries. The PCA was performed with centering, but without scaling of the absorbance values.

### 2.6 Modeling approach

In the following we describe the method we used to assess the performance of MBL for predicting TC and TN for six distinct regions at different scenarios of regional soil extrapolation. Three specific modeling strategies were tested on the selected regional sets which we call validation sets (see subsubsection 2.6.2). With the regional analysis we demonstrate how predictions of soil properties within new sites from distinct regions—which are compositionally less variable than the available SSLs—perform and profit from knowledge present in the AfSIS SSL. The analysis also demonstrates the added value of our new CSSL in addition to the AfSIS SSL alone. Doing so, the aims of the modeling scenarios were 1) to minimize the costs and time for traditional methods by optimizing the transfer of stored spectral information to the new region of interest, and 2) to test different levels of geographical extrapolations for new regions, when no chemical analyses of local samples are available.

### 2.6.1 Modeling and prediction data

We used two main data sources and subsets as follows:

1. The AfSIS data set (A): Continental SSL from Sub-Saharan Africa including 1902 soil samples with both data MIR spectra and analytical reference data (Figure 1).

2. The central African data set ($\mathcal{C}$): The central African set comprises a total of 1578 soil samples which originate from six regions ($G_i$) named Haut-Katanga (119 samples), South Kivu (367 samples), Tshopo (134 samples), Tshuapa (738 samples), Iburengerazuba (104 samples) and Kabarole (100 samples) after the removal of one outlier sample from South Kivu with a large Mahalanobis distance to the AfSIS SSL and therefore high prediction uncertainties (distance $> 3$; results not shown). Each regional set was split up into a regional validation set ($G_i \setminus K_i$) and into a spiking set ($K_i$). For this work we differentiated between three different subsets which are defined as follows:

   (a) The union of the six regional subsets $\mathcal{C}$:

$$\mathcal{C} = \bigcup_1^6 G_i \tag{9}$$

   (b) Regional validation subsets which are the regional sets without the spiking samples $G_i \setminus K_i$.

   (c) Six representative regional spiking subsets $K_i$ which were selected from each regional set $G_i$, using the k-means sampling method, which selects one sample per cluster calculated on a principal component analysis as described in Næs (1987). For examples on k-means sampling in soil spectroscopy, we refer the reader to Ramirez-Lopez et al. (2014); Vohland et al. (2016); Viscarra Rossel and Brus (2018). A size of 20 samples per region was selected to show a pronounced effect of spiking that avoided any geographical extrapolation.

### 2.6.2 Modeling strategies

Three different scenarios were compared which are related to the degree of the geographical extrapolation:

– Strategy 1: MBL predictions for the regional validation subsets ($G_i \setminus K_i$) were computed from models built only with A. This scenario represents an extreme case of extrapolation (from the geographical perspective) because no samples from the entire central African area are present in the AfSIS set Figure 1, which is the only data used to build the predictive models.

– Strategy 2: Predictions for every $G_i \setminus K_i$ are computed by using MBL models built from the pooled AfSIS data A together with the data from the remaining five regions $\mathcal{C}_i$, i.e. $A \cup \mathcal{C}_i$, where

$$\mathcal{C}_i = \bigcup_{\substack{j=1 \\ j \notin i}}^{6} G_j \setminus K_j \tag{10}$$

Strategy 2 evokes less pronounced geographical extrapolation than strategy 1.

– Strategy 3: This time, strategy 2 was repeated, but extrapolation was avoided by using the spiking samples from the same geographical region; Each regional set $G_i \backslash K_i$ was predicted by the pooled AfSIS data, the data of the remaining regions and the respective spiking set, i.e. $A \cup C_i \cup K_i$.

## 2.7  Predictive modeling

We used MBL as our predictive modeling approach. In the chemometrics literature, MBL is also known as local modeling
which describes a family of (non-linear) machine learning methods designed to handle complex spectral datasets (Ramirez-Lopez et al., 2013b). This type of learning method does not attempt to fit a general (global) predictive function using all available data. Instead, a new and unique function ($\hat{f}_i$) is built on-demand, every time a new prediction for a given response variable is required. This new function is built using only a subset of relevant observations from a reference set that are queried through $k$-nearest neighbor search. The MBL method implemented for this study uses a spectral nearest neighbor search based
on a moving window correlation dissimilarity. To measure the dissimilarity ($r$) between two spectra ($x_i$ and $x_j$), the following equation was used:

$$r(x_i, x_j; w) = \frac{1}{2w} \sum_{k=1}^{d-w} 1 - \rho(x_{i,\{k:k+w\}}, x_{j,\{k:k+w\}}) \tag{11}$$

where $d$ is the number of spectral variables, $\rho$ representing the Pearson's correlation function and $w$ the window size. The window size was optimized based on a spectral nearest-neighbor search within the AfSIS library. For every sample in the AfSIS
library, its closest sample (in the spectral space) was identified. Then, samples were compared against their closest neighbors in terms of TC and TN and root mean squared differences (RMSD) computed according to the following equations:

$$j(i) = NN(xa_i, Xa^{-i}) \tag{12}$$

$$RMSD = \sqrt{\frac{1}{2m} \sum_{i=1}^{n} \sum_{h=1}^{2} (ya_{i,h} - ya_{j(i),h})^2} \tag{13}$$

where $NN(xa_i, Xa^{-i})$ represents a function to obtain the index of the nearest neighbor of the $i$-th observation found in $Xa$
(excluding the $i$-th observation), $yc_{i,h}$, is the value of the $i$-th observation for the $h$-th property variable (either TC or TN). In total 10 window sizes were evaluated using this approach (from 31 up to 121 in steps of 10) and according to the RMSD, an optimal window size $w$ of 71 was chosen.

After nearest neighbor retrieval, our MBL method fits a local model using the Weighted Average Partial Least Squares (WA-PLS) regression algorithm proposed by Shenk et al. (1997). In this WA-PLS, the final prediction is a weighted average

of multiple predictions generated by PLS models built from different PLS factors. A range of latent variables from 5 to 30 in increments of 1 was used for the WA-PLS calculations. The weight for each component is calculated as follows:

$$w_j = \frac{1}{s_{1:j} \times g_j} \tag{14}$$

where $s_{1:j}$ is the root mean square of the spectral residuals of the new observation when a total of $j$ PLS components are used (i.e., all the components from the first one to the $j$th one) and $g_j$ is the root mean square of the regression coefficients corresponding to the $j$th PLS component (see Shenk et al. (1997) for more details).

The number of neighbors that needed to be retrieved was optimized using nearest neighbor (NN) cross-validation (Ramirez-Lopez et al., 2013b). Using this method, for each observation to be predicted, its nearest neighbor was excluded from the group of neighbors and then a WA-PLS model is fitted using the remaining ones. This model is then used to predict the value of the response variable of the nearest observation. Predicted values are finally cross-validated with the actual values (see Ramirez-Lopez et al. (2013b) for additional details). For the optimization of the nearest neighbor search, i.e. the nearest neighbor cross-validation, a grouping factor was used to avoid overfitting: keeping the nearest neighbor out, the model was trained with the remaining neighbors which were not from the same region as the hold-out neighbor (region corresponds to the sentinel sites within the AfSIS SSL). The minimum number of available neighbors was tested for each region prior to training the respective final models, which were then trained with neighborhood sizes varying from 150 to 500 neighbors in increments of 10. The best model and the optimal number of neighbors were determined by the minimal RMSE (Equation 15) of the nearest neighbor cross-validation, where $n$ is the number of neighbors used for the model, $y_i$ is the measured value of the hold-out neighbor, and $\widehat{y_i}$ is the value predicted by the remaining neighbors.

Subsequently, independent from their distances to the validation set, 1 to 20 spiking samples were added from the target region and forced into the neighborhood of every observation and thus used in the predictive models. Our approach differs from previous studies using local modeling methods in combination with spiking, where the samples were not forced into the neighborhoods (e.g. Barthès et al. (2020), Lobsey et al. (2017)). Our approach guarantees that the spiking set (which is assumed to carry important information) is fully used.

Stepwise spiking was applied to test the effect of spiking in general, and to find the smallest number of samples required for satisfying model performances. This was necessary, since soil samples from the same geographical region are usually governed by very similar formation processes (spatial autocorrelation (Fortin et al., 2016)) and MIR spectra partially reflect the compositional characteristics of these samples. Moreover, it is widely accepted that the most accurate predictions can be achieved by models built with samples originating from the same region because large non-linear complexity is avoided (e.g., Tziolas et al., 2019).

## 2.8 Model validation and prediction accuracy

For model validation, the RMSE statistics of the nearest neighbor cross-validation described in the previous section were used. Prediction accuracy of the predicted vs. the measured values was also calculated using RMSE (Equation 15), where in this case $y_i$ is the actual measured reference value and $\widehat{y}_i$ the prediction of the final model.

$$RMSE = \sqrt{\frac{1}{n}\sum_{i=1}^{n}(y_i - \widehat{y}_i)^2} \tag{15}$$

Model validation and prediction performance were additionally evaluated using the Mean Error (ME; mean of the absolute difference between predicted and observed values) and the Ratio of Performance to the InterQuartile distance (RPIQ; Bellon-Maurel et al. (2010)). For calculating RPIQ, the interquartile range of the observed reference data is divided by the RMSE of the nearest neighbor validation or by the RMSE of the prediction ($RMSE_{pred}$), respectively. This is particularly useful since RPIQ does not make any assumptions about the distribution of the reference data.

## 3 Results

The samples that comprise the CSSL exhibited a wide range of TC and TN contents (Figure 2). Validation and spiking sets for four of the six regions (Haut-Katanga, Tshopo, Tshuapa, Kabarole) had mean TC and TN of $9.30$–$18.10\,\mathrm{g\,kg^{-1}}$ and $0.95$–$1.74\,\mathrm{g\,kg^{-1}}$, respectively. Maximum TC and TN values for these four regions were $56.69\,\mathrm{g\,kg^{-1}}$ and $5.05\,\mathrm{g\,kg^{-1}}$, respectively. The other two regions, South Kivu in Eastern DRC and Iburengerazuba in Western Rwanda, had considerably higher TC and TN contents, with mean values of $23.55$–$35.43\,\mathrm{g\,kg^{-1}}$ and $1.34$–$3.07\,\mathrm{g\,kg^{-1}}$, respectively. The AfSIS SSL had generally lower mean TC and TN contents of $12.37\,\mathrm{g\,kg^{-1}}$ and $0.82\,\mathrm{g\,kg^{-1}}$, respectively.

### 3.1 Principal components and spectral variability in the two libraries

The first three principal components accounted for $85\,\%$ of the spectral variability (Figure 3). These components indicate that the majority of CSSL samples lie within the spectral domains of the AfSIS SSL as their PCA scores overlap. This overlapping is, however, less evident for the spectra of the South Kivu region and, to a lesser extent, for the samples of the Iburengerazuba and Tshuapa regions, which suggests that the type of soils in these regions may not be well represented by the AfSIS SSL compared to the other regions.

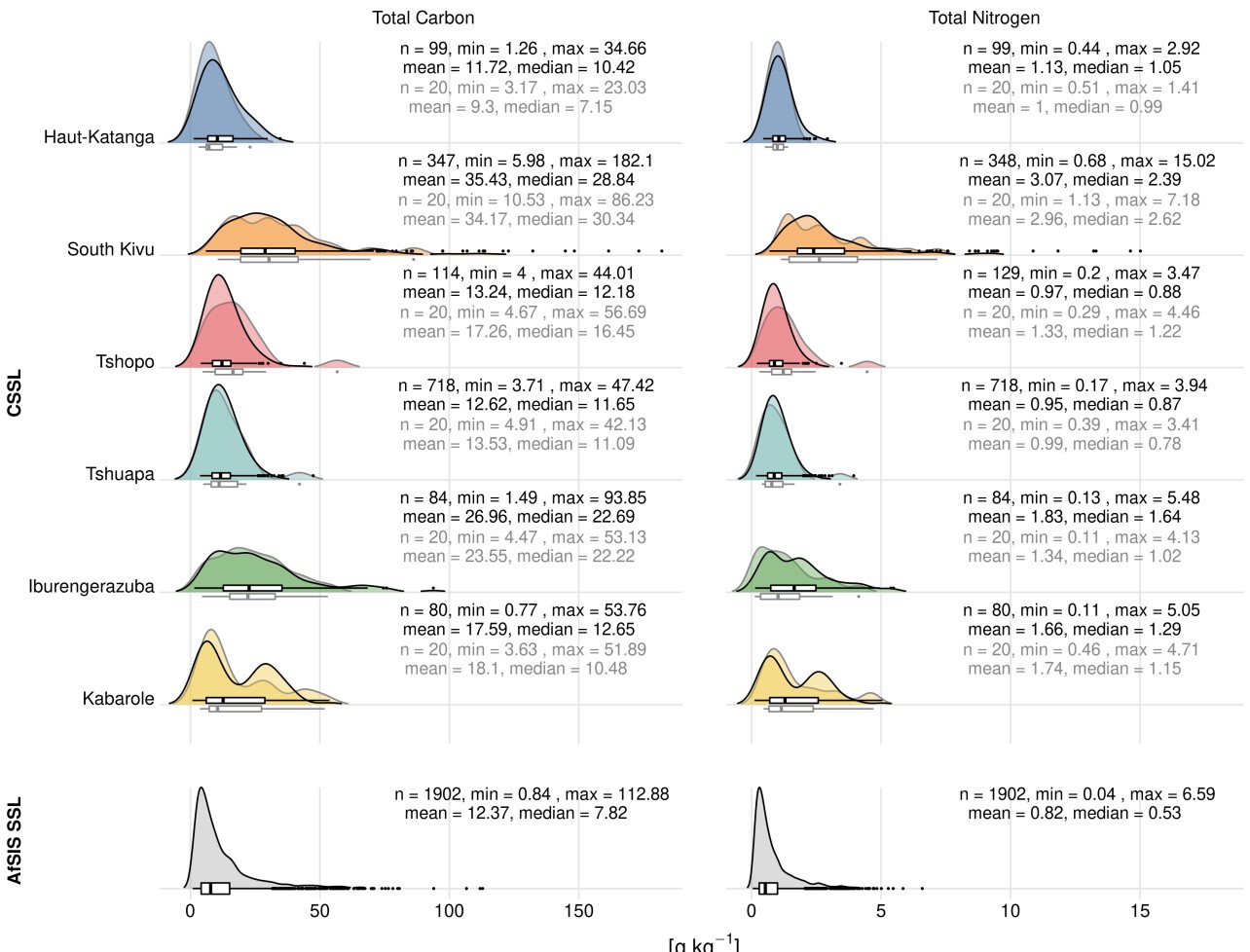

**Figure 2.** Summary of the reference data for total carbon (TC) and total nitrogen (TN) of the two soil spectral libraries (SSLs): the central African SSL (CSSL) and the continental SSL (AfSIS SSL). The CSSL is divided into the six regions (Haut-Katanga, South Kivu, Tshopo, Tshuapa, Iburengerazuba, Kabarole). The black lines and text indicate regional validation sets while the gray lines and text indicate the spiking sets.

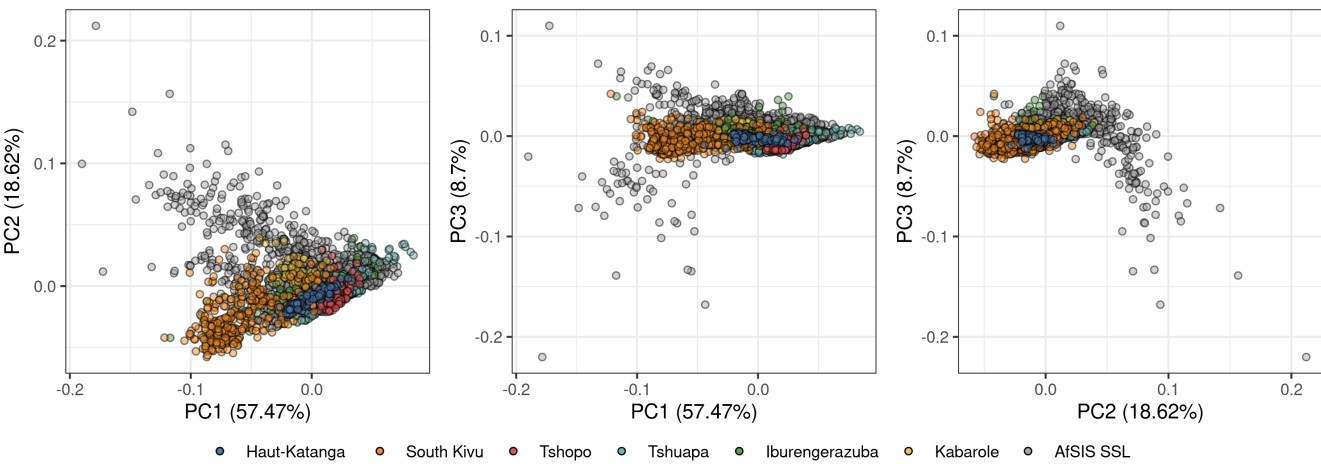

**Figure 3.** Score plots of the first three principal components of the pre-processed MIR spectra from the central African soil spectral library (six regions; coloured) and the large-scale continental library (AfSIS SSL; gray).

## 3.2 Predictive performance of the three strategies

In general, MBL retrieved accurate TC and TN predictions for all the strategies (with RMSE$_{pred}$ values below $9\,\mathrm{g\,kg^{-1}}$ for TC and below $1.7\,\mathrm{g\,kg^{-1}}$ for TN). South Kivu and Iburengerazuba regions showed the highest RMSE$_{pred}$, which was mainly due to the high TC and TN ranges (Figure 2). Prediction errors for Haut-Katanga, Tshopo and Tshuapa were comparably smaller, however, the RPIQ$_{pred}$ were among the smallest across regions as well (RPIQ$_{pred}$ 0.59–2.72). Relative to their TC and TN ranges, predictions for these three regions were less accurate than for South Kivu and Iburengerazuba (PRIQ$_{pred}$ 1.10–4.45). The TC and TN predictions in Kabarole were the most accurate compared to the other five regions (RPIQ$_{pred}$ 2.65–5.48 for TC and TN and all strategies; Table 3).

**Table 3.** Statistics of the independent validations of the predictions of total carbon and total nitrogen for each region and three strategies. Strategy 1: Predictions of the combined six regions by the AfSIS soil spectral library (SSL), Strategy 2: Predictions of the individual regions by the remaining five regions together with the AfSIS SSL, Strategy 3: Spiking six regional models from Strategy 2 with 20 samples from each target area.

| Strategy | Region | Total carbon [g kg$^{-1}$] | | | | | Total nitrogen [g kg$^{-1}$] | | | | |
|---|---|---|---|---|---|---|---|---|---|---|---|
| | | $n_{pred}$ | RMSE$_{pred}$ | R$^2_{pred}$ | ME$_{pred}$ | RPIQ$_{pred}$ | $n_{pred}$ | RMSE$_{pred}$ | R$^2_{pred}$ | ME$_{pred}$ | RPIQ$_{pred}$ |
| Strategy 1 | Haut-Katanga | 99 | 5.99 | 0.79 | 4.99 | 1.62 | 99 | 0.81 | 0.31 | 0.70 | 0.59 |
| | South Kivu | 347 | 8.61 | 0.94 | 3.50 | 2.43 | 348 | 1.66 | 0.85 | 1.32 | 1.10 |
| | Tshopo | 114 | 7.34 | 0.47 | 2.61 | 0.96 | 129 | 0.55 | 0.52 | 0.34 | 0.93 |
| | Tshuapa | 718 | 3.85 | 0.71 | 2.06 | 1.84 | 718 | 0.40 | 0.68 | 0.29 | 1.37 |
| | Iburengerazuba | 84 | 8.73 | 0.84 | 4.46 | 2.60 | 84 | 0.82 | 0.81 | 0.60 | 2.13 |
| | Kabarole | 80 | 5.73 | 0.86 | 1.10 | 3.95 | 80 | 0.65 | 0.84 | 0.47 | 2.86 |
| Strategy 2 | Haut-Katanga | 99 | 4.22 | 0.72 | 1.84 | 2.30 | 99 | 0.32 | 0.59 | 0.02 | 1.50 |
| | South Kivu | 347 | 8.88 | 0.95 | 4.72 | 2.36 | 348 | 1.17 | 0.89 | 0.72 | 1.55 |
| | Tshopo | 114 | 5.38 | 0.64 | 0.30 | 1.31 | 129 | 0.34 | 0.72 | 0.07 | 1.49 |
| | Tshuapa | 718 | 4.12 | 0.78 | 2.21 | 1.71 | 718 | 0.29 | 0.77 | 0.12 | 1.88 |
| | Iburengerazuba | 84 | 7.96 | 0.86 | 2.69 | 2.84 | 84 | 0.54 | 0.82 | 0.02 | 3.21 |
| | Kabarole | 80 | 8.56 | 0.83 | 4.29 | 2.65 | 80 | 0.64 | 0.86 | 0.40 | 2.90 |
| Strategy 3 | Haut-Katanga | 99 | 3.57 | 0.80 | 1.35 | 2.72 | 99 | 0.26 | 0.71 | 0.06 | 1.87 |
| | South Kivu | 347 | 7.32 | 0.95 | 1.53 | 2.86 | 348 | 0.89 | 0.89 | 0.32 | 2.05 |
| | Tshopo | 114 | 4.93 | 0.69 | 0.11 | 1.43 | 129 | 0.31 | 0.75 | 0.03 | 1.62 |
| | Tshuapa | 718 | 3.19 | 0.80 | 0.90 | 2.22 | 718 | 0.24 | 0.79 | 0.03 | 2.25 |
| | Iburengerazuba | 84 | 6.34 | 0.91 | 1.14 | 3.57 | 84 | 0.39 | 0.91 | 0.03 | 4.45 |
| | Kabarole | 80 | 4.13 | 0.94 | 1.72 | 5.48 | 80 | 0.44 | 0.91 | 0.23 | 4.27 |

### 3.2.1 Strategy 1: Predicted central African soils by the large-scale continental library

The TC and TN predictions for the six regions of central Africa were characterized by errors (RMSE$_{pred}$) ranging from 3.85–8.73 g kg$^{-1}$ and 0.40–1.66 g kg$^{-1}$, respectively. The best prediction accuracies for TC were achieved for South Kivu, Iburengerazuba, and Kabarole, where RPIQ$_{pred}$ values were between 2.43–3.95, while Tshopo, Tshuapa, and Haut-Katanga performed worse with RPIQ$_{pred}$ <= 1.84. For TN, Iburengerazuba and Kabarole performed well with RPIQ$_{pred}$ above 2. However, the four other regions Haut-Katanga, South Kivu, Tshopo, and Tshuapa, exhibited even lower RPIQ$_{pred}$ <= 1.37. For South Kivu, samples with high TC and TN contents ($> 100$ g kg$^{-1}$ TC and $> 5$ g kg$^{-1}$ TN) deviated from the 1:1 line (Figure 4). Moreover, TC predictions for Haut-Katanga, Tshopo, Tshuapa and Iburengerazuba, as well as TN predictions, in all six regions showed a clear trend towards underestimation (Figure 4). This can be caused by one or the combination of the three

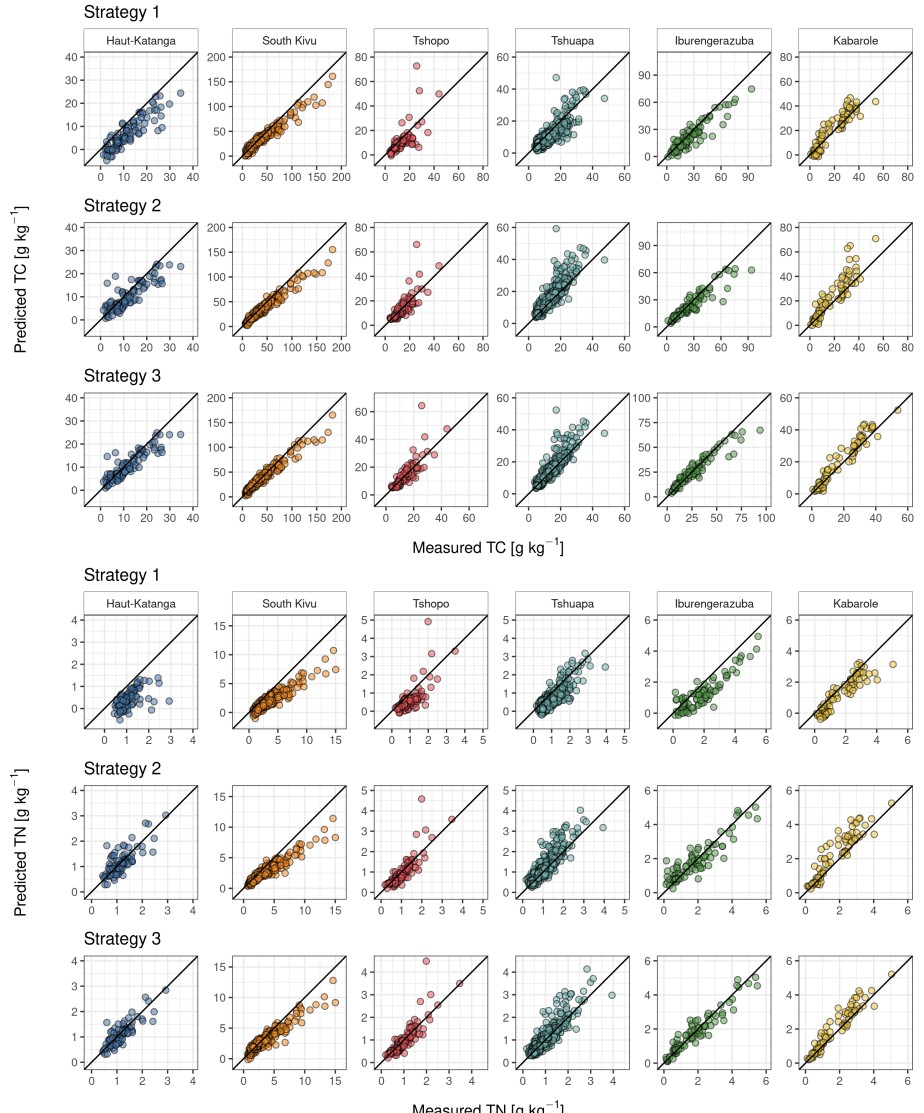

**Figure 4.** Predicted vs. measured total carbon (TC) and total nitrogen (TN) for soil samples of the six central African regions. Predictions for each region were made using Memory-based learning and (i) the large-scale continental soil spectral library (AfSIS SSL; strategy 1), (ii) the remaining five central African regions together with the AfSIS SSL (strategy 2), and (iii) 20 local spiking samples from each target region together with the remaining five central African regions and the AfSIS SSL (strategy 3). A 1:1 line is indicated as a visual aid.

following effects: *i)* the central African samples were poorly represented by the continental AfSIS SSL due to the differing pedogenic features (Figure 3), *ii)* spectral offset and/or multiplicative effects in the spectra (due to instrument differences) were not completely accounted by the pre-processing methods *iii)* performance differences exist between the conventional laboratory analyses used to obtain TC and TN reference values.

### 3.2.2 Strategy 2: Regional predictions by soil spectral libraries

Compared to strategy 1, strategy 2 partially showed better predictive performance for TC and in all the cases retrieved better TN predictions. These improvements are exemplified by the larger $RPIQ_{pred}$ and smaller $RMSE_{pred}$ values in strategy 2 (Table 3). The most accurate predictions for TC were obtained for the regions Haut-Katanga, South Kivu, Iburengerazuba and Kabarole ($RPIQ_{pred} > 2.30$). The predictive performances for TC of Tshopo and Tshuapa were similar with $RPIQ_{pred}$ values of 1.31 and 1.71, respectively. For TN, the predictive performance was best for Iburengerazuba and Kabarole ($RPIQ_{pred} > 2$). For the regions Haut-Katanga, South Kivu, Tshopo and Tshuapa the $RPIQ_{pred}$ values for TN were between 1.49–1.88. The predictions in strategy 2 exhibited errors ($RMSE_{pred}$) ranging between 4.12–8.88 $\mathrm{g\,kg^{-1}}$ and 0.29–1.17 $\mathrm{g\,kg^{-1}}$ for TC and TN, respectively (Table 3). Comparing the TC $RMSE_{pred}$ of each region across the first two strategies, errors for Haut Katanga, Tshopo and Iburengerazuba were substantially reduced in strategy 2. Two regions performed equally well (South Kivu and Tshuapa) in both strategies and only one region (Kabarole) saw an increase in errors (Table 3). For all regions, TN prediction errors ($RMSE_{pred}$) were consistently lower in strategy 2 than strategy 1 (Table 3). The $R^2_{pred}$ of the TC and TN predictions indicate that the precision of such models was, in general, equal or slightly better for strategy 2 than for strategy 1.

### 3.2.3 Strategy 3: Spiking of the regional models

For all regions, spiking the regional models with up to 20 local samples from each corresponding regional spiking set $K_i$ consistently produced lower prediction errors (Figure 5) compared to strategy 1 and strategy 2. For Haut-Katanga, Tshopo, Tshuapa, and Iburengerazuba the $RMSE_{pred}$ for TC and TN could be reduced with 10 to 13 spiking samples and did not change substantially thereafter (Figure 5). In contrast, for South Kivu and Kabarole, $RMSE_{pred}$ values were minimized with 16 or more spiking samples from each target region (Figure 5). To present the strong and contrasting effect of foregoing any spatial extrapolation in strategy 3, the results for 20 spiking samples are presented in Table 3 and Figure 4. The strongest reduction of the $RMSE_{pred}$ for TC in strategy 3 (with 20 spiking samples) compared to strategy 2 (no spiking) was achieved for Kabarole (4.44 $\mathrm{g\,kg^{-1}}$), Iburengerazuba (1.62 $\mathrm{g\,kg^{-1}}$) and South Kivu (1.56 $\mathrm{g\,kg^{-1}}$), followed by Tshuapa, Haut-Katanga, and Tshopo which decreased by 0.45–0.93 $\mathrm{g\,kg^{-1}}$. Similarly, shifting from strategy 2 to 3 had the strongest effect on the $RMSE_{pred}$ for TN for South Kivu (0.2 $\mathrm{g\,kg^{-1}}$), for Kabarole (0.2 $\mathrm{g\,kg^{-1}}$) and for Iburengerazuba (0.15 $\mathrm{g\,kg^{-1}}$), whereas differences were smaller for Haut-Katanga, Tshuapa, and Tshopo (0.03–0.06 $\mathrm{g\,kg^{-1}}$). Strategy 3 also resulted in predictions that better represented the measured values (consistently higher $R^2_{pred}$ and $RPIQ_{pred}$ values than in strategy 1 or 2; Table 3). Kabarole region showed the best predictive performance for TC in strategy 3 ($RPIQ_{pred}$ of 5.48), followed by Iburengerazuba, South Kivu, Haut-Katanga, and Tshuapa ($RPIQ_{pred}$ 2.22–3.57). For TN, Iburengerazuba, Kabarole, South Kivu, Tshuapa, and Haut-Katanga showed accurate predictions ($RPIQ_{pred}$ of 1.87–4.45). $RPIQ_{pred}$ values for the predictions of TC and TN for Tshopo were less than 2 ($RPIQ_{pred}$ TC: 1.43 and $RPIQ_{pred}$ TN: 1.62). However, the trend from strategy 1 to strategy 3, was a clear reduction in prediction errors and an increase in accuracy.

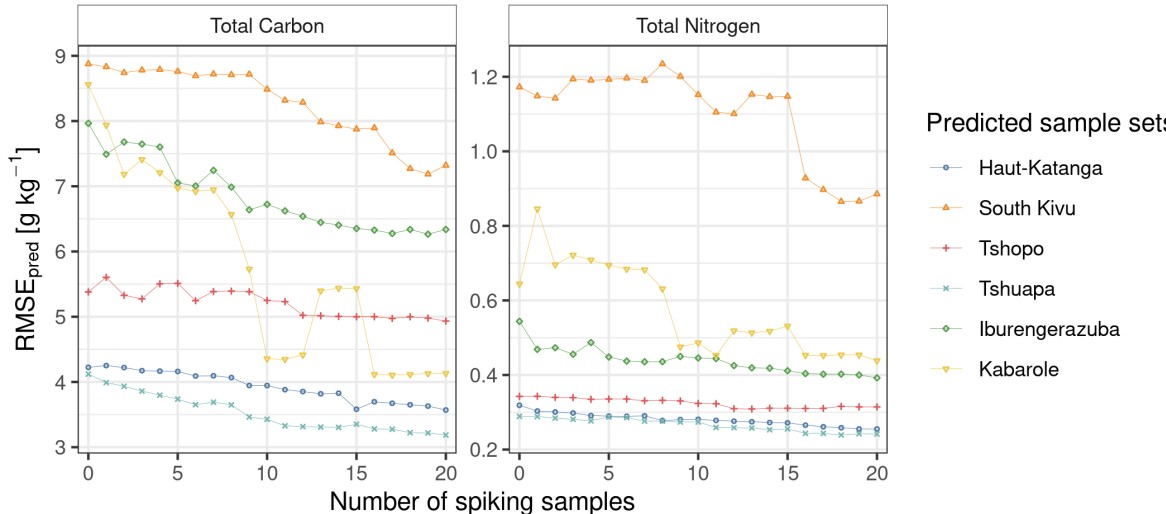

**Figure 5.** Root Mean Square Error of predicted total carbon (left) and total nitrogen (right; RMSE$_{pred}$) for the six regions of central Africa built from pooled continental library (AfSIS SSL) together with the five remaining central African regions and zero up to 20 spiking samples. No spiking samples represents strategy 2 and one up to 20 spiking samples shows strategy 3. The 20 spiking samples were selected from each particular target area and stepwise added to the predictive models in order to find the lowest number of spiking samples that reduces the prediction accuracy to a satisfactory tolerance level.

## 4   Discussion

### 4.1   Strategy 1 and strategy 2: Using soil spectral libraries outside of their respective geographical domains

Our analysis showed that TC and TN in six regions of our CSSL can be reasonably well predicted through the use of existing SSLs comprised of soils from completely different geographical areas and without any local samples using MBL methods (RMSE$_{pred} < 9\,\mathrm{g\,kg^{-1}}$ TC and $< 0.17\,\mathrm{g\,kg^{-1}}$ TN, Table 3). The resulting prediction errors were comparable to other large-scale MIR prediction studies (e.g., Dangal et al., 2019; Angelopoulou et al., 2020) and also to other soil infrared studies, which analyze geographical extrapolation possibilities (e.g., Padarian et al., 2019; Briedis et al., 2020; Gomez et al., 2020). The advantage of using MBL as the method to build prediction models is that it finds similar spectral observations for every new observation to fit suitable models. This approach works efficiently since spectral similarity is in fact reflecting the similarity between observations in terms of soil composition, information which is largely contained in the MIR features of a sample. This means that the predictive success of MBL models largely depends on the quality of the spectra dissimilarity methods used to find spectral neighbors. In other words, MBL can be described as a method driven by compositional similarity search. The improved prediction accuracy (lower RMSE$_{pred}$ and higher RPIQ$_{pred}$) when reducing extrapolation (strategy 2) can be explained by the addition of more proximal central African soil samples to the library that are more similar to each predicted region. The continental AfSIS SSL is missing data for most of central Africa (Figure 1); none of the tropical forest soils with

high contents of organic carbon or with distinctive mineral-organic composition are covered by this large-scale SSL. Naturally, this variability impacts the generalization ability of any predictive model or modeling strategy. Moreover, variance arising from instrument and reference laboratory differences was avoided through the use of local models. However, it is not clear why Kabarole exhibited higher prediction errors in strategy 2. A possible reason could be random variance (Figure 4) or non-linearity. Two regions (South Kivu and Tshuapa) did not show any substantial changes on $RMSE_{pred}$ and $RPIQ_{pred}$ values for TC when comparing strategy 1 and strategy 2. Note that both South Kivu and to some extent also Tshuapa cover a distinct score space in Figure 3 and therefore are not well represented by the remaining central African regions, nor by the AfSIS SSL.

All central African regions from the CSSL show large variability in TC and TN contents (Figure 2) and contain samples from various land cover (forest/croplands), altitudes (Table 2), and parent materials. These differences suggest that soils have developed and been transformed under a variety of environmental conditions. For example, high diversity in organic compounds and their stabilization in soils (i.e. organo-mineral association, complexation, aggregation) can introduce non-linear relationships that are difficult to predict with locally linear calibration methods (i.e., memory-based learning in combination with PLS regression). Thus, we conclude that the particularly high soil diversity in these two regions, in terms of biogeochemical and physical properties, introduces additional complexity in the soil spectral prediction workflow. Similarly high RMSEs have been shown in other studies for samples with organic carbon higher than $150\,\mathrm{g\,kg^{-1}}$ (Nocita et al., 2014). As in our study, these high errors were attributed to high TC contents. To improve predictions for these diverse regions, more data is needed for calibrating the CSSL, and ultimately deliver better regional estimates using local methods. The creation of subsets from large spectral libraries via spectral similarities, for example, has been shown to be effective to train calibration models (e.g., Wetterlind and Stenberg, 2010; Clairotte et al., 2016; Sanderman et al., 2020). Hence, in order to reduce uncertainties for regions in central Africa that are diverse in terms of soil chemical composition, in particular for the Great Lakes region, there is a pressing need to fill the existing gaps in the continental library by gathering more data on the ground.

## 4.2 Strategy3: Effect of spiking with local samples on prediction performance

The effect of spiking of the calibration models with local target samples had a positive effect for all included regions (Figure 5 and Table 3). Kabarole, Iburengerazua, and South Kivu, which showed the most substantial reductions of $RMSE_{pred}$ for TC and TN by spiking, cover different land uses, high altitudes along the Albertine Rift, and larger climatic ranges (Table 2). These soils are not adequately represented by the continental AfSIS SSL, nor by the remaining central African regions, and therefore exhibited a strong effect when spiked with local soil data. Although the effect of spiking on $RMSE_{pred}$ for TC and TN was somewhat smaller for the other included regions (Haut-Katanga, Tshopo and Tshuapa), it still produced noticeable improvements compared to strategy 1 and strategy 2 (smaller $RMSE_{pred}$ and larger $RPIQ_{pred}$ values). The TC and TN ranges of Haut-Katanga, Tshopo, and Tshuapa were narrower and they seem also to be better represented by each other and by the AfSIS SSL (with the exception of a few samples of Tshuapa; Figure 3). In these three regions, sufficiently similar spectra were available and the MBL found the required neighbors to build accurate models and predict TC and TN, and thus lowering the positive effect of spiking. Additionally, the weaker influence of spiking on soils of Tshopo ($RPIQ_{pred}$ TC: 1.43 and $RPIQ_{pred}$ TN: 1.62) can be explained by an outlier in the predictions (Figure 4) and a slightly uneven distribution of the reference data

between the validation and spiking sets (Figure 2). In summary, spiking has already been shown to improve performance (e.g., Guerrero et al., 2014; Seidel et al., 2019; Barthès et al., 2020) and also proved its value in our study. However, a threshold of 20 samples poses non-negligible additional costs for laboratory reference analysis and the benefit in terms of gain of accuracy by spiking depends on the region and is not always guaranteed. In some cases, however, a smaller number of spiking samples can substantially reduce the RMSE$_{pred}$ (e.g. Iburengerazuba and Kabarole). The required prediction accuracy and additional investments depend hereby on the field of application. The achieved predictions and their errors from this study are more than satisfactory for the study of TC and TN dynamics and will improve the availability of high-resolution soil data of central Africa. Thus, spiking is recommended, when soils are highly variable and show large distances to existing spectral libraries.

### 4.3 Suggestions for building new models and extending the existing spectral library

Our regional predictions of TC and TN show promising results when analyzing soils from geographically distinct areas in central Africa that are not covered by the continental AfSIS SSL (Figure 1). Six central African regions were predicted for soil TC and TN with sufficient accuracy using the large-scale AfSIS soil spectral library only. The general positive effect of adding geographically closer samples to the AfSIS SSL (strategy 2) underlines the usability of spectral libraries for new regions. The generally positive effect of strategy 3, spiking of all regional predictions for TC and TN with samples from the target area, encourages the future amendment of currently existing libraries to improve prediction accuracy. To improve future soil analyses and to extend the geographical area covered by an SSL, we suggest the following workflow:

1. **Pre-processing**: Different spectral pre-processing methods influence model and prediction performance. We suggest selecting the best pre-processing strategies using spectral projections and minimizing the reconstruction error (see subsection 2.4).

2. **Estimate uncertainty for new samples**: When analyzing new soil samples from a region which is not covered by the existing SSL, samples with different composition and hence chemical properties are more likely to be introduced. Samples with high distances in the score space to the SSL cannot be predicted accurately with a high certainty, since they are often highly divergent from the SSL. We recommend that a preliminary graphical inspection of resampled and pre-processed spectra can already allow for recognition of differences. A further dimension reduction (e.g. with a PCA) with a subsequent 2D or 3D visualization of the first factors provides additional insights into dissimilarity.

3. **Reference analysis for independent validation**: If the new samples are from a completely new region or the new sample set tends to differ from the SSL, a certain number of validation samples is recommended to test for prediction accuracy. The number is dependent on the similarity/dissimilarity to the SSL.

4. **Search for nearest neighbors and train a model**: run an MBL algorithm to find the nearest neighbors of the new set and train a subsequent weighted average PLS regression.

5. **Model validation**: For predicting soil TC and TN and quantifying the error of these predictions in new geographical regions, a new model validation is required. The nearest neighbor validation is a suitable method, as demonstrated in this study.

6. **Make data and libraries available to the community:** The created CSSL is freely available to use and build upon at our GitHub repository (https://doi.org/10.5281/zenodo.4351254). As shown with the AfSIS SSL, the application of already existing libraries and the extrapolation to new regions is accurate and suitable to estimate soil properties. However, to make predictions more accurate, especially for more diverse, heterogeneous and complex soils, more data is required. As demonstrated, the addition of new geographical regions improves the overall prediction accuracy when more proximal central African regions were added to the large-scale library. These results encourage the use and amendment of existing libraries, rather than the construction of new, separate, and extensive databases. Given the existing distribution of samples in the new CSSL, it is especially important to increase the number of forest soils with high TC contents, which represent a large portion of the Congo Basin. The future enlargement of the CSSL, preferably facilitated by our suggested workflow, is crucial to fill the gap of soil information in this highly understudied part of the world and can be assisted by the soil science community by adopting a sharing-oriented open data policy.

## 5 Conclusions

Our study presents the results and workflow for building the first central African SSL for predicting soil properties (TC and TN) using lab based MIR spectroscopy in a crucial but understudied area of the African continent. Extrapolations were possible for central Africa and for all the six selected regions. Our results further demonstrate how MBL algorithms are useful to find spectral similarities and reduce the need for spiking when a new set covers the same score space as the existing library. These encouraging insights highlight the utility of spectral libraries for future applications, since they are not necessarily limited to certain geographical areas. Our approach of augmenting a smaller SSL with a continental SSL, even when scanned on different instruments, leads to reasonably accurate predictions for new regions which allows analyses of TC and TN dynamics in soils, but also meets a competitive cost-benefit trade-off. Furthermore, the CSSL fills an appreciable continental gap of the continental scale AfSIS SSL and contributes to cover an important range of soil variability with spectral data, particularly from tropical forests. However, in order to improve the accuracy of predicting soil organic matter across regions, especially for soil compartments with high TC and TN contents, our study highlights the need to extend the existing library into new regions. The inclusion of more samples and regions, in particular with more (varying) data of humid tropical forest soils is crucial to fill existing gaps. Combining spectral libraries will allow fast analyses of soil samples and provide spatially explicit data across humid tropical Africa.

*Code and data availability.* Data and R codes are available on our GitHub repository 'ssl-central-africa' and can also be found under Zenodo with the DOI 10.5281/zenodo.4351254 to reproduce our results presented in the submitted manuscript.

# Appendix A: Supplementary Figures and Tables

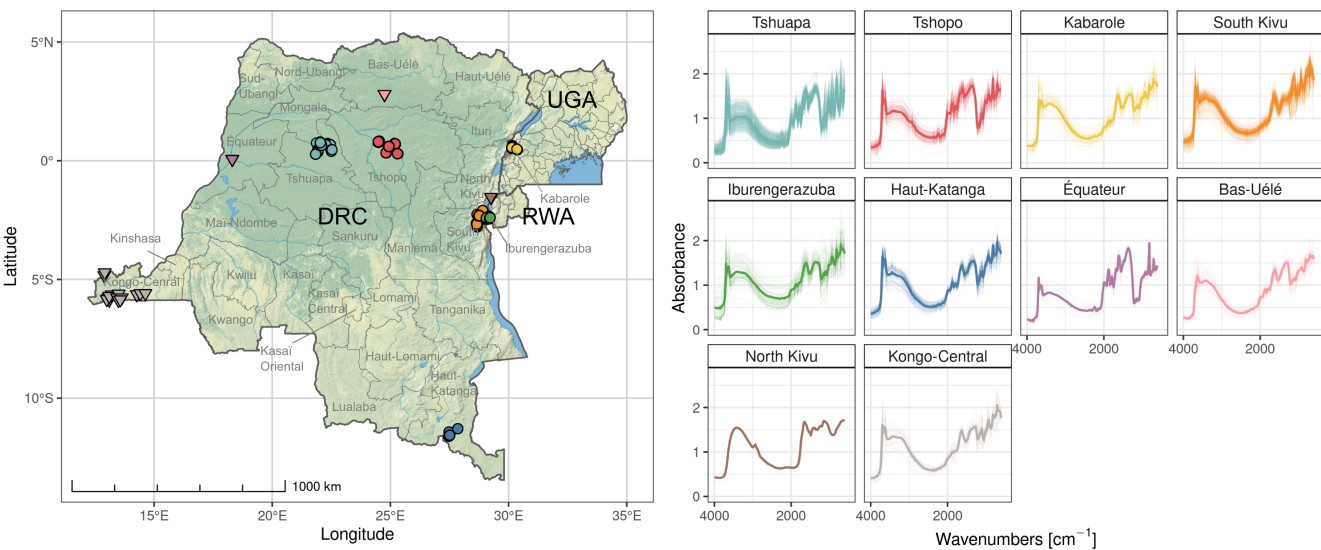

**Figure A1.** Locations and resampled spectra for the sampling regions (six selected central African regions with a ○ symbol and the remaining four regions with a ▽ symbol). All samples are included in the archive of the spectral library for central Africa. For the Democratic Republic of Congo (DRC) and Rwanda (RWA), the regions correspond to provinces, for Uganda (UGA), the sampling region corresponds to a district (left). The average spectra of each region are shown (bold line) along with the individual sample spectra (transparent lines; right).

**Table A1.** Number of samples, GPS coordinates, elevation, annual precipitation (AP), mean annual temperature (MAT), Koeppen-Geiger climate classifications and soil types for entire soil spectral library for the Democratic Republic of Congo, Rwanda and Uganda. Data were extracted for all coordinates from raster files: Climate data is sourced from Fick and Hijmans (2017), elevation from SRTM (90m resolution; Jarvis et al. (2008)), Köppen-Geiger climate classifications from Beck et al. (2018) and soil types from the *Soil Atlas of Africa* (Jones et al., 2013; IUSS Working Group WRB, 2015)

| Region | $n$ | Longitude (° E) | Latitude (° N) | Elevation (m) | MAT (°C) | AP (mm) | Köppen-Geiger | Soil types |
|---|---|---|---|---|---|---|---|---|
| Haut-Katanga | 119 | 27.48–27.85 | -11.61– -11.29 | 1197–1323 | 20.6 | 1223 | Cwa | Rhodic/Haplic Ferralsols |
| South Kivu | 369 | 28.64–28.91 | -2.79– -2.1 | 1487–2310 | 17.6 | 1627 | Cfb, Csb, Aw, Cwb | Umbric Ferralsols, Haplic Acrisols |
| Tshopo | 315 | 24.48–25.32 | 0.29–0.83 | 380–506 | 24.9 | 1789 | Af | Xanthic/Haplic Ferralsols |
| Tshuapa | 738 | 21.84–22.53 | 0.28–0.8 | 385–578 | 24.7 | 2090 | Af | Xanthic/Haplic Ferralsols |
| Iburengerazuba | 107 | 29.05–29.22 | -2.47– -2.34 | 1565–1939 | 17.6 | 1496 | Csb, Aw, Cwb | Haplic/Umbric Acrisols |
| Kabarole | 101 | 30.13–30.37 | 0.46–0.63 | 1271–1824 | 19.7 | 1360 | Af, Cfb, Am | Haplic Phaeozems, Rhodic Nitisols, Albic Luvisols |
| Équateur | 12 | 18.31 | 0.06 | 322 | 25.5 | 1685 | Af | Eutric Ferralsols |
| Bas-Uélé | 49 | 24.75 | 2.8 | 423 | 25.2 | 1641 | Aw | Haplic Ferralsols |
| North Kivu | 4 | 29.25–29.27 | -1.55– -1.53 | 2276–3250 | 12.8 | 1834 | Cfb | Umbric Silandic Andosols |
| Kongo-Central | 40 | 12.89–14.63 | -5.88– -4.71 | 30–470 | 25.5 | 1088 | Aw | Ferralic Cambisols, Haplic Acrisols, Umbric Nitisols, Xanthic Ferralsols, Mollic Gleysols |

**Table A2.** Summary of the reference data for total carbon (TC) and total nitrogen(TN) of the two soil spectral libraries for central Africa (CSSL) and for continental Sub-Saharan Africa (AfSIS SSL).

| SSL | Covered region | TC [g kg$^{-1}$] | | | | | TN [g kg$^{-1}$] | | | | |
|---|---|---|---|---|---|---|---|---|---|---|---|
| | | $n$ | Mean | Median | Min | Max | $n$ | Mean | Median | Min | Max |
| CSSL | Haut-Katanga | 119 | 11.31 | 9.66 | 1.26 | 34.66 | 119 | 1.10 | 1.04 | 0.44 | 2.92 |
| | South Kivu | 367 | 35.37 | 29.28 | 5.98 | 182.10 | 368 | 3.06 | 2.40 | 0.68 | 15.02 |
| | Tshopo | 134 | 13.84 | 12.37 | 4.00 | 56.69 | 149 | 1.02 | 0.90 | 0.20 | 4.46 |
| | Tshuapa | 738 | 12.64 | 11.64 | 3.71 | 47.42 | 738 | 0.95 | 0.87 | 0.17 | 3.94 |
| | Iburengerazuba | 104 | 26.31 | 22.69 | 1.49 | 93.85 | 104 | 1.73 | 1.54 | 0.11 | 5.48 |
| | Kabarole | 100 | 17.69 | 11.95 | 0.77 | 53.76 | 100 | 1.68 | 1.21 | 0.11 | 5.05 |
| | Équateur | 12 | 13.17 | 10.19 | 1.24 | 50.53 | 12 | 0.75 | 0.75 | 0.23 | 1.37 |
| | Bas-Uélé | 49 | 10.93 | 9.64 | 2.73 | 28.37 | 49 | 0.87 | 0.73 | 0.24 | 2.25 |
| | Nord-Kivu | 4 | 310.16 | 319.67 | 189.65 | 411.65 | 4 | 19.32 | 18.04 | 11.96 | 29.24 |
| | Kongo-Central | 40 | 16.78 | 12.41 | 3.36 | 54.96 | 40 | 1.38 | 1.18 | 0.44 | 4.88 |
| AfSIS SSL | Sub-Saharan Africa | 1902 | 12.37 | 7.82 | 0.84 | 112.88 | 1902 | 0.82 | 0.53 | 0.04 | 6.59 |

*Author contributions.* J.S. conceived the study. L.S., P.B. and L.R.-L. were the main contributors to the conceptualization, methodology (modeling strategies) and data analyses. MattiB. supported the conceptualization, provided technical support and project coordination. P.B., L.R.-L., S.D., MattiB., MarijnB. and J.S. helped with writing of the manuscript. L.S., MarijnB., B.B., M.R., P.B., E.K., K.V.O., B.V., D.C., A.B.H, P.M., A.S., K.S., B.B.M., E.V.R., G.B., S.D. substantially contributed to the work by organizing, preparing and providing soil samples and the corresponding reference data. All co-authors revised the manuscript.

*Competing interests.* The authors declare that they have no conflict of interest.

*Acknowledgements.* We are truly grateful to all collaborators in the Democratic Republic of Congo, in Rwanda and in Uganda who made this study possible and helped us with the organization and coordination of numerous field campaigns. We would like to express our gratitude to all the farmers and their families for their hospitality and help during the soil sampling. We would like to warmly thank the teams of the International Institute of Tropical Agriculture (IITA) in Bukavu (Kalambo), in Kinshasa and in Nairobi, the World Agroforestry centre
(ICRAF) in Nairobi, the University of Lubumbashi, the Catholic University of Bukavu, the Mountains of The Moon University Fort Portal and of the African Wildlife Foundation (AWF) in Kinshasa for their support and generosity. Moreover, we acknowledge the research funding organisation in Germany (DFG; Deutsche Forschungsgemeinschaft) for their funding for the TropSOC project. Additionally, we would like to acknowledge the Walter Hochstrasser foundation for their generous financial support of a unique master thesis for six months in Djolu,

the province of Tshuapa. Thanks also to Heather Maclean and Travis Drake, for the additional editing of the manuscript. Finally, we would like to thank the handling editors of SOIL and two anonymous reviewers for their valuable comments and insights that helped us to improve the manuscript and guided us during the review process.

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
