# Peer review of "The Central African Soil Spectral Library: A new soil infrared repository and a geographical prediction analysis"

_SOIL, 2020_

## Author Comment (AC1)

**1 Review 1**

**2 GENERAL COMMENT**

The paper presents MIRS predictions of soil total C and N concentrations (TC, TN) in six regions of
Central Africa separately, using the AfSIS Sub-Saharan library with no Central African soils (Strategy
1), possibly completed with the samples from the five other regions (Strategy 2), possibly completed
with spiking samples from the same region (Strategy 3). This is done with the Memory-based learning
(MBL) regression procedure, which uses spectral calibration neighbors for building a PLS regression
for each target sample individually.

This is very interesting, but the paper suffers several drawbacks. Some methodological aspects are
not presented (selection of the number of latent variables in global calibrations developed for
optimizing spectral pretreatment; window size for calculating spectral similarity; possible cut-off value
for spectral similarity; minimum and maximum number of latent variables for calculating weighted
average predictions) or not discussed (pretreatment selection on X residues instead of Y residues, as
usually; forcing spiking samples into neighborhoods; why not testing a strategy without AfSIS dataset,
to evaluate its usefulness), some terms are not introduced/defined (hold-out and validation sets;
MEpred; notion of accurate prediction), and some points are unclear (what were Central African
samples out of the six core regions used for? why were AfSIS sentinel sites divided into hold-out and
validation sets?). Some results are misinterpreted (using RMSE for comparing predictions between
regions with different distributions of TC or TN; differences between strategies), others are not
presented in the text (effect of the number of spiking samples) or not discussed (negative effects of
Strategy 2 in several cases), and conclusions often seem too optimistic ("accurate predictions" etc.
while error represented >=30% of observed mean in most cases).

For these reasons, I recommend moderate revision.

We thank the reviewer for the detailed comments and constructive criticism. We fully agree that there
is some methodological information missing, which we will add accordingly or discuss in the
corresponding comments below. We also acknowledge that some terms need further explanation and
the structure of the presentation needs to be more clear. We will therefore only present the six regions
we actually worked with in our manuscript and will add a table for the entire spectral library in the
annex. We also agree that there were some misinterpretations that arose from considering only
RMSE$_{pred}$. To analyse the predictions between the regions, we will use the RPIQ$_{pred}$ instead, and only
when comparing strategies within the same region, we use the RMSE$_{pred}$. Additionally, we propose to
replace Table 3 by a figure (see below), which visually depicts each distribution of total carbon and
total nitrogen contents, including a boxplot, showing the interquartile ranges. Furthermore, we discuss
the partially negative effect of strategy 2 compared to 1 in detail and also report and discuss the effect of spiking in detail. Overall, we are convinced that we will be able to implement these changes as discussed below and it will improve the readability and quality of the manuscript.

Please note that in the answers we provide here we do not distinguish between the comments made by the reviewer in capital letters and comments made in lowercase letters.

SPECIFIC COMMENTS

The title is short, which may be an advantage, but I wonder if it is informative enough; moreover the genericity of the work is not highlighted (i.e. using a large spectral library for predictions in poorly documented areas).

The actual title is a general title meant for a broad audience interested in quantitative soil assessments, who may not necessarily be experts in spectroscopy. However, we agree that the title should be more specific and will change it accordingly:

**A new soil infrared library for central Africa and a geographical prediction analysis.**

L8-9. What was done with the six core regions, and what the three levels of extrapolation consisted of, should probably be specified a little bit. Moreover, specifying the size of AfSIS SSL would be useful.

We agree with the reviewer that the strategies in the abstract lacked some details described in the abstract and will implement the following changes starting from line 7 (see below):

"*For the analysis, we used six regions from the CSSL, which we predicted using an existing*

*continental SSL (African Soil Information Service, AfSIS; n = 1902) that does not include central*

*African soils. We explored three different strategies, at decreasing degree of geographic*

*extrapolation, to predict total carbon (TC) and total nitrogen (TN) contents of the six selected regions*

*using models built with (1) the AfSIS SSL only, (2) AfSIS SSL combined with the five remaining*

*regions, and (3) a combination of AfSIS SSL, the remaining five regions, and selected samples from*

*the target region.*"

L13-14. Improvement was not clear for TC, from RMSE=0.38-0.86% to 0.41-0.89%. more details?

split up into regions? Moreover, I wonder if such prediction errors allow considering the approach as particularly useful (i.e. is information ACCURATE ENOUGH?). Note that RMSE is not particularly informative as long as distribution has not been specified (e.g. RMSE=3 is small if mean=30 and

SD=10, but high if mean=10 and SD=5), so adding RPIQ would be useful.

We agree with the reviewer that the presentation of these ranges does not show improvements. This is now clarified in the abstract (see above); we will change the abstract as proposed above and will suggest that readers assess the cost-benefit of investing in new sampling versus gaining accuracy.

L38. Cost is one reason, there are probably others.

The reviewer is correct, there are numerous other reasons for missing soil data in central Africa,
including but not limited to accessibility to sampling areas, infrastructure, and political instability. We
will include these other factors in the revision.

L52-53. The notion of "positive predictive transfer" is unclear for me.

We thank the reviewer for the comment and we agree that this notion was not clearly formulated. With
"positive predictive transfer" we describe the information transferred from a large infrared library for a
new calibration of a local set as described by Padarian et al. (2019). The calibration of a new local set
using a large-scale spectral library can be complex in soil science, especially when the local set
covers a different geographical domain than the library. Soil spectral libraries become particularly
useful when a large amount of their relevant information can be extracted in a way that it improves
prediction accuracy (positive transfer) and minimizes the number of additional costly local reference
measurements for quantifying soil properties in the local set (accuracy-cost trade-off). To avoid
technical jargon we will rephrase the paragraph L48-L55 and move it to L60, where it fits better into
the context:

*"One of the main aims of establishing large-scale SSLs is to minimize the need for future wet*
*chemical analyses (e.g., Nocita et al., 2014; Stevens et al., 2013; Shi et al., 2014; Viscarra Rossel et*
*al., 2016). However, these libraries often span vast geographical areas that include different soil types*
*and climate zones, which comprise complex soil organic C forms and mineral compositions. Due to*
*this heterogeneity, predictions rendered by global linear regression models are often unfeasible for*
*new local soil property assessments at a regional, field or plot-scale, especially when the new set*
*covers another geographical domain than the library. Pandiran et al. (2019) could considerably*
*improve prediction accuracies for a new local set when using a compositionally related subset from a*
*large-scale SSL together with a small number of local reference analyses. The cost-accuracy trade-off*
*can be met when the accuracy of the library-based prediction is similar to the one made when*
*applying a local but more costly calibration strategy. Several data-driven methods have proven to be*
*successful to overcome this issue, for example RS-LOCAL (Lobsey et al., 2017) and memory-based*
*learning (a.k.a local learning e.g. Ramirez-Lopez et al., 2013; Shenk et al., 1997; Naes 1990). In*
*addition, other promising approaches have also been proposed, although they require more research*
*(e.g. deep learning (Ng et al. 2019), fuzzy rule-based systems (Tsakiridis et al. 2019))."*

Padarian, J., Minasny, B., McBratney, A.B.: Transfer learning to localise a continental soil vis-NIR
calibration model, Geoderma, 340, 279-288, https://doi.org/10.1016/j.geoderma.2019.01.009, 2019.

L64-67. LOCAL and Locally weighted PLSR should probably be cited, as they also aim at selecting
spectral calibration neighbors, and were used earlier in soil spectroscopy.

We agree with the reviewer and propose the following changes, together with the next comment (L64-
70) (see below).

L64-70. In my opinion, approach complexity should be considered: some approaches are rather
simple (e.g. spiking) thus widely usable, while others are complex thus usable only by experts (e.g.
the fuzzy rule-based system proposed by Tsakiridis et al. 2019).

We will rephrase the paragraph as following and add two references to the reference list:

*"Several data-driven methods have proven to be successful in overcoming this issue, for example RS-*
*LOCAL (Lobsey et al., 2017) and memory-based learning (a.k.a local learning e.g. Ramirez-Lopez et*
*al., 2013; Shenk et al., 1997; Naes 1990). In addition, other promising approaches have also been*
*proposed, although they require more research (e.g. deep learning (Ng et al. 2019), fuzzy rule-based*
*systems (Tsakiridis et al. 2019))."*

Naes, T., Isaksson, T., & Kowalski, B.: Locally weighted regression and scatter correction for near-
infrared reflectance data. Analytical Chemistry, *62*, 664–673, https://doi.org/10.1021/ac00206a003,
1990.

Tsakiridis, N., Theocharis, J., Panagos, P., & Zalidis, G.: An evolutionary fuzzy rule-based system
applied to the prediction of soil organic carbon from soil spectral libraries. Applied Soft Computing, 81,
1-18, https://doi.org/10.1016/j.asoc.2019.105504, 2019.

Ng, W., Minasny, B., Montazerolghaem, M., Padarian, J., Ferguson, R., Bailey, S., McBratney, A.B.:
Convolutional neural network for simultaneous prediction of several soil properties using visible/near-
infrared, mid-infrared, and their combined spectra, Geoderma, 352, 251-267,
https://doi.org/10.1016/j.geoderma.2019.06.016, 2019.

L86. "covers a large geographic area" is questionable as the sample population is clustered, and a
wide area is not represented (i.e. between Kinshasa, Tshopo and Katanga).

The reviewer is correct! The sampling locations did not cover the entire area and the term is
potentially misleading. We will address this comment in line 86 accordingly:

*"The sample locations are clustered in eight regions distributed over a large geographical area of*
*central Africa, from a latitude of …"*

L99. The way samples were dried should be specified, moreover they had probably been 2-mm
sieved previously.

We thank the reviewer for requesting this information. The samples were all sieved through a 2 mm
mesh and either air dried or oven-dried at temperatures of 50 °C, 60 °C or 105 °C, all of them suitable
for total carbon and nitrogen analyses. After sieving and drying, soil samples were ground to a powder
(< 50 µm) using a ball mill. We will include these details in the revised manuscript

Tab.2. I've not understood how samples from Equateur, Bas-Uélé, North Kivu and Kongo-Central
were used (they are not mentioned in Strategy 2, L204-205).

The regions Equateur, Bas-Uélé, North Kivu and Kongo-Central were excluded for the further
analyses because they did not have enough samples to allow for reliable analysis (< 80 samples per
region). With this table, we intended to present the entire infrared library we created. However, we
fully understand that this is confusing here and we will remove these regions from this table but
present the full library (including these four regions: Équateur, Bas-Uélé, North Kivu and Kongo-
Central) in a supplementary table in the appendix.

L106-107. Does this suggest charcoals were considered organic, or negligible?

This is a legitimate question, since slash-and-burn is commonly used to clear fields in central Africa
which adds charcoal to the topsoils. For our soil analyses, visible pieces of charcoal were removed,
which could clearly influence TC measurements in certain samples. This detail will be added in the
methods.

L112. SPECIFYING PARTICLE SIZE WOULD BE USEFUL (< 0.2 mm? < 0.1 mm?).

All samples were grinded to a powder (<50 μm) using a ball mill, which is sufficiently accurate for soil
spectral diagnostics. Diess et al. (2020) report sufficiently accurate model estimates when grinding
below 0.5mm, and Guillou et al. (2015) even report no significant differences at particle size
thresholds of 1.0mm, 0.5mm and 0.25mm thresholds. We will add this information to the method
section.

Deiss, L., Culman, S. W., & Demyan, M. S.: Grinding and spectra replication often improves mid-
DRIFTS predictions of soil properties, Soil Science Society of America Journal, 84, 914–929.
https://doi.org/10.1002/saj2.20021, 2020.
Guillou, F. L., Wetterlind, W., Viscarra Rossel, R. A., Hicks, W., Grundy, M., & Tuomi, S.: How does
grinding affect the mid-infrared spectra of soil and their multivariate calibrations to texture and organic
carbon? Soil Research, 53, 913-921, https://doi.org/10.1071/SR15019, 2015.

L113, L125. Spectral range and resolution should probably be specified.

We thank the reviewer for bringing this to our attention. We fully agree and will change the sentences
accordingly:

*All samples were measured with a VERTEX70 Fourier Transform-IR (FT-IR) spectrometer with a High*
*Throughput Screening Extension (HTS-XT) (Bruker Optics GmbH, Germany) in order to measure their*
*MIR reflectance spectra. Spectra were acquired in a resolution of 2 $cm^{-1}$ within a range of 7500 $cm^{-1}$*
*to 600 $cm^{-1}$, which corresponds to a wavelength range of 1333 nm to 16667 nm. A gold coated*
*reflectance standard (Infragold NIR-MIR Reflectance Coating, Labsphere) was used as a background*
*material for all measured soils in order to normalize the sample spectra. Reflectance was transformed*
*into absorbance using log(1/reflectance) prior to further processing and subsequent modeling.*

L125. Spectra were collected on AfSIS and CSSL samples with different spectrometers, so the
question of compatibility should be addressed (e.g. was there standardization?).

The reviewer raises an important point regarding the compatibility of data form two different spectral
libraries. Luckily, the two instruments were both FT-IR spectrometers from BRUKER which use the
same settings and the same internal standards. The scanning methods of the CSSL were adapted to
the ICRAF standard operating procedures. For these reasons, no instrument standardization was
necessary and all spectra between the libraries can be compared one to one. This information will be
added to the methods section of the revised manuscript.

L132. A reference dealing specifically with soils would probably be more appropriate.

We agree that a more soil specific reference would help to point out the importance of the effect of
pre-processing and we therefore suggest the two following publications:

Seybold, C.A., Ferguson, R., Wysocki, D., Bailey, S., Anderson, J., Nester, B., Schoeneberger, P.,
Wills, S., Libohova, Z., Hoover, D. and Thomas, P.: Application of Mid-Infrared Spectroscopy in Soil
Survey. Soil Science Society of America Journal, 83, 1746-1759,
https://doi.org/10.2136/sssaj2019.06.0205, 2019.

Sila, A. M., Shepherd, K. D., and Pokhariyal, G. P.: Evaluating the utility of mid-infrared spectral
subspaces for predicting soil properties, Chemometrics and Intelligent Laboratory Systems, 153, 92–
105, https://doi.org/10.1016/j.chemolab.2016.02.013, 2016.

L140. p is not defined. Actually P is a d x l matrix, not a d x p matrix.

We thank the reviewer for spotting this typo! "d $\times$ l matrix" is correct and we will change it as
suggested by the reviewer.

L145-161. The error E depends on the NUMBER OF LATENT VARIABLES (l). HOW WAS THIS
PARAMETER DEFINED? Moreover, the EXPECTED BENEFIT OF THIS APPROACH (i.e. computing
Xcssl residues) for optimizing spectral pretreatment SHOULD BE PRESENTED, when compared with
examining RMSE associated with every pretreatment (i.e. computing Ycssl residues, as commonly
done).

We fully acknowledge that this was not clearly explained in the text and will address these issues. We
explain that the analysis of spectral reconstruction error is indeed commonly used in spectroscopy for
outlier identification. This error is also known as the Q-statistic and it indicates how well a given new
sample conforms to the PLS model. Since the response values in the prediction set are unknown, we
can use the Q-statistic as a proxy for the response errors. In the revised version, we will explain that
we assume that if a given set of pre-processing steps lead to large Q-values, then it is expected that it
will also lead to large errors in the prediction of the response values. We will also add references to
support this assumption. In the new version of the manuscript, we will mention that for this analysis
we fixed the number of PLS factors to 20, as projected variables beyond this dimension did not capture a considerable amount of the original spectral variance. For example, PLS variable 21
amounted for less than 0.01% of the original variance in all the cases.

L165."spectral matrices which can be properly represented by a PLS model" is unclear. Moreover, the
assumption that SIMILAR PRETREATMENTS OPTIMIZED GLOBAL AND LOCAL CALIBRATION
SHOULD BE DISCUSSED (e.g. according to literature).

The ideas behind this sentence will be clarified with the description of the Q-statistic (see previous
reply L145-161) and the advantages of its use for pre-processing optimization. We indicate now that
according to Wise and Roginsky (2015), large $Q_c$ values are proxies to large prediction errors and
therefore Q-statistic can be used to judge the suitability of a set of pre-processing steps.

Wise, B. M., & Roginski, R. T.: A calibration model maintenance roadmap. IFAC-PapersOnLine, 48,
260-265, https://doi.org/10.1016/j.ifacol.2015.08.191, 2015.

L170. The problem with multiplicative scatter correction is that the transformed spectrum depends on
the spectrum population it belongs to, so changes when this population changes.

We thank the reviewer for raising this concern but do not see this as a problem. Multiplicative scatter
correction (MSC) aligns or rotates a given spectrum towards a reference one which is fixed. This
reference spectrum can be seen as a parameter of the MSC transformation. By doing this,
multiplicative and additive shifts between spectra are removed. Although, in many applications the
average spectrum of the calibration set is used as the reference one, in theory any spectrum can be
used (See Rinnan et al., 2009). Therefore, MSC is not necessarily affected by changes in the spectral
population. The reference spectrum parameter of a defined MSC step should not be modified as long
as it guarantees successful removal of the multiplicative and additive scattering effects across the
spectra.

Rinnan, Å., Van Den Berg, F., & Engelsen, S. B.: Review of the most common pre-processing
techniques for near-infrared spectra, TrAC Trends in Analytical Chemistry, 28, 1201-1222.
https://doi.org/10.1016/j.trac.2009.07.007, 2009.

L195. Why 20 spiking samples per regional set, not 10 or 30?

We agree with the reviewer, that the selection of the number of spiking samples has not been
adequately described in the manuscript. Generally, the number of spiking samples should be
minimized to reduce costs for laboratory reference analyses. We set the maximum number of spiking
samples to 20, which can already mean quite a high financial investment but we feel that it is worth
these costs given the reduction of geographical extrapolation and the effect of using spatially close
samples on the predictive performance. We tested one to 20 spiking samples and compared the
prediction accuracy, which was on average best with 20 spiking samples (Figure 5). We will add more
details about the spiking effect in the results and discussion sections of the manuscript.

L197. The way k-means works could (should?) be briefly presented.

Since this method is widely used and well documented in pedometrics and chemometrics for sampling
calibration datasets, we considered that it was sufficient to refer the reader to other studies, where k-
means sampling is explained. However, we agree that it is useful when we explain it in a sentence
and will change it as following:

*"... for each complete regional set, 20 samples were selected using the k-means sampling algorithm.*
*This sampling strategy is implemented in the R package prospectr (Stevens and Ramirez-Lopez,*
*2020) and selects one sample per cluster calculated with a k-means algorithm on a principal*
*component analysis of the pre-processed spectra (Næs, 1987).*

L199. The strategies considered are: AfSIS alone; AfSIS +other Gi; AfSIS +other Gi +Ki. Other
strategies would have been interesting: only using other Gi, or other Gi + Ki, to EVALUATE THE
USEFULNESS OF AfSIS (which would be very interesting); AfSIS +Ki, to evaluate the usefulness of
other Gi; Ki only, to evaluate the usefulness of AfSIS and other Gi. But this would require much
additional work!

Our aim was to propose strategies that could leverage the use of the AfSIS spectral library to
accurately predict soil properties in regions which are poorly covered by it. Therefore, we only
evaluated modeling approaches that involved the use of this library. There is clear evidence that very
accurate soil predictions can be achieved by using models built only with samples originating from the
same region or area where these predictions are required. This is because large non-linear
complexity is avoided in local-scale models (See e.g., Tziolas et al., 2019). Despite this, we consider
that this implies that every undersampled region will require a representative calibration sample set
which might be expensive or impractical. In this respect, the evaluation of models using only other Gi,
or only other Gi + Ki was not considered as they do not really solve the problem of using a large
spectral library in poorly sampled areas.

Tziolas, N., Tsakiridis, N., Ben-Dor, E., Theocharis, J., & Zalidis, G.: A memory-based learning
approach utilizing combined spectral sources and geographical proximity for improved VIS-NIR-SWIR
soil properties estimation,Geoderma, 340, 11-24, https://doi.org/10.1016/j.geoderma.2018.12.044,
2019.

L217-219. HOW WAS w DEFINED? Moreover, WHAT p STANDS FOR IS NOT CLEAR: it has not
been defined, but according to L140, was apparently used in place of l (number of latent variables);
but I'm not sure this makes sense here. Furthermore, I'm not sure to understand what k=1 means. I
also note that d has already been used (number of wavelengths; L139). So CLARIFICATION IS
REQUIRED. We might also wonder why evaluate dissimilarity (1-S) and not similarity (S), when the
objective is to select calibration samples *similar* to the target sample (cf. L311). Furthermore, I
WONDER IF A SIMILARITY/DISSIMILARITY CUT-OFF VALUE WAS DEFINED, below/above which
spectra were not considered neighbors (i.e. no prediction for target samples with too few neighbors);
and if yes, how this cut-off value was defined.

We are thankful that the reviewer noticed the use of letters for multiple variables, which is misleading.

Again, the reviewer is correct, spotting the mistake in L140, which leads to confusion in L215-219.

Correcting this as suggested above, this issue should be resolved here.

The window size (w) was optimized based on a spectral nearest-neighbor search within the AfSIS

library. For every sample in the AfSIS library, its closest sample (in the spectral space) was identified.

The samples were compared against their closest samples in terms of TC and TN and the root mean squared differences (RMSD) were computed according to the following equations:

$$j(i) \; = \; NN(Xc_i, Xc^{-i})$$

and

$$RMSD \; = \; \sqrt{\frac{1}{2m} \sum_{i=1}^{m} \sum_{h=1}^{2} (yc_{i,h} - yc_{j(i),h})^2}$$

where $Xc$ is the spectra of the AfSIS library, $NN(Xc_i, Xc^{-i})$ represents a function to obtain the index of the nearest neighbor observation of the ith sample found in $Xc$ (excluding the ith sample), $yc_{i,h}$ is the value of the i-*th* observation for the *h*-th property variable (either TC or TN). A total of 10 window sizes were evaluated (from 31 up to 121 in steps of 10). According to the RMSDs obtained, the optimal *w*

was 71.

Concerning the comment about using the concept of similarity or dissimilarity, we believe that is not actually relevant. It is clear that similarity or dissimilarity measures can be both used to identify similar samples. Many examples of the use of correlation dissimilarity for nearest neighbor identification can be found in the NIR spectroscopy literature (See for example Wadoux et al., 2021; Khosravi et al.,

2020; Gholizadeh et al., 2018; Zhu et al., 2011).

Gholizadeh, A., Saberioon, M., Carmon, N., Boruvka, L., & Ben-Dor, E.: Examining the performance of PARACUDA-II data-mining engine versus selected techniques to model soil carbon from reflectance spectra. Remote Sensing, 10, 1172, https://doi.org/10.3390/rs10081172, 2018.

Khosravi, V., Ardejani, F. D., Aryafar, A., Yousefi, S., & Karami, S.: Prediction of copper content in waste dump of Sarcheshmeh copper mine using visible and near-infrared reflectance spectroscopy.

Environmental Earth Sciences, 79, 1-13, https://doi.org/10.1007/s12665-020-8901-0, 2020.

Wadoux, A., Malone, B., Minasny, B., Fajardo, M., McBratney, A.B. (Eds.): Soil Spectral Inference with R: Analysing Digital Soil Spectra using the R Programming Environment, Springer Nature, Cham,

Switzerland, 2021.

Zhu, Z., Corona, F., Lendasse, A., Baratti, R., & Romagnoli, J. A.: Local linear regression for soft- sensor design with application to an industrial deethanizer, IFAC Proceedings Volumes, 44, 2839-

2844, https://doi.org/10.3182/20110828-6-IT-1002.02357, 2011.

L220-225. According to Shenk et al. (1997), the weighted average is calculated over a range of latent
variables, i.e. from a MINIMUM TO A MAXIMUM NUMBER OF LATENT VARIABLES CONSIDERED,
AND THESE PARAMETERS HAVE TO BE SPECIFIED. Moreover, both s1:j and gj are calculated for
the jth latent variable, so writing "s1:j" instead of "sj" is unclerar. Furthermore, Shenk et al. (1997) did
not call this approach "Weighted averaged PLS"; but why not…

As correctly pointed out by the reviewer, details about the WA-PLS are missing in the current version
of the manuscript. We will add the missing information to the text. The weighted average was
calculated using a range of latent variables from 5 to 30 in increments of 1, which we will add to the
manuscript accordingly.

To compute the weights we use the exact same method as described by Shenk et al. (1997, see page
227 of their paper). In the equation used to compute the weights, s1:j represent the root mean square
of the spectral residuals of the query spectrum. The reconstruction is done by multiplying the scores
of the projected query spectrum by the (transposed) loading matrix of the PLS model built from its
neighbor samples. In this multiplication the first $j$ rows of the scores and loading matrices are used.
Using sj instead of s1:j, would wrongly indicate that only the $j$th row of the scores is multiplied by the
$j$th transposed row of the loadings. Furthermore, in the equation we also use the term gj to refer to the
root mean square of the regression coefficients corresponding to the $j$th PLS component. In this case
we do not use the subscript 1:j as we are using only the $j$th row of the matrix of regression coefficients
(instead of the first $j$ rows). We will extend the explanation of this notation for a new version of the
manuscript.

Indeed Shenk et al., (1997) do not explicitly call this regression method "weighted averaged PLS".
Although, what this method  does is to compute a "weighted average of the individual model predicted
values with from the minimum to the maximum number of factors" as explained by Shenk and
Westerhaus (1998) in the following patent filing: https://patents.google.com/patent/US5798526A/en.
Therefore, we do not see the term "weighted averaged PLS" as incorrect in our manuscript.

L230-232. Hold-out and validation sets have not been introduced, so this part is not very clear (e.g.
why dividing regional AfSIS sub-libraries into hold-out and validation sets? L256 and Tab.3 these sub-
libraries were not separated).

We thank the reviewer for spotting this point of confusion. We will clarify this issue as following:
T*he grouping factor was used for the optimization of the nearest neighbor search, i.e. the nearest*
*neighbor cross-validatio*n (see L226) t*o avoid overfitting: keeping the nearest neighbor out, the model*
*was trained with the remaining neighbors which were not from the same region as the hold-out*
*neighbor (region corresponds to the sentinel sites within the AfSIS SSL).*
This will be changed accordingly to avoid a misunderstanding as shown in this comment.

L233. I understand the minimum requested number of neighbors was 150, and the maximum possible
number of neighbors was 500. WHAT IF A TARGET SAMPLE HAD LESS THAN 150 NEIGHBORS?

This is an important question of the reviewer. Of course, a sample could have less than 150 neighbors
in the used spectral library. We tested the minimum number of available neighbors prior training the
final model. We agree that the minimum number of neighbors should have been adjusted downwards
if there would not have been enough neighbors, which was luckily not the case. We will explain this
more in detail in the manuscript.

L236. FORCING SPIKING SAMPLES INTO THE NEIGHBORHOOD of every target sample is
questionable, and the discussion should address this point.

Unfortunately the reviewer does not provide an explanation on why forcing spinking samples into the
neighborhood is questionable.

Spectral Neighbor identification is a mathematical attempt to select soil observations that share similar
compositional characteristics with the observation that requires a prediction. MIR spectra partially
reflect the compositional characteristics of the samples. We assume that soils originating from the
same geographical region might be governed by very similar soil formation processes. This is a
concept of spatial autocorrelation which is widely used (Fortin et al. 2016). Furthermore, it is widely
accepted that the best spectral models (most accurate) that can be built for a given area are those
that are calibrated with samples from the same area (see also comment L199). For these reasons, we
assume that forcing samples of a given area to belong to the neighborhoods of samples from the
same area guarantees that samples originating from similar soil formation processes are included in
the models. Therefore, our approach is not arbitrary as it is expected that these samples improve
prediction accuracy.

Fortin, M.-J., Dale, M.R. and Ver Hoef, J.M.: Spatial Analysis in Ecology. In Wiley StatsRef: Statistics
Reference Online (eds N. Balakrishnan, T. Colton, B. Everitt, W. Piegorsch, F. Ruggeri and J.L.
Teugels). https://doi.org/10.1002/9781118445112.stat07766.pub2, 2016.

Fig.3. Beside orange and green circles, many grey circles were also outside AfSIS black circles, and it
would be useful to mention where they originated from.

The transparency of the black AfSIS symbols is misleading. They seem gray, while the remaining
samples are black, where the density is high. We will increase the transparency and change the style
of the symbols, so that it becomes clear which points belong to the AfSIS library.

L265-267. CRITERIA FOR "GOOD PREDICTIVE RESULTS" HAVE NOT BEEN SPECIFIED. Actually
many results were not so good, especially for TN, especially with Strategy 1 (e.g. RMSE for TC and
TN was >=50% of observed mean for 2-3 regions with Strategy 1, and >=30% of the mean for 4-5
regions with Strategy 2). And ACCORDING TO RPIQ, PREDICTIONS FOR SOUTH KIVU AND
IBURENGERAZUBA WERE OFTEN AMONG THE BEST ONES, so the reasons for considering they
"showed the lowest accuracy levels" should be revised, or at least explained.

The reviewer is correct, we have not introduced criteria to define a good or accurate prediction which
we will add. However, the required prediction accuracy depends on the field of application. We will
work on a method to assess prediction accuracy for a hypothetical new sample set.

As the reviewer points out, RMSE$_{pred}$ is useful to estimate prediction accuracy within the same region
but not to make comparisons between regions since ranges for TC and TN differ between the regions,
especially for Iburengerazuba and South Kivu with forest soils with high TC and TN contents. We
agree that it does not make sense to classify the statistical performance of the South Kivu and
Iburengerazuba regions as poor. Indeed, when looking at the RPIQ$_{pred}$ values, they performed well.
This is due to the large interquartile (IQ) range of these regions compared to Tsuapa and Tshopo,
which exhibited considerably smaller IQ ranges. We thank the reviewer for this careful attention to
these statistical descriptions and will modify the results and discussion accordingly. Moreover, we will
replace Table 3 with the proposed plot (see below), which clearly shows the distribution of TC and TN
in each region including their IQ ranges in the boxplots. They cray coloured line and text indicate the
spiking sets  ($K_i$), the black coloured lines and text represent the six regional sets ($G_i$) after removal of
each $K_i$ and the AfSIS SSL data set $A$. These details to the figure will be added to the caption.

[Figure]

L271-272. RMSEpred is useful for comparing strategies for a given region, but CANNOT BE THE
FIRST PARAMETER CONSIDERED FOR COMPARING PREDICTION ACCURACY BETWEEN

REGIONS WHERE DISTRIBUTIONS OF TC OR TN WERE DIFFERENT. R² describes
proportionality, not similarity; so, though understood by a wide audience, should be used with care.
Comparison between regions should firstly be based on RPIQ, which showed good results for
Kabarole, Iburengerazuba and (for TC) South Kivu and poor results for the other regions, especially
Tshopo for TC and Haut-Katanga for TN.

*Yes we agree and as we detailed in the response above, we will modify the results and discussion*
*such that we only use RMSE$_{pred}$ to compare the same regions across strategies and RPIQ$_{pred}$ and*
*R$^2$$_{pred}$ to compare regions within a given strategy. We suggest the following changes:*

*"The best prediction accuracies for TC were achieved for the regions South Kivu, Iburengerazuba and*
*Kabarole, where RPIQ$_{pred}$ values were between 2.43–3.95, while Tshopo, Tshuapa and Haut-Katanga*
*performed less good with RPIQ$_{pred}$ <= 1.84. For TN, Iburengerazuba and Kabarole performed well*
*with RPIQ$_{pred}$ 2.14 and 2.86, respectively. However, the four other regions Haut-Katanga, South Kivu,*
*Tshopo and Tshuapa exposed smaller RPIQ$_{pred}$ <= 1.37. "*

L277-279. The fact that CENTRAL AFRICAN SAMPLES WERE POORLY REPRESENTED BY AfSIS
SHOULD ALSO BE MENTIONED AS POSSIBLE REASON.

*We agree with the reviewer that this should be highlighted at this point and we will add this*
*accordingly. We also suggest to put Figure A1 (continental map) in the main text and move Figure 1*
*to the appendix.*

L282-283. Again, RMSEpred should not be used for comparisons between regions.

*We agree with the reviewer and will change it as described more in detail in L271–272. We suggest*
*the following changes:*

*"The predictive performance in strategy 2 exhibited errors (RMSE$_{pred}$) ranging between 0.41–0.89 %*
*and 0.03–0.12 % for TC and TN, respectively (Table 4). The most accurate predictions for TC were as*
*in strategy 1 obtained for the regions Iburengerazuba, Kabarole and South Kivu (RPIQ$_{pred}$ > 2.36), but*
*RPIQ$_{pred}$ value of Haut-Katanga was remarkably higher than in strategy 1 (2.30 vs 1.62). Predictive*
*performance for TC of Tshopo and Tshuapa were still below an RPIQ$_{pred}$ of 2.*

*For TN, similarly to strategy 1, prediction accuracy was good for Iburengerazuba and Kabarole. For*
*the regions Haut-Katanga, South Kivu, Tshopo and Tshuapa the RPIQ$_{pred}$ values were higher than in*
*strategy 1, but they were still below 2. "*

L284-286. RMSEpred for TC increased in three regions from Strategy 1 to 2, strongly sometimes,
which is counter-intuitive so should be underlined, and POSSIBLE REASONS SHOULD BE
PROPOSED (as was done for better TN predictions with Strategy 2 than 1).

*The reviewer is correct in that RMSE$_{pred}$ increased for 3 regions, however it only increased by 0.03%*
*in two of the cases. So, in total, from Strategy 1 to 2 the RMSE$_{pred}$ decreased substantially in 3*

regions, barely changed in 2, and increased in one, which in our opinion signals an overall
improvement in performance. At the moment, it appears that the inclusion of the additional CSSL
regions reduced the accuracy of the Kabarole region but it is unclear why the model did not fall back
on the same prediction subset as Strategy 1. This will be investigated and corrected in the revised
manuscript.

L287. Better TN predictions with strategy 2 than 1 "was due", not "might be due".

Thank you, this will be modified accordingly.

L290. RPIQ for TC "tended to be the same" except for Kabarole; but actually RPIQ decreased in
South Kivu and Tshuapa, not much, but this is counter-intuitive.

As detailed above, we will modify the discussion of these results in the text to explain the observed
patterns.

L292. South Kivu was not an exception, as TN prediction was also improved.

Thank you for this correction. We will modify the text accordingly.

Fig.5. THESE RESULTS SHOULD BE PRESENTED in the text, and an optimal number of spiking
samples could be proposed for each region.

We thank the reviewer for requesting that these results be included in the text and will add them to the
revised version.

L309, L317, L391. "Accurately predicted/model" "highly accurate predictions" are OVEROPTIMISTIC,
e.g. when RPIQ <2 or RMSE > mean/2.

We thank the reviewer for this suggestion and will tone down the language to "reasonably accurate".

L317-318. The point is that for TC, Strategy 2 reduced RMSEpred in only 3 out of the 6 regions
considered; so "improved prediction accuracy" is questionable. And POOREST PREDICTION WITH
STRATEGY 2 than 1 FOR 3 REGIONS SHOULD BE DISCUSSED.

We again thank the reviewer for pointing out this idiosyncrasy and as detailed in the responses above
will modify the results and discussion to detail these prediction results.

L322-325. There is STRONG MISINTERPRETATION, as in these two regions, TC (and TN in
Iburengerazuba) was accurately predicted (RPIQ >2.3).

As detailed above, these discussion points surrounding the prediction results will be modified.

L338. These results have not fully presented in the results section.

We thank the reviewer for pointing this out and will detail the spiking results in the results section.

L339. Three regions are cited, not two. Moreover, Strategy 3 yielded highest RPIQ whatever the
region for both TC and TN; and the improvement was strong sometimes, with 10 spiking samples only
(Kabarole and Iburengerazuba).

We thank the reviewer for pointing out this mistake. The text should read "three regions" and we will
remove the word "somewhat" to reflect the strong improvement. We will further modify this section to
say that Strategy 3 had a positive effect on all regions but an even stronger effect on the three regions
we originally listed.

L343-344. For TN in South Kivu, RPIQ increased from 1.1 to 1.6 from Strategy 1 to Strategy 2, so
prediction was noticeably improved.

We thank the reviewer for clarifying this point and will modify the text to say how the prediction
noticeably improved.

L345. "RMSE remained relatively high", but TC and TN were much higher than elsewhere!
Considering RMSE without considering TC and TN distributions leads to misinterpretation.

Indeed, the reviewer is correct. We will contextualize the $RMSE_{pred}$ with the higher TC and TN and
instead focus on the $RPIQ_{pred}$ as a more reliable indicator given the different distributions. We also
use the new graph to show this distribution of TC and TN for each region more precisely (see above).

L345. "slightly" does not seem appropriate: e.g. for Iburengerazuba RPIQ increased from 2.8 to 3.6
for TC and from 3.2 to 4.5 for TN.

We agree with the reviewer that "slightly" is not the correct word and will change it to "substantially".

L349. As said above, the effect of spiking was strong sometimes (Iburengerazuba and Karabole).

We thank the reviewer for pointing this out and will modify the text accordingly.

TECHNICAL CORRECTIONS

L6. 1800 soils or 1800 soil samples?

Soil samples. We will clarify this in the text.

L7. "wider" is not clear for me in "Congo Basin and wider African Great Lakes region".

We will remove the word "wider" from this sentence. Moreover, we will correct "African Great Lakes
region" to Albertine Rift, which is a more precise name for the region.

L10. % is not a SI unit and may cause confusion for comparisons or changes (e.g. TC increased by
5%), so G KG-1 WOULD BE MUCH PREFERABLE.

We thank the reviewer for this comment. We will convert the % unit into the SI unit $g\ kg^{-1}$ as
suggested by the reviewer.

L59. sol vs. soil.

Thank you for pointing out this typo.

L77. Predicting a region is confusing.

The reviewer is correct, this sentence does not make sense. We will specify accordingly in the
updated version of the manuscript:

"... (2) To establish a workflow to accurately predict soils from variable locations within six selected
geographical regions of the CSSL ...

L84. The sentence should be checked (e.g. layers vs. layer).

Thanks for spotting this typo, we will correct the word to the plural form.

Tab.1. Université catholique de Louvain and IITA/ICRAF are not references. Moreover, for the last
reference, 2021a,b would be more appropriate than 2021b,a (this is detail).
We thank the reviewer for pointing out this detail. Indeed Université catholique de Louvain, IITA and
ICRAF are not references. We will remove them and add an additional column named "Data
Contributor".

L103. Total Al, Fe, Ca etc., or some particular fractions?

Total contents of cations have been analysed using aqua regia extractions. We agree with the
reviewer that this should be specified and will add information about the methods for analysing pH,
texture and cations.

L115. In general absorbance = log(1/reflectance), not 1/reflectance.

The reviewer is of course correct about this and it will be corrected accordingly.

L118. I note the manufacture place is mentioned here, which should probably be systematic.

This is correct, we thank the reviewer for seeing this detail. We will remove the place to be consistent
through the entire manuscript.

L134. Actually PLS has most often been defined as Partial least squares.

The reviewer is correct, PLS is an abbreviation, used for Partial Least Squares. We also defined the
term accordingly in the manuscript (L56). With the sentence the reviewer brings up, we do not want to
give PLS another meaning. We rather want to explain that the Partial Least Squares method can also
be described as a projection of latent structures, which has by accident the equal letters and the same
order.

L207. The sentence should be checked.

We agree with the reviewer and will make the sentence clearer.

*"– Strategy 3: This time, strategy 2 was repeated, but in this case, extrapolation was avoided by using*
*the spiking samples from the same geographical region as the region to be predicted;"*

L234, L243. Equation 8? Equations have not been numbered.

We thank the reviewer for spotting this mistake. We will remove "Equation 8" from the text.

Fig.3 is not very readable; projections on PC1-PC2 and PC1-PC3 would probably be more suitable.

We agree with the reviewer, that the 3D plot is not appropriate. We will therefore plot the three
different components as suggested by the reviewer.

[Figure]

Tab.4. What MEpred stands for should be specified.

This is correct, we did not specify ME and will add this information to the methods section.

L275. Tshopo, not Tschopp. Four regions are cited, not three.

We thank the reviewer for clarifying this, and will change it accordingly.

L426-427, L432-433, L436, L439, L445, etc. Are two DOIs or two URLs necessary? I note that non-
DOI URLs do not always work ("error 404", "page not found", etc.).

We thank the reviewer for checking the DOIs and URLs in the reference list. We will check them
carefully.

L442, L445, L508, L540, L564, L567, L570, L573, L584-585, L615, L617-618. Same (or almost same)
DOI mentioned twice.

We will also check these DOIs and remove the ones, which are not necessary. We thank the reviewer
for spotting this issue.

L448, L469, L473, L485, L512, L599. DOI should be added.

Missing DOIs will be added to these references.

L482. What ISMEJ is should be specified.

We will specify this abbreviation, which is "Multidisciplinary Journal of Microbial Ecology".

L487, L498, L530, L590, L591, L593, L611. The references do not seem complete.

We thank the reviewer for this comment and will add missing information to these references.

L501. European Commission Edn? Soil Atlas Series?

We thank the reviewer for spotting this typo and will correct the reference as requested in the
corresponding document:

Jones, A., Breuning-Madsen, H., Brossard, M., Dampha, A., Deckers, J., Dewitte, O., Gallali, T.,
Hallett, S., Jones, R., Kilasara, M., Le Roux, P., Micheli, E., Montanarella, L., Spaargaren, O.,
Thiombiano, L., Van Ranst, E., Yemefack, M. , Zougmoré R., (Eds.): Soil Atlas of Africa, European
Commission, Publications Office of the European Union, Luxembourg. 176 pp, 2013.

L530. The publisher should be specified.

The publisher is Geoderma and we will add it accordingly, we thank the reviewer for spotting this
issue:

Mujinya, B. B., Mees, F., Boeckx, P., Bodé, S., Baert, G., Erens, H., Delefortrie, S., Verdoodt, A.,
Ngongo, M., and Van Ranst, E.: The origin of carbonates in termite mounds of the Lubumbashi area,
D.R. Congo, Geoderma, 165, 95-105, https://doi.org/10.1016/j.geoderma.2011.07.009, 2011.

613. This reference does not seem at the right place (Vagen et al. after Vollset et al.).

Following the Danish/Norwegian alphabet, "å" follows "z". Therefore "Vågen et. al." is at the correct
alphabetic position after "Vollset et al.".

L615. The end of the reference should be checked.

The reviewer is correct, the end of this reference includes some unnecessary information, which we
will remove.

L622. I.W.G.?

I.W.G stands for IUSS Working Group WRB. We will correct this in the reference and replace it by the
more recent version:

IUSS Working Group WRB: World Reference Base for Soil Resources 2014, update 2015
International soil classification system for naming soils and creating legends for soil maps, World Soil
Resources Reports No. 106, Food and Agriculture Organization of the United Nations, Rome, Italy,
pp. 193, ISBN978-92-5-108369-7, 2015.

---

## Author Response (AR1)

**Response with revised manuscript**

The Central African Soil Spectral Library: A new soil infrared repository and a geographical prediction analysis" by Summerauer et al. (SOIL-2020-99)

**Dear Editor, Dear Reviewers,**

Many thanks for your constructive and insightful comments on our manuscript draft entitled "The Central African Soil Spectral Library: A new soil infrared repository and a geographical prediction analysis" by Summerauer et al. (SOIL-2020-99). The comments improved the quality of the manuscript and helped us to present and discuss the obtained results in a more appropriate and clearer way.

Please find below a list of all relevant changes followed by the point-by-point responses to the two reviewers, with the original comments in black and our point-by-point responses directly below colored in blue. A track-changed version of the manuscript is also provided separately.

We hope we addressed all comments to your satisfaction.

On behalf of all co-authors,

Laura Summerauer

**1 List of relevant changes**

**Title**

As suggested by both reviewers, we changed the title, which describes more precisely the content of the manuscript:

*The Central African Soil Spectral Library: A new soil infrared repository and a geographical prediction analysis*.

**Abstract**

The modeling strategies, spiking of the models and the corresponding results were reformulated to present these sections more clearly.

**Methods**

- Sample preparation details (drying, sieving) and more information on the measuring spectrometer (wavenumber ranges) were added.
- Missing details within the pre-processing section were added and modified (number of latent variables, Q statistics).
- Renaming of abbreviated variables which had the same letters.
- Additional information to the modeling section was included (optimization of window size for dissimilarity approach/search of nearest neighbors, minimum/maximum of Partial Least Squares (PLS) factors used for the weighted average partial least squares regression).
- The methods to perform the principal component analysis with the pre-processed spectra was added.

**Results and Discussion**

- The units from total carbon and total nitrogen were converted from % into g kg$^{-1}$.
- The results and discussion sections were rewritten: Comparisons of prediction accuracies between regions were done using the root mean square error (RMSE) in relation to interquartile range (Ratio of Performance to InterQuartile distance; RPIQ). For analysing the differences between strategies for each region, the RMSE was used.
- The results of spiking (strategy 3) were presented and discussed more in detail.
- The effects (negative/positive/no) of strategy 2 (reduced geographical extrapolation) and strategy 3 (avoided extrapolation) vs. strategy 1 (extreme case of extrapolation) were presented and discussed more in detail.

**Graphs**

- The continental map (Figure A1) was moved to main text and Figure 1 was moved to the appendix.
- A new graph with the reference data was created presenting the distribution and interquartile ranges (boxplots) of the laboratory data.
- The flow chart was removed because it did not present the workflow clearly.
- The principal component analysis (PCA) plots with two dimensions (PC1-PC2, PC1-PC3, PC2-PC3) were created and the one with three dimensions was removed.

**Tables**

- Table 1, Table 2, Table 3: The central African regions, which were not used in this study, were removed from these tables in the main text. Since we still want to show the entire central African library in the manuscript as well, we moved the tables of all the regions to the appendix.

**References**

All the unnecessary URLs were removed and the DOIs were carefully tested. Moreover, the references were carefully checked
and re-evaluated.

**2 Responses to reviewer 1**

GENERAL COMMENT

The paper presents MIRS predictions of soil total C and N concentrations (TC, TN) in six regions of Central Africa separately,
using the AfSIS Sub-Saharan library with no Central African soils (Strategy 1), possibly completed with the samples from the
five other regions (Strategy 2), possibly completed with spiking samples from the same region (Strategy 3). This is done with
the Memory-based learning (MBL) regression procedure, which uses spectral calibration neighbors for building a PLS
regression for each target sample individually.

This is very interesting, but the paper suffers several drawbacks. Some methodological aspects are not presented (selection of
the number of latent variables in global calibrations developed for optimizing spectral pretreatment; window size for
calculating spectral similarity; possible cut-off value for spectral similarity; minimum and maximum number of latent variables
for calculating weighted average predictions) or not discussed (pretreatment selection on X residues instead of Y residues, as
usually; forcing spiking samples into neighborhoods; why not testing a strategy without AfSIS dataset, to evaluate its
usefulness), some terms are not introduced/defined (hold-out and validation sets; MEpred; notion of accurate prediction), and
some points are unclear (what were Central African samples out of the six core regions used for? why were AfSIS sentinel
sites divided into hold-out and validation sets?). Some results are misinterpreted (using RMSE for comparing predictions
between regions with different distributions of TC or TN; differences between strategies), others are not presented in the text
(effect of the number of spiking samples) or not discussed (negative effects of Strategy 2 in several cases), and conclusions
often seem too optimistic ("accurate predictions" etc. while error represented >=30% of observed mean in most cases).

For these reasons, I recommend moderate revision.

We thank the reviewer for the detailed comments and constructive criticism. We fully agree that there is some methodological
information missing, which we added accordingly or discuss in the corresponding comments below. We also acknowledge
that some terms need further explanation and the structure of the presentation needed to be clearer. We therefore only present
the six regions we actually worked with in our manuscript and will add a table for the entire spectral library in the appendix.
We also agree that there were some misinterpretations that arose from considering only RMSEpred. To analyse the predictions
between the regions, we will use the RPIQpred instead, and only when comparing strategies within the same region, we use
the RMSEpred. Additionally, we propose to replace Table 3 by a figure (see below), which visually depicts each distribution of total carbon and total nitrogen contents, including a boxplot, showing the interquartile ranges. Furthermore, we discussed the partially negative effect of strategy 2 compared to 1 in detail and also report and discussed the effect of spiking in detail. Overall, we are convinced that we were be able to implement these changes as discussed below and it improved the readability and quality of the manuscript.

SPECIFIC COMMENTS

The title is short, which may be an advantage, but I wonder if it is informative enough; moreover the genericity of the work is not highlighted (i.e. using a large spectral library for predictions in poorly documented areas).

The actual title is a general title meant for a broad audience interested in quantitative soil assessments, who may not necessarily be experts in spectroscopy. However, we agree that the title should be more specific and changed it accordingly:

**The Central African Soil Spectral Library: A new soil infrared repository and a geographical prediction analysis**

L8-9. What was done with the six core regions, and what the three levels of extrapolation consisted of, should probably be specified a little bit. Moreover, specifying the size of AfSIS SSL would be useful.

We agree with the reviewer that the strategies in the abstract lacked some details described in the abstract and implemented
the following changes starting from line 7 (see below):

*"For the analysis, we used six regions from the CSSL, which we predicted using an existing continental SSL (African Soil Information Service, AfSIS; n = 1902) that does not include central African soils. We explored three different strategies, at decreasing degree of geographic extrapolation, to predict total carbon (TC) and total nitrogen (TN) contents of the six selected*
*regions using models built with (1) the AfSIS SSL only, (2) AfSIS SSL combined with the five remaining regions, and (3) a combination of AfSIS SSL, the remaining five regions, and selected samples from the target region."*

L13-14. Improvement was not clear for TC, from RMSE=0.38-0.86% to 0.41-0.89%. more details? split up into regions? Moreover, I wonder if such prediction errors allow considering the approach as particularly useful (i.e. is information
ACCURATE ENOUGH?). Note that RMSE is not particularly informative as long as distribution has not been specified (e.g. RMSE=3 is small if mean=30 and SD=10, but high if mean=10 and SD=5), so adding RPIQ would be useful.

We agree with the reviewer that the presentation of these ranges does not show improvements. This is now clarified in the abstract; we changed the abstract suggest in the discussion that readers assess the cost-benefit of investing in new sampling versus gaining accuracy.

L38. Cost is one reason, there are probably others.

The reviewer is correct, there are numerous other reasons for missing soil data in central Africa, including but not limited to accessibility to sampling areas, infrastructure, and political instability. We included these other factors in the revision.

L52-53. The notion of "positive predictive transfer" is unclear for me.

We thank the reviewer for the comment, and we agree that this notion was not clearly formulated. With "positive predictive transfer" we describe the information transferred from a large infrared library for a new calibration of a local set as described by Padarian et al. (2019). The calibration of a new local set using a large-scale spectral library can be complex in soil science, especially when the local set covers a different geographical domain than the library. Soil spectral libraries become particularly useful when a large amount of their relevant information can be extracted in a way that it improves prediction accuracy (positive transfer) and minimizes the number of additional costly local reference measurements for quantifying soil properties in the local set (accuracy-cost trade-off). To avoid technical jargon, we rephrased the paragraph L48-L55 and move it to L60, where it fits better into the context:

*"One of the main aims of establishing large-scale SSLs is to minimize the need for future wet chemical analyses (e.g., Nocita et al., 2014; Stevens et al., 2013; Shi et al., 2014; Viscarra Rossel et al., 2016). However, these libraries often span vast geographical areas that include different soil types and climate zones, which comprise complex soil organic C forms and mineral compositions. Due to this heterogeneity, predictions rendered by global linear regression models are often unfeasible for new local soil property assessments at a regional, field or plot-scale, especially when the new set covers another*

*geographical domain than the library. Pandiran et al,. (2019) could considerably improve prediction accuracies for a new local set when using a compositionally related subset from a large-scale SSL together with a small number of local reference analyses. The cost-accuracy trade-off can be met when the accuracy of the library-based prediction is similar to the one made when applying a local but more costly calibration strategy. Several data-driven methods have proven to be successful to overcome this issue, for example RS-LOCAL (Lobsey et al., 2017) and memory-based learning (a.k.a local learning e.g.*

*Ramirez-Lopez et al., 2013; Shenk et al., 1997; Naes 1990). In addition, other promising approaches have also been proposed, although they require more research (e.g. deep learning (Ng et al. 2019), fuzzy rule-based systems (Tsakiridis et al. 2019))."*

Padarian, J., Minasny, B., McBratney, A.B.: Transfer learning to localise a continental soil vis-NIR calibration model, Geoderma, 340, 279-288, https://doi.org/10.1016/j.geoderma.2019.01.009, 2019.

L64-67. LOCAL and Locally weighted PLSR should probably be cited, as they also aim at selecting spectral calibration neighbors, and were used earlier in soil spectroscopy.

We agree with the reviewer and propose the following changes, together with the next comment (L64-70) (see below).

L64-70. In my opinion, approach complexity should be considered: some approaches are rather simple (e.g. spiking) thus widely usable, while others are complex thus usable only by experts (e.g. the fuzzy rule-based system proposed by Tsakiridis et al. 2019).

We rephrased the paragraph as following and added two references to the reference list:

*"Several data-driven methods have proven to be successful in overcoming this issue, for example RS-LOCAL (Lobsey et al., 2017) and memory-based learning (a.k.a local learning e.g. Ramirez-Lopez et al., 2013; Shenk et al., 1997; Naes 1990). In addition, other promising approaches have also been proposed, although they require more research (e.g. deep learning (Ng et al. 2019), fuzzy rule-based systems (Tsakiridis et al. 2019))."*

Naes, T., Isaksson, T., & Kowalski, B.: Locally weighted regression and scatter correction for near-infrared reflectance data. Analytical Chemistry, 62, 664–673, https://doi.org/10.1021/ac00206a003, 1990.

Tsakiridis, N., Theocharis, J., Panagos, P., & Zalidis, G.: An evolutionary fuzzy rule-based system applied to the prediction of soil organic carbon from soil spectral libraries. Applied Soft Computing, 81, 1-18, https://doi.org/10.1016/j.asoc.2019.105504,
2019.

Ng, W., Minasny, B., Montazerolghaem, M., Padarian, J., Ferguson, R., Bailey, S., McBratney, A.B.: Convolutional neural network for simultaneous prediction of several soil properties using visible/near-infrared, mid-infrared, and their combined spectra, Geoderma, 352, 251-267, https://doi.org/10.1016/j.geoderma.2019.06.016, 2019.

L86. "covers a large geographic area" is questionable as the sample population is clustered, and a wide area is not represented (i.e. between Kinshasa, Tshopo and Katanga).

The reviewer is correct! The sampling locations did not cover the entire area and the term is potentially misleading. We addressed this comment in line 86 accordingly:

*"The sample locations are clustered in eight regions distributed over a large geographical area of central Africa, from a latitude of ..."*

L99. The way samples were dried should be specified, moreover they had probably been 2-mm sieved previously.
We thank the reviewer for requesting this information. The samples were all sieved through a 2 mm mesh and either air dried or oven-dried at temperatures of 50 °C or 60 °C, all of them suitable for total carbon and nitrogen analyses. After sieving and drying, soil samples were ground to a powder (< 50 µm) using a ball mill. We included these details in the revised manuscript

Tab.2. I've not understood how samples from Equateur, Bas-Uélé, North Kivu and Kongo-Central were used (they are not
mentioned in Strategy 2, L204-205).

The regions Equateur, Bas-Uélé, North Kivu and Kongo-Central were excluded for the further analyses because they did not have enough samples to allow for reliable analysis (< 80 samples per region). With this table, we intended to present the entire infrared library we created. However, we fully understand that this is confusing here and we removed these regions from this table but presented the full library (including these four regions: Équateur, Bas-Uélé, North Kivu and Kongo-Central) in a supplementary table in the appendix.

L106-107. Does this suggest charcoals were considered organic, or negligible?

This is a legitimate question, since slash-and-burn is commonly used to clear fields in central Africa which adds charcoal to the topsoils. For our soil analyses, visible pieces of charcoal were removed, which could clearly influence TC measurements in certain samples. This detail was added in the methods.

L112. SPECIFYING PARTICLE SIZE WOULD BE USEFUL (< 0.2 mm? < 0.1 mm?).

All samples were ground to a powder (<50 µm) using a ball mill, which is sufficiently accurate for soil spectral diagnostics. Diess et al. (2020) report sufficiently accurate model estimates when grinding below 0.5mm, and Guillou et al. (2015) even report no significant differences at particle size thresholds of 1.0mm, 0.5mm and 0.25mm thresholds. We added this information to the method section.

Deiss, L., Culman, S. W., & Demyan, M. S.: Grinding and spectra replication often improves mid-DRIFTS predictions of soil properties, Soil Science Society of America Journal, 84, 914–929. https://doi.org/10.1002/saj2.20021, 2020.

Guillou, F. L., Wetterlind, W., Viscarra Rossel, R. A., Hicks, W., Grundy, M., & Tuomi, S.: How does grinding affect the mid-infrared spectra of soil and their multivariate calibrations to texture and organic carbon? Soil Research, 53, 913-921, https://doi.org/10.1071/SR15019, 2015.

L113, L125. Spectral range and resolution should probably be specified.

We thank the reviewer for bringing this to our attention. We fully agree and changed the sentences accordingly:

*"All samples were measured with a VERTEX70 Fourier Transform-IR (FT-IR) spectrometer with a High Throughput Screening Extension (HTS-XT) (Bruker Optics GmbH, Germany) in order to measure their MIR reflectance spectra. Spectra were acquired in a resolution of 2 cm-1 within a range of 7500 cm-1 to 600 cm-1, which corresponds to a wavelength range of 1333 nm to 16667 nm. A gold coated reflectance standard (Infragold NIR-MIR Reflectance Coating, Labsphere) was used as a background material for all measured soils in order to normalize the sample spectra. Reflectance was transformed into absorbance using log(1/reflectance) prior to further processing and subsequent modeling. "*

L125. Spectra were collected on AfSIS and CSSL samples with different spectrometers, so the question of compatibility should be addressed (e.g. was there standardization?).

The reviewer raises an important point regarding the compatibility of data form two different spectral libraries. Luckily, the two instruments were both FT-IR spectrometers from BRUKER which use the same settings and the same internal standards. The scanning methods of the CSSL were adapted to the ICRAF standard operating procedures. For these reasons, no instrument standardization was necessary and all spectra between the libraries can be compared one to one. This information was added to the methods section of the revised manuscript.

L132. A reference dealing specifically with soils would probably be more appropriate.

We agree that a more soil specific reference would help to point out the importance of the effect of pre-processing and we therefore suggest the two following publications:

Seybold, C.A., Ferguson, R., Wysocki, D., Bailey, S., Anderson, J., Nester, B., Schoeneberger, P., Wills, S., Libohova, Z., Hoover, D. and Thomas, P.: Application of Mid-Infrared Spectroscopy in Soil Survey. Soil Science Society of America Journal, 83, 1746-1759, https://doi.org/10.2136/sssaj2019.06.0205, 2019.

Sila, A. M., Shepherd, K. D., and Pokhariyal, G. P.: Evaluating the utility of mid-infrared spectral subspaces for predicting soil properties, Chemometrics and Intelligent Laboratory Systems, 153, 92–105, https://doi.org/10.1016/j.chemolab.2016.02.013, 2016.

L140. p is not defined. Actually P is a d x l matrix, not a d x p matrix.

We thank the reviewer for spotting this typo! "d $\times$ l matrix" is correct and we changed it as suggested by the reviewer.

L145-161. The error E depends on the NUMBER OF LATENT VARIABLES (l). HOW WAS THIS PARAMETER DEFINED? Moreover, the EXPECTED BENEFIT OF THIS APPROACH (i.e. computing Xcssl residues) for optimizing spectral pretreatment SHOULD BE PRESENTED, when compared with examining RMSE associated with every pretreatment (i.e. computing Ycssl residues, as commonly done).

We fully acknowledge that this was not clearly explained in the text and addressd these issues. We explain that the analysis of spectral reconstruction error is indeed commonly used in spectroscopy for outlier identification. This error is also known as the Q-statistic and it indicates how well a given new sample conforms to the PLS model. Since the response values in the prediction set are unknown, we can use the Q-statistic as a proxy for the response errors. In the revised version, we will explain that we assume that if a given set of pre-processing steps lead to large Q-values, then it is expected that it will also lead to large errors in the prediction of the response values. We will also add references to support this assumption. In the new version of the manuscript, we will mention that for this analysis we fixed the number of PLS factors to 20, as projected variables beyond this dimension did not capture a considerable amount of the original spectral variance. For example, PLS variable 21 amounted for less than 0.01% of the original variance in all the cases.

L165."spectral matrices which can be properly represented by a PLS model" is unclear. Moreover, the assumption that SIMILAR PRETREATMENTS OPTIMIZED GLOBAL AND LOCAL CALIBRATION SHOULD BE DISCUSSED (e.g. according to literature).

The ideas behind this sentence was clarified with the description of the Q-statistic (see previous reply L145-161) and the advantages of its use for pre-processing optimization. Now we indicate that according to Wise and Roginsky (2015), large Qc values are proxies to large prediction errors and therefore Q-statistic can be used to judge the suitability of a set of pre-processing steps.

Wise, B. M., & Roginski, R. T.: A calibration model maintenance roadmap. IFAC-PapersOnLine, 48, 260-265, https://doi.org/10.1016/j.ifacol.2015.08.191, 2015.

L170. The problem with multiplicative scatter correction is that the transformed spectrum depends on the spectrum population it belongs to, so changes when this population changes.

We thank the reviewer for raising this concern but do not see this as a problem. Multiplicative scatter correction (MSC) aligns or rotates a given spectrum towards a reference one which is fixed. This reference spectrum can be seen as a parameter of the MSC transformation. By doing this, multiplicative and additive shifts between spectra are removed. Although, in many applications the average spectrum of the calibration set is used as the reference one, in theory any spectrum can be used (See Rinnan et al., 2009). Therefore, MSC is not necessarily affected by changes in the spectral population. The reference spectrum parameter of a defined MSC step should not be modified as long as it guarantees successful removal of the multiplicative and additive scattering effects across the spectra.

Rinnan, Å., Van Den Berg, F., & Engelsen, S. B.: Review of the most common pre-processing techniques for near-infrared spectra, TrAC Trends in Analytical Chemistry, 28, 1201-1222. https://doi.org/10.1016/j.trac.2009.07.007, 2009.

L195. Why 20 spiking samples per regional set, not 10 or 30?

We agree with the reviewer, that the selection of the number of spiking samples has not been adequately described in the manuscript. Generally, the number of spiking samples should be minimized to reduce costs for laboratory reference analyses. We set the maximum number of spiking samples to 20, which can already mean quite a high financial investment but we feel that it is worth these costs given the reduction of geographical extrapolation and the effect of using spatially close samples on the predictive performance. We tested one to 20 spiking samples and compared the prediction accuracy, which was on average best with 20 spiking samples (Figure 5). We added more details about the spiking effect in the results and discussion sections of the manuscript.

L197. The way k-means works could (should?) be briefly presented.

Since this method is widely used and well documented in pedometrics and chemometrics for sampling calibration datasets, we considered that it was sufficient to refer the reader to other studies, where k-means sampling is explained. However, we agree that it is useful when we explain it in a sentence and changed it as following:

*"... for each complete regional set, 20 samples were selected using the k-means sampling algorithm. This sampling strategy is implemented in the R package prospectr (Stevens and Ramirez-Lopez, 2020) and selects one sample per cluster calculated with a k-means algorithm on a principal component analysis of the pre-processed spectra (Næs, 1987).*

L199. The strategies considered are: AfSIS alone; AfSIS +other Gi; AfSIS +other Gi +Ki. Other strategies would have been
interesting: only using other Gi, or other Gi + Ki, to EVALUATE THE USEFULNESS OF AfSIS (which would be very interesting); AfSIS +Ki, to evaluate the usefulness of other Gi; Ki only, to evaluate the usefulness of AfSIS and other Gi. But this would require much additional work!

Our aim was to propose strategies that could leverage the use of the AfSIS spectral library to accurately predict soil properties in regions which are poorly covered by it. Therefore, we only evaluated modeling approaches that involved the use of this
library. There is clear evidence that very accurate soil predictions can be achieved by using models built only with samples originating from the same region or area where these predictions are required. This is because large non-linear complexity is avoided in local-scale models (See e.g., Tziolas et al., 2019). Despite this, we consider that this implies that every undersampled region will require a representative calibration sample set which might be expensive or impractical. In this respect, the evaluation of models using only other Gi, or only other Gi + Ki was not considered as they do not really solve the problem of
using a large spectral library in poorly sampled areas.

Tziolas, N., Tsakiridis, N., Ben-Dor, E., Theocharis, J., & Zalidis, G.: A memory-based learning approach utilizing combined spectral sources and geographical proximity for improved VIS-NIR-SWIR soil properties estimation,Geoderma, 340, 11-24, https://doi.org/10.1016/j.geoderma.2018.12.044, 2019.

L217-219. HOW WAS w DEFINED? Moreover, WHAT p STANDS FOR IS NOT CLEAR: it has not been defined, but according to L140, was apparently used in place of l (number of latent variables); but I'm not sure this makes sense here. Furthermore, I'm not sure to understand what k=1 means. I also note that d has already been used (number of wavelengths; L139). So CLARIFICATION IS REQUIRED. We might also wonder why evaluate dissimilarity (1-S) and not similarity (S),
when the objective is to select calibration samples similar to the target sample (cf. L311). Furthermore, I WONDER IF A

SIMILARITY/DISSIMILARITY CUT-OFF VALUE WAS DEFINED, below/above which spectra were not considered neighbors (i.e. no prediction for target samples with too few neighbors); and if yes, how this cut-off value was defined.

We are thankful that the reviewer noticed the use of letters for multiple variables, which is misleading. Again, the reviewer is correct, spotting the mistake in L140, which leads to confusion in L215-219. Correcting this as suggested above, this issue should be resolved here.

The window size (w) was optimized based on a spectral nearest-neighbor search within the AfSIS library. For every sample in the AfSIS library, its closest sample (in the spectral space) was identified. The samples were compared against their closest samples in terms of TC and TN and the root mean squared differences (RMSD) were computed according to the following equations:

$$j(i) = NN(Xc_i, Xc^{-i})$$

and

$$RMSD = \sqrt{\frac{1}{2m} \sum_{i=1}^{m} \sum_{h=1}^{2} (yc_{i,h} - yc_{j(i),h})^2}$$

where $Xc$ is the spectra of the AfSIS library, NN($⟦Xc⟧\_i$, $⟦Xc⟧^{(-i)}$) represents a function to obtain the index of the nearest neighbor observation of the ith sample found in Xc(excluding the ith sample), $⟦yc⟧\_{(i,h)}$ is the value of the i-th observation for the h-th property variable (either TC or TN). A total of 10 window sizes were evaluated (from 31 up to 121 in steps of 10). According to the RMSDs obtained, the optimal w was 71.

Concerning the comment about using the concept of similarity or dissimilarity, we believe that is not actually relevant. It is clear that similarity or dissimilarity measures can be both used to identify similar samples. Many examples of the use of correlation dissimilarity for nearest neighbor identification can be found in the NIR spectroscopy literature (See for example Wadoux et al., 2021; Khosravi et al., 2020; Gholizadeh et al., 2018; Zhu et al., 2011).

The selection of the number of neighbors was optimized by testing a range of numbers of neighbors instead of using a dissimilarity threshold value (see Ramirez-Lopez (2020) for more information).

Gholizadeh, A., Saberioon, M., Carmon, N., Boruvka, L., & Ben-Dor, E.: Examining the performance of PARACUDA-II data-mining engine versus selected techniques to model soil carbon from reflectance spectra. Remote Sensing, 10, 1172, 350 https://doi.org/10.3390/rs10081172, 2018.

Khosravi, V., Ardejani, F. D., Aryafar, A., Yousefi, S., & Karami, S.: Prediction of copper content in waste dump of Sarcheshmeh copper mine using visible and near-infrared reflectance spectroscopy. Environmental Earth Sciences, 79, 1-13, https://doi.org/10.1007/s12665-020-8901-0, 2020.

Ramirez-Lopez, L.: resemble: Regression and Similarity Evaluation for Memory-Based Learning in Spectral Chemometrics, https://CRAN.R-project.org/package=resemble, r package version 2.1.1, 2020.

Wadoux, A., Malone, B., Minasny, B., Fajardo, M., McBratney, A.B. (Eds.): Soil Spectral Inference with R: Analysing Digital

Soil Spectra using the R Programming Environment, Springer Nature, Cham, Switzerland, 2021.

Zhu, Z., Corona, F., Lendasse, A., Baratti, R., & Romagnoli, J. A.: Local linear regression for soft-sensor design with application to an industrial deethanizer, IFAC Proceedings Volumes, 44, 2839-2844, https://doi.org/10.3182/20110828-6-IT-1002.02357, 2011.

L220-225. According to Shenk et al. (1997), the weighted average is calculated over a range of latent variables, i.e. from a MINIMUM TO A MAXIMUM NUMBER OF LATENT VARIABLES CONSIDERED, AND THESE PARAMETERS HAVE TO BE SPECIFIED. Moreover, both s1:j and gj are calculated for the jth latent variable, so writing "s1:j" instead of "sj" is unclerar. Furthermore, Shenk et al. (1997) did not call this approach "Weighted averaged PLS"; but why not…

As correctly pointed out by the reviewer, details about the WA-PLS are missing in the current version of the manuscript. We added the missing information to the text. The weighted average was calculated using a range of latent variables from 5 to 30 in increments of 1, which we added to the manuscript accordingly.

To compute the weights, we use the exact same method as described by Shenk et al. (1997, see page 227 of their paper). In the equation used to compute the weights, s1:j represent the root mean square of the spectral residuals of the query spectrum. The reconstruction is done by multiplying the scores of the projected query spectrum by the (transposed) loading matrix of the PLS model built from its neighbor samples. In this multiplication the first j rows of the scores and loading matrices are used. Using sj instead of s1:j, would wrongly indicate that only the jth row of the scores is multiplied by the jth transposed row of the loadings. Furthermore, in the equation we also use the term gj to refer to the root mean square of the regression coefficients corresponding to the jth PLS component. In this case we do not use the subscript 1:j as we are using only the jth row of the matrix of regression coefficients (instead of the first j rows). We will extend the explanation of this notation for a new version of the manuscript.

Indeed Shenk et al., (1997) do not explicitly call this regression method "weighted averaged PLS". Although, what this method does is to compute a "weighted average of the individual model predicted values with from the minimum to the maximum number of factors" as explained by Shenk and Westerhaus (1998) in the following patent filing: https://patents.google.com/patent/US5798526A/en. Therefore, we do not see the term "weighted averaged PLS" as incorrect in our manuscript.

L230-232. Hold-out and validation sets have not been introduced, so this part is not very clear (e.g. why dividing regional AfSIS sub-libraries into hold-out and validation sets? L256 and Tab.3 these sub-libraries were not separated).
We thank the reviewer for spotting this point of confusion. We clarified this issue as following:

*"The grouping factor was used for the optimization of the nearest neighbor search, i.e. the nearest neighbor cross-validation*
*(see L226) to avoid overfitting: keeping the nearest neighbor out, the model was trained with the remaining neighbors which were not from the same region as the hold-out neighbor (region corresponds to the sentinel sites within the AfSIS SSL). "*

This was changed accordingly to avoid a misunderstanding as shown in this comment.

L233. I understand the minimum requested number of neighbors was 150, and the maximum possible number of neighbors was 500. WHAT IF A TARGET SAMPLE HAD LESS THAN 150 NEIGHBORS?
This is an important question of the reviewer. Of course, a sample could have less than 150 neighbors in the used spectral library. We tested the minimum number of available neighbors prior training the final model. We agree that the minimum number of neighbors should have been adjusted downwards if there would not have been enough neighbors, which was luckily
not the case. We explained this more in detail in the revised manuscript.

L236. FORCING SPIKING SAMPLES INTO THE NEIGHBORHOOD of every target sample is questionable, and the discussion should address this point.
Unfortunately, the reviewer does not provide an explanation on why forcing spinking samples into the neighborhood is
questionable.

Spectral Neighbor identification is a mathematical attempt to select soil observations that share similar compositional characteristics with the observation that requires a prediction. MIR spectra partially reflect the compositional characteristics of the samples. We assume that soils originating from the same geographical region might be governed by very similar soil
formation processes. This is a concept of spatial autocorrelation which is widely used (Fortin et al. 2016). Furthermore, it is widely accepted that the best spectral models (most accurate) that can be built for a given area are those that are calibrated with samples from the same area (see also comment L199). For these reasons, we assume that forcing samples of a given area to belong to the neighborhoods of samples from the same area guarantees that samples originating from similar soil formation processes are included in the models. Therefore, our approach is not arbitrary as it is expected that these samples improve prediction accuracy.

Fortin, M.-J., Dale, M.R. and Ver Hoef, J.M.: Spatial Analysis in Ecology. In Wiley StatsRef: Statistics Reference Online (eds N. Balakrishnan, T. Colton, B. Everitt, W. Piegorsch, F. Ruggeri and J.L. Teugels). https://doi.org/10.1002/9781118445112.stat07766.pub2, 2016.

Fig.3. Beside orange and green circles, many grey circles were also outside AfSIS black circles, and it would be useful to mention where they originated from.

The transparency of the black AfSIS symbols is misleading. They seem gray, while the remaining samples are black, where the density is high. We change the style of the symbols, so that it becomes clear which points belong to the AfSIS library.

L265-267. CRITERIA FOR "GOOD PREDICTIVE RESULTS" HAVE NOT BEEN SPECIFIED. Actually many results were not so good, especially for TN, especially with Strategy 1 (e.g. RMSE for TC and TN was >=50% of observed mean for 2-3 regions with Strategy 1, and >=30% of the mean for 4-5 regions with Strategy 2). And ACCORDING TO RPIQ, PREDICTIONS FOR SOUTH KIVU AND IBURENGERAZUBA WERE OFTEN AMONG THE BEST ONES, so the reasons for considering they "showed the lowest accuracy levels" should be revised, or at least explained.

The reviewer is correct, we have not introduced criteria to define a good or accurate prediction, which we added. However, the required prediction accuracy depends on the field of application.

As the reviewer points out, RMSEpred is useful to estimate prediction accuracy within the same region but not to make comparisons between regions since ranges for TC and TN differ between the regions, especially for Iburengerazuba and South Kivu with forest soils with high TC and TN contents. We agree that it does not make sense to classify the statistical performance of the South Kivu and Iburengerazuba regions as poor. Indeed, when looking at the RPIQpred values, they performed well. This is due to the large interquartile (IQ) range of these regions compared to Tsuapa and Tshopo, which exhibited considerably smaller IQ ranges. We thank the reviewer for this careful attention to these statistical descriptions and modified the results and discussion accordingly. Moreover, we replaced Table 3 with the proposed plot (see below), which clearly shows the distribution of TC and TN in each region including their IQ ranges in the boxplots. They gray coloured line and text indicate the spiking sets  (Ki), the black coloured lines and text represent the six regional sets (Gi) after removal of each Ki and the AfSIS SSL data set A. These details to the figure were added to the caption.

[Figure]

Total Carbon / Total Nitrogen raincloud distribution plots by region

L271-272. RMSEpred is useful for comparing strategies for a given region, but CANNOT BE THE FIRST PARAMETER
CONSIDERED FOR COMPARING PREDICTION ACCURACY BETWEEN REGIONS WHERE DISTRIBUTIONS OF
TC OR TN WERE DIFFERENT. R² describes proportionality, not similarity; so, though understood by a wide audience,
should be used with care. Comparison between regions should firstly be based on RPIQ, which showed good results for
Kabarole, Iburengerazuba and (for TC) South Kivu and poor results for the other regions, especially Tshopo for TC and Haut-
Katanga for TN.

Yes we agree and as we detailed in the response above, we modified the results and discussion such that we only use RMSEpred
to compare the same regions across strategies and RPIQpred and R2pred to compare regions within a given strategy. We
suggest the following changes:

*"The best prediction accuracies for TC were achieved for the regions South Kivu, Iburengerazuba and Kabarole, where*
*RPIQpred values were between 2.43–3.95, while Tshopo, Tshuapa and Haut-Katanga performed less good with RPIQpred*
*<= 1.84. For TN, Iburengerazuba and Kabarole performed well with RPIQpred 2.14 and 2.86, respectively. However, the*
*four other regions Haut-Katanga, South Kivu, Tshopo and Tshuapa exposed smaller RPIQpred <= 1.37. "*

L277-279. The fact that CENTRAL AFRICAN SAMPLES WERE POORLY REPRESENTED BY AfSIS SHOULD ALSO BE MENTIONED AS POSSIBLE REASON.

We agree with the reviewer that this should be highlighted at this point and we added this accordingly. We also suggest to put Figure A1 (continental map) in the main text and move Figure 1 to the appendix.

L282-283. Again, RMSEpred should not be used for comparisons between regions.

We agree with the reviewer and changed it as described more in detail in L271–272. We suggest the following changes:

*"The predictive performance in strategy 2 exhibited errors (RMSEpred) ranging between 0.41–0.89 % and 0.03–0.12 % for TC and TN, respectively (Table 4). The most accurate predictions for TC were as in strategy 1 obtained for the regions Iburengerazuba, Kabarole and South Kivu (RPIQpred > 2.36), but RPIQpred value of Haut-Katanga was remarkably higher than in strategy 1 (2.30 vs 1.62). Predictive performance for TC of Tshopo and Tshuapa were still below an RPIQpred of 2."*

*"For TN, similarly to strategy 1, prediction accuracy was good for Iburengerazuba and Kabarole. For the regions Haut-Katanga, South Kivu, Tshopo and Tshuapa the RPIQpred values were higher than in strategy 1, but they were still below 2. "*

L284-286. RMSEpred for TC increased in three regions from Strategy 1 to 2, strongly sometimes, which is counter-intuitive so should be underlined, and POSSIBLE REASONS SHOULD BE PROPOSED (as was done for better TN predictions with Strategy 2 than 1).

The reviewer is correct in that RMSEpred increased for 3 regions, however it only increased by 0.03% in two of the cases. So, in total, from Strategy 1 to 2 the RMSEpred decreased substantially in 3 regions, barely changed in 2, and increased in one, which in our opinion signals an overall improvement in performance. At the moment, it appears that the inclusion of the additional CSSL regions reduced the accuracy of the Kabarole region, but it is unclear why the model did not fall back on the same prediction subset as Strategy 1. This was investigated and corrected in the revised manuscript.

L287. Better TN predictions with strategy 2 than 1 "was due", not "might be due".

Thank you, this was modified accordingly.

L290. RPIQ for TC "tended to be the same" except for Kabarole; but actually RPIQ decreased in South Kivu and Tshuapa, not much, but this is counter-intuitive.

As detailed above, we modified the discussion of these results in the text to explain the observed patterns.

L292. South Kivu was not an exception, as TN prediction was also improved.

Thank you for this correction. We modified the text accordingly.

Fig.5. THESE RESULTS SHOULD BE PRESENTED in the text, and an optimal number of spiking samples could be proposed for each region.

We thank the reviewer for requesting that these results be included in the text and will added them to the revised version.

L309, L317, L391. "Accurately predicted/model" "highly accurate predictions" are OVEROPTIMISTIC, e.g. when RPIQ <2 or RMSE > mean/2.

We thank the reviewer for this suggestion and toned down the language to "reasonably accurate".

L317-318. The point is that for TC, Strategy 2 reduced RMSEpred in only 3 out of the 6 regions considered; so "improved prediction accuracy" is questionable. And POOREST PREDICTION WITH STRATEGY 2 than 1 FOR 3 REGIONS SHOULD BE DISCUSSED.

We again thank the reviewer for pointing out this idiosyncrasy and as detailed in the responses above modified the results and discussion to detail these prediction results.

L322-325. There is STRONG MISINTERPRETATION, as in these two regions, TC (and TN in Iburengerazuba) was accurately predicted (RPIQ >2.3).

As detailed above, these discussion points surrounding the prediction results was modified.

L338. These results have not fully presented in the results section.

We thank the reviewer for pointing this out and detailed the spiking results in the results section.

L339. Three regions are cited, not two. Moreover, Strategy 3 yielded highest RPIQ whatever the region for both TC and TN; and the improvement was strong sometimes, with 10 spiking samples only (Kabarole and Iburengerazuba).

We thank the reviewer for pointing out this mistake. The text should read "three regions" and we removed the word "somewhat" to reflect the strong improvement. We further modified this section to say that Strategy 3 had a positive effect on all regions but an even stronger effect on the three regions we originally listed.

L343-344. For TN in South Kivu, RPIQ increased from 1.1 to 1.6 from Strategy 1 to Strategy 2, so prediction was noticeably improved.

We thank the reviewer for clarifying this point and modified the text to say how the prediction noticeably improved.

L345. "RMSE remained relatively high", but TC and TN were much higher than elsewhere! Considering RMSE without considering TC and TN distributions leads to misinterpretation.

Indeed, the reviewer is correct. We contextualized the RMSEpred with the higher TC and TN and instead focus on the RPIQpred as a more reliable indicator given the different distributions. We also used the new graph to show this distribution of TC and TN for each region more precisely (see above).

L345. "slightly" does not seem appropriate: e.g. for Iburengerazuba RPIQ increased from 2.8 to 3.6 for TC and from 3.2 to 4.5 for TN.

We agree with the reviewer that "slightly" is not the correct word and changed it to "substantially".

L349. As said above, the effect of spiking was strong sometimes (Iburengerazuba and Karabole).

We thank the reviewer for pointing this out and modified the text accordingly.

TECHNICAL CORRECTIONS

L6. 1800 soils or 1800 soil samples?

Soil samples. We clarified this in the text.

L7. "wider" is not clear for me in "Congo Basin and wider African Great Lakes region".

We removed the word "wider" from this sentence. Moreover, we corrected "African Great Lakes region" to Albertine Rift, which is a more precise name for the region.

L10. % is not a SI unit and may cause confusion for comparisons or changes (e.g. TC increased by 5%), so G KG-1 WOULD BE MUCH PREFERABLE.

We thank the reviewer for this comment. We converted the % unit into the SI unit g kg-1 as suggested by the reviewer.

L59. Sol vs. soil.

Thank you for pointing out this typo.

L77. Predicting a region is confusing.

The reviewer is correct, this sentence does not make sense. We specified accordingly in the updated version of the manuscript:

"... (2) To establish a workflow to accurately predict soils from variable locations within six selected geographical regions of the CSSL ...

L84. The sentence should be checked (e.g. layers vs. layer).

Thanks for spotting this typo, we corrected the word to the plural form.

Tab.1. Université catholique de Louvain and IITA/ICRAF are not references. Moreover, for the last reference, 2021a,b would be more appropriate than 2021b,a (this is detail).

We thank the reviewer for pointing out this detail. Indeed Université catholique de Louvain, IITA and ICRAF are not references. We renamed the column to „Data Source and Contributor".

L103. Total Al, Fe, Ca etc., or some particular fractions?

Total contents of cations have been analysed using aqua regia extractions. We agree with the reviewer that this should be specified and added information about the methods for analysing pH, texture and cations.

L115. In general absorbance = log(1/reflectance), not 1/reflectance.

The reviewer is of course correct about this and it was corrected accordingly.

L118. I note the manufacture place is mentioned here, which should probably be systematic.

This is correct, we thank the reviewer for seeing this detail. We removed the place to be consistent through the entire manuscript.

L134. Actually PLS has most often been defined as Partial least squares.

The reviewer is correct, PLS is an abbreviation, used for Partial Least Squares. We also defined the term accordingly in the manuscript (L56). With the sentence the reviewer brings up, we do not want to give PLS another meaning. We rather want to explain that the Partial Least Squares method can also be described as a projection of latent structures, which has by accident the equal letters and the same order.

L207. The sentence should be checked.

We agree with the reviewer and made the sentence clearer.

*"– Strategy 3: This time, strategy 2 was repeated, but in this case, extrapolation was avoided by using the spiking samples from the same geographical region as the region to be predicted;"*

L234, L243. Equation 8? Equations have not been numbered.

We thank the reviewer for spotting this mistake. We added numbers to all the equations in the revised manuscript.

Fig.3 is not very readable; projections on PC1-PC2 and PC1-PC3 would probably be more suitable.

We agree with the reviewer, that the 3D plot is not appropriate. We therefore plotted the three different components as suggested by the reviewer.

[Figure]

Tab.4. What MEpred stands for should be specified.

This is correct, we did not specify ME and added this information to the methods section.

L275. Tshopo, not Tschopp. Four regions are cited, not three.

We thank the reviewer for clarifying this, and changed it accordingly.

L426-427, L432-433, L436, L439, L445, etc. Are two DOIs or two URLs necessary? I note that non-DOI URLs do not always work ("error 404", "page not found", etc.).

We thank the reviewer for checking the DOIs and URLs in the reference list. We checked them carefully.

L442, L445, L508, L540, L564, L567, L570, L573, L584-585, L615, L617-618. Same (or almost same) DOI mentioned twice.

We also checked these DOIs and removed the ones, which were not necessary. We thank the reviewer for spotting this issue.

L448, L469, L473, L485, L512, L599. DOI should be added.

Missing DOIs was added to these references.

L482. What ISMEJ is should be specified.

We specified the full journal name in this reference

Gallarotti, N., Barthel, M., Verhoeven, E., Pereira, E. I. P., Bauters, M., Baumgartner, S., Drake, T. W., Boeckx, P., Mohn, J., Longepierre, M., Mugula, J. K., Makelele, I. A., Ntaboba, L. C., and Six, J.: In-depth analysis of N2O fluxes in tropical forest soils of the Congo Basin combining isotope and functional gene analysis, International Society for Microbial Ecology Journal (ISME J), https://doi.org/10.1038/s41396-021-01004-x, 2021.

L487, L498, L530, L590, L591, L593, L611. The references do not seem complete.

We thank the reviewer for this comment and added missing information to these references.

L501. European Commission Edn? Soil Atlas Series?

We thank the reviewer for spotting this typo and corrected the reference as requested in the corresponding document:

Jones, A., Breuning-Madsen, H., Brossard, M., Dampha, A., Deckers, J., Dewitte, O., Hallett, S., Jones, R., Kilasara, M., Le Roux, P., Micheli, E., Montanarella, L., Spaargaren, O., Tahar, G., Thiombiano, L., Van Ranst, E., Yemefack, M., and Zougmore, R.: Soil Atlas of Africa, European Commission, Publication Office of the European Union, Luxembourg, https://doi.org/10.2788/52319, 2013.

L530. The publisher should be specified.

The publisher is Geoderma and we added it accordingly, we thank the reviewer for spotting this issue:

Mujinya, B. B., Mees, F., Boeckx, P., Bodé, S., Baert, G., Erens, H., Delefortrie, S., Verdoodt, A., Ngongo, M., and Van Ranst, E.: The origin of carbonates in termite mounds of the Lubumbashi area, D.R. Congo, Geoderma, 165, 95-105, https://doi.org/10.1016/j.geoderma.2011.07.009, 2011.

613. This reference does not seem at the right place (Vagen et al. after Vollset et al.).

Following the Danish/Norwegian alphabet, "å" follows "z". Therefore "Vågen et. al." is at the correct alphabetic position after "Vollset et al.".

L615. The end of the reference should be checked.

The reviewer is correct, the end of this reference includes some unnecessary information, which we removed.

L622. I.W.G.?

I.W.G stands for IUSS Working Group WRB. We corrected this in the reference and replaced it by the more recent version:

IUSS Working Group WRB: World Reference Base for Soil Resources 2014, update 2015 International soil classification
system for naming soils and creating legends for soil maps, World Soil Resources Reports No. 106, Food and Agriculture
Organization of the United Nations, Rome, Italy, pp. 193, ISBN978-92-5-108369-7, 2015.

**3 Responses to reviewer 2**

The research aimed to present a mid-infrared soil spectral library (SSL) for central Africa (CSSL) to predict key soil properties, thus allowing (i) for future soil estimates with (ii) a minimal need for expensive and time-consuming soil laboratory analysis.
The CSSL contains over 1,800 soils from ten distinct geo-climatic regions (from the Congo Basin and wider African Great Lakes region) for a whole of six hold-out core regions.

The paper is affected by several issues, and therefore I must suggest its rejection.

We thank the reviewer for the time and effort in reading and commenting on our proposed manuscript. We are confident that
the issues the reviewer raises could be addressed in the revised manuscript. We understand that certain methods and interpretations were not clearly formulated and added missing information and rephrased unclear sentences. Additionally, as described in our response above to Reviewer 1, we reported and discussed the effect of spiking more in detail. We respectfully disagree with several repeated main concerns of the reviewer that the pedogenic heterogeneity of the soils would be a critical problem of our study. Presenting the differences between central African soils to the soils covered by the Sub-Saharan spectral
library indicates, on the contrary, the importance of our presented data analysis. Due to the new variability of the soil samples that our central African spectral library adds to the existing continental library, prediction accuracies will be significantly improved for these regions. Our findings and platform also encourage the future addition of new data. Our infrared library therefore helps to more accurately predict central African soil samples and represents a first step towards filling a critical knowledge gap of this understudied area.

In the following points, my main concerns:

General comment: used methods or obtained results do not justify several sentences. In the following points, some example are reported, but many other occurs;

Abstract: "we present a mid-infrared soil spectral library (SSL) for central Africa (CSSL) that can predict key soil properties"…but after the author state, "We present three levels of geographical extrapolation, deploying Memory-based learning (MBL) to accurately predict carbon (TC) and nitrogen (TN) contents in the selected regions.". So, you are not presenting a CSSL to predict key soil properties, but "only" some selected soil properties! The authors should be consistent throughout the text.

The reviewer is correct, we present a workflow on how to predict total carbon and total nitrogen of soil samples using our central African SSL together with an existing continental library. We also made all data (spectra, metadata and wet laboratory measurements) and accompanying code openly available on a Github repository. This will not only allow for the reproduction of our analyses but also for new analysis and predictions of soil properties in new studies. Importantly, this will facilitate new soil analyses for this highly understudied area. As we state in subsection 2.2, L103-106, additional soil properties, which are included in the repository, were analysed. These include pH, texture, total Al, Fe, Ca, Mg, Mn, Na, P, and K. We chose to highlight TC and TN as example properties to demonstrate our predictive models in a concise way. The additional data for the parameters listed above and also the results of the same analyses are available on the GitHub repository. We modified the methods and the discussion to re-iterate the availability of these auxiliary data.

Abstract and Discussion: "The Root Mean Square Error of the predictions (RMSEpred) values were between 0.38–0.86 % and 0.04–0.17 % for TC and TN, respectively, when using the AfSIS SSL only to predict the six regions. Prediction accuracy could be improved for four out of six regions when adding central African soils to the AfSIS SSL. This reduction of extrapolation resulted in RMSEpred ranges of 0.41–0.89% for TC and 0.03–0.12% for TN." Ok, but immediately after I read, "In general, MBL leveraged spectral similarity and thereby predicted the soils in each of the six regions accurately; the effect of avoiding geographical extrapolation and forcing regional samples in the local neighborhood (MBL-spiking) was small)" or, even along the Discussion section (line 309), "We showed that TC and TN in six regions of our CSSL can be accurately predicted"…so, in the same paper, the authors write two opposite things. I agree, according to your results, that the first sentence was more closes to reality than the second one, but this bring to an additional issue, i.e., see point 4;

We agree with the reviewer that these sentences provide limited context for which circumstances the inclusion of chemically associated spectral information was beneficial. As described in our responses to Reviewer 1, the effect of spiking on the prediction accuracy was substantial. We modified the abstract and body text to maintain consistency of this result throughout.

Abstract, Discussion, and Conclusions: your results don't look so "promising" (lines 17, 352) as you state, and some of your results and the following discussion are too much speculative;

We thank the reviewer for their perspective but respectfully disagree that the results do not look promising. Compared to other large-scale mid-infrared prediction studies (e.g. Dangal et al. (2019), Angelopoulou et al. (2020)) and also to other soil infrared studies, which look at geographical extrapolation strategies (e.g. Padarian et al (2019), Briedis et al. (2020), Gomez et al. (2020)), our results for TC and TN provide a method that yields satisfactory results in a simple and cost-effective manner. In fact, given the variability in soil properties covered by our data the accuracy of prediction exceeded our initial expectation and provides now a tool to further study the role of large scale patterns of soil properties in one of the least studied but fastest changing regions of the world.

Angelopoulou, T., Balafoutis, A., Zalidis, G., Bochtis, D.: From Laboratory to Proximal Sensing Spectroscopy for Soil Organic Carbon Estimation—A Review. Sustainability, 12, https://doi.org/10.3390/su12020443, 2020.

Briedis, C., Baldock, J., de Moraes Sá, J.C., dos Santos, J.B., Milori, D.M.B.P.: Strategies to improve the prediction of bulk soil and fraction organic carbon in Brazilian samples by using an Australian national mid-infrared spectral library, Geoderma, 373, https://doi.org/10.1016/j.geoderma.2020.114401, 2020.

Dangal, S., Sanderman, J., Wills, S., and Ramirez-Lopez, L.: Accurate and Precise Prediction of Soil Properties from a Large Mid-Infrared Spectral Library, Soil Systems, 3, https://doi.org/10.3390/soilsystems3010011, 2019.

Gomez, C., Chevallier, T., Moulin, P., Bouferra, I., Hmaidi, K., Arrouays, D., Jolivet, C., Barthès, B.G.: Prediction of soil organic and inorganic carbon concentrations in Tunisian samples by mid-infrared reflectance spectroscopy using a French
national library, Geoderma, 375, https://doi.org/10.1016/j.geoderma.2020.114469, 2020.

Padarian, J., Minasny, B., McBratney, A.B.: Transfer learning to localise a continental soil vis-NIR calibration model, Geoderma, 340, 279-288, https://doi.org/10.1016/j.geoderma.2019.01.009, 2019.

Results and Discussion: authors didn't explore limits in their proposed method. For instance: issues arising from the use of RMSE to compare predictions among regions with different pedoenvironmental features and, consequently, total C and total N.

We thank the reviewer for this comment and agree that it can make sense to use the RMSEpred to compare between regions but strictly together with the range of the measured attribute since data distributions are different. Nevertheless, the RMSEpred
is an appropriate error metric to compare the predictive capacity across the of the three modeling strategies, as assessed by individual regions (e.g., Table 3). As answered above in response to Reviewer 1, we modified the manuscript to better reflect limitations of obtained accuracies. We also discussed geophysical and environmental variability between the regions more in depth.

Soil sampling method and approach: soils were sampled according to a prefixed depth technique (Table 1) without considering soil variability in terms of main genetic horizons. So, this means that there is huge variability in processes and, consequently, pedogenetic features. But this problem is not considered as a possible cause of errors in obtained results. This is totally a mistake for this reviewer. Indeed, looking at Table 2, it was clear that a quite high pedovariability exists in investigated soil samples (samples comes from five different RG);
We respectfully disagree with the reviewer's opinion. We agree that using samples that were sampled per horizon would have been an advantage for using the data for pedogenetic interpretation later on. However, such data is rare at continental scales.

We would also like to have a complete chemical and pedological (soil forming factors) characterisation of the collected, analysed and modeled soils, but this is a cumbersome endeavour to explore in full detail (XRD, geomorphology, land use (history), etc.). This is simply not feasible for the size and extent of our soil collection and thus deemed beyond the purview of this study. Given that the depth increments for samples included here did in most cases not exceed 10 cm increments we believe that our predictions can still yield considerable depth explicit information. Samples were taken in a way that a large variety of mineral and organic mixtures are covered. Soil spectroscopy can naturally deal with such soil complexity. In terms of the methodology used here, since our data covers a significant variability of soil conditions, our library can be used for samples taken with fixed depth increments or sampled pedogenetically following horizon boundaries. An additional advantage of depth explicit sampling is the fact that for example TC and TN stocks can now be accounted for by various volumes of soil. One of the nice features of infrared reflectance spectroscopy is that it generates signals arising from absorption features of chemical bonds that are distinctive of functional groups and the organic or mineral compounds that contain them. Spectra offer an integrative fingerprint to comprehend major chemical complexity and selected physical properties in soils in combination with statistical modeling. One of the key assumptions (and generally the foundation of predictive capacity) is that chemical relatedness is sufficiently reflected in the spectra. In the case of memory-based learning with a nearest neighbour (distance) approach, chemical relatedness and thus the pedogenic resemblance is even enforced in the modeling process via a nearest neighbor approach. Variability in soil processes and soil dynamics are undoubtedly the latent driving forces behind the chemical composition of the measured soils. However, we specifically highlight that the predictive errors must be directly related to the representativeness in terms of chemical composition of soils and number of samples that were available in respective modeling strategies and regions (see Table 3). Furthermore, information on soil transforming factors such as parent material, and other environmental conditions, which affect the biogeochemical attributes of soils, was already included in the submitted version of the manuscript (see e.g. Table 1 and Table 2).

Whole paper: a group of references should always be avoided. It could be preferred to use a max of 2 refs. after every important statement. Otherwise, it could be quite impossible to verify if reported references was cited in a good way;

While we appreciate the reviewer's perspective, we tried to limit chains of citation where possible. We were careful in our selection of references and are confident that each reference we cited is suitable for the given statement. Since infrared spectroscopy is at the boundary of disciplines; it involves interdisciplinary methods that were developed in different fields, e.g. statistics, statistical learning, general soil science, chemistry, physics, chemometrics, pedometrics, electrical engineering (signal processing). In these situations, it is necessary to cite often a series of papers that describe complementary parts of the overall approach and method.

We also checked the SOIL guidelines and there is no limit with regards to the number of references that can be cited together for supporting our statements.

Whole paper: several acronyms appear without any explanation!.

We thoroughly checked to make sure acronyms are all defined in the revised version.

Whole paper: several typing mistakes occur. Some are reported here (vide infra), but many others occur. Additionally, the correctness of some sentences is questionable;

We thank the reviewer for the comment. The mentioned typos were corrected, and the manuscript was be carefully reviewed for spelling and grammar by a native English speaker.

Title: too generic and not fully in agreement with obtained results (vide infra). Indeed, I am not sure that you have filled a gap; at least in an accurate way;

The results clearly show that the presented soil spectral library drastically reduces the need of novel chemical measurements because the new library adds complementary information which improves the trade-off between the amount of classical re-analysis to be done in the lab and estimation accuracy. This is an important step forward in order to enable researchers from developing countries with limited funds to gain data on soil properties without the need of extensive chemical analyses (something that was not possible for tropical Africa before). For many soil parameters Infrared spectroscopy can reach similar accuracies together with traditional laboratory reference measurements. Every method has flaws and errors occur also in wet chemistry analyses (e.g. preparation). If there is considerable uncontrollable variation in the chemical measurements, spectroscopy-based approaches excel at reducing the bias in the measure of interest. The estimation accuracies obtained in the regions using the relevant spectral data and libraries were very close to typically reported accuracy limits for total carbon, for example (for references and more details see comment above).

Abstract (line 11): AfSIS!?!

Thanks for spotting this acronym standing for Africa Soil Information Service. We replaced the acronym with the full title in the revised manuscript.

Introduction (from l. 28-30): "Despite the expected severity of these impacts, our understanding of the effects in the humid tropics are limited by sparse data and uneven distribution of low-latitude research". Too vague and generic sentences. For instance, such a sentence is not true for many areas of Brazil;

We agree this sentence was perhaps too vague, however, it is true that there is a general tendency of sparse soil data availability in the humid tropics. We rephrased the sentence to say more explicitly that there is in particular a lack of soil data for the humid tropics of Africa.

Introduction (l. 30-31): "which contains the second largest tropical forest ecosystem on Earth and represents a considerable reservoir of soil C (FAO and ITTO, 2011)". Old reference. Ten years are already gone by. In case of such important statement more recent, an updated information must be reported;

We replaced the reference from FAO and ITTO (2011) by a more recent publication.

Hansen, M. C., Potapov, P. V., Moore, R., Hancher, M., Turubanova, S. A., Tyukavina, A., Thau, D., Stehman, S. V., Goetz, S. J., Loveland, T. R., Kommareddy, A., Egorov, A., Chini, L., Justice, C. O., and Townshend, J. R. G.: High-Resolution Global Maps of 21st-Century Forest Cover Change, Science, 342, 850–853, https://doi.org/10.1126/science.1244693, 2013.

Introduction (l. 33): "Thus, the projected drastic population growth in the coming decades (Vollset et al., 2020)" a quantification in terms of percentage, or something like this, is always required; otherwise, it is just a vague statement;

We agree with the reviewer that a quantification is useful and changed the sentence as following:

*"Human populations in Uganda, Rwanda and the DRC are projected to more than double in the coming 80 years (Vollset et al. 2020). Such dramatic growth will likely contribute to further agricultural conversion. "*

Introduction (l. 35-36): "In the wake of these current and future impacts, more spatially explicit soil information is urgently needed in many research fields." Again, too vague and generic sentence. Which field of research?;

We thank the reviewer for this comment. Soil data applies to multiple disciplines in environmental science, ranging from agricultural to soil, biogeochemistry and climate sciences.

Introduction (l. 44): "low cost" always depends on the point of view. What does for the authors "low cost" means? Why not introducing a specific brief paragraph for cost estimation by comparing soil analysis vs. DRIFT spectroscopy;

We thank the reviewer for this comment. With costs we mean the monetary expenses for soil laboratory analyses. In our opinion, this sentence already explains why these costs are low: fast, simple handling, less work, minimal chemical consumables. This further allows high repeatability and coverage of spatial soil heterogeneity, which we added to the sentence.

Introduction (l. 50-55): too speculative sentences. It seems more an authors' self-convincement rather than a scientifically based questions;

We are not fully sure at what the reviewer is getting at. The paragraphs elaborate on the benefits of soil spectroscopy including defined, targeted workflows. References are given. No scientific questions were raised.

Introduction (l. 52-53): sorry, I really don't know what "positive predictive transfer" means;

Thanks for this hint, we repeated the answer to the exact same question reviewer 1 posed above:

With "positive predictive transfer" we describe the information transferred from a large infrared library for a new calibration of a local set as described by Padrian et al. (2019). The calibration of a new local set using a large-scale spectral library can be complex in soil science, especially when the local set covers a different geographical domain than the library. Soil spectral libraries become particularly useful when a large amount of their relevant information can be extracted in a way that it improves prediction accuracy (positive transfer) and minimizes the number of additional costly local reference measurements for quantifying soil properties in the local set (accuracy-cost trade-off). To avoid technical jargon, we rephrased the paragraph L48-L55 and moved it to L60, where it fits better into the context:

*"One of the main aims of establishing large-scale SSLs is to minimize the need for future wet chemical analyses (e.g., Nocita et al., 2014; Stevens et al., 2013; Shi et al., 2014; Viscarra Rossel et al., 2016). However, these libraries often span vast geographical areas that include different soil types and climate zones, which comprise complex soil organic C forms and mineral compositions. Due to this heterogeneity, predictions rendered by global linear regression models are often unfeasible*

*for new local soil property assessments at a regional, field or plot-scale, especially when the new set covers another geographical domain than the library. Pandiran et al. (2019) could considerably improve prediction accuracies for a new local set when using a compositionally related subset from a large-scale SSL together with a small number of local reference analyses. The cost-accuracy trade-off can be met when the accuracy of the library-based prediction is similar to the one made when applying a local but more costly calibration strategy. Several data-driven methods have proven to be successful to*

*overcome this issue, for example RS-LOCAL (Lobsey et al., 2017) and memory-based learning (a.k.a local learning e.g. Ramirez-Lopez et al., 2013; Shenk et al., 1997; Naes 1990). In addition, other promising approaches have also been proposed, although they require more research (e.g. deep learning (Ng et al. 2019), fuzzy rule-based systems (Tsakiridis et al. 2019))."*

Method (l. 91): WRB, 2006? Really? Are you aware of the 2015 updated version?

We updated the reference to the newer version, thank you!

Method (general comment): What about the way you selected "latent variables" for the global calibration you did for optimizing spectral pretreatment?;

We agree that we missed to add this important information and therefore changed it.

Find our suggested changes under Review 1, L145-161

Method: "Note, even if the proportion of samples with inorganic carbon was very low (5%), the term TC will be used in the study." As usual! Why do you need to specify such an obvious aspect?;

Highly weathered tropical soils are often acidic (pH < 6) and don't contain any inorganic carbon and therefore assumptions might be made that total carbon would correspond to organic carbon.

Method: I think that the way you pretreated your soil samples should be specified;

We added the required information (see reviewer 1, L99)

Method: "A gold standard was used as a background material for all measured soils" which kind of "standard"? It was a reference soil certified material? Why not including such important information?;

This was changed accordingly (see reviewer 1, L113, L125)

Method (Table 2): For this reviewer, it was not so clear if you used all the reported nr. of soil samples. It would help if you
were more clear from this point of view;

Some cluster areas were excluded because they did not have enough samples to provide reliable results (< 80 samples per region). We agree that this was not clearly presented. For a new version of the manuscript we removed these regions from this table. A new table with all the regions is now presented in a supplementary table in the appendix.

Method: "Reflectance was transformed into absorbance (1/reflectance) before further processing and subsequent modeling." No reference!;

The transformation from reflectance into absorbance is not arbitrary. Instead it is based on the Lambert-Beer's law (please see https://en.wikipedia.org/wiki/Beer%E2%80%93Lambert_law) which dictates that the concentration of the components in a matrix influence the way in which that matrix absorbs radiation. Although this law does not 100 % apply for opaque materials,
it serves as the fundamental theoretical basis for quantitative analysis in vibrational infrared spectroscopy and it is the underlying reason why scientists use the calculated absorbance as the starting point for the numerical analysis of their spectra. This is evidenced by countless studies (e.g. Baes and Bloom, 1990; Baharom et al., 2015; Barthès, et al., 2020; Gogé, et al., 2014; Minasny et al., 2013; Peng et. al, 2013). Therefore, since conversion from reflectance into absorbance is considered as elemental in vibrational spectroscopy, we do not see the need to provide detailed justification and references to support this
procedure. However, if the reviewer has a particular reference in mind, we would be happy to consider it for citation in our manuscript.

Baes, A. U., & Bloom, P. R.:. Fulvic acid ultraviolet-visible spectra: Influence of solvent and pH, Soil Science Society of America Journal, 54, 1248-1254, https://doi.org/10.2136/sssaj1990.03615995005400050008x, 1990.

Baharom, S. N. A., Shibusawa, S., Kodaira, M., & Kanda, R.: Multiple-depth mapping of soil properties using a visible and near infrared real-time soil sensor for a paddy field, Engineering in Agriculture, Environment and Food, 8, 13-17, https://doi.org/10.1016/j.eaef.2015.01.002., 2015.

Barthès, B. G., Kouakoua, E., Coll, P., Clairotte, M., Moulin, P., Saby, N. P., ... & Chevallier, T.: Improvement in spectral library-based quantification of soil properties using representative spiking and local calibration–The case of soil inorganic carbon prediction by mid-infrared spectroscopy, Geoderma, 369, https://doi.org/10.1016/j.geoderma.2020.114272, 2020.

Gogé, F., Gomez, C., Jolivet, C., & Joffre, R.: Which strategy is best to predict soil properties of a local site from a national
Vis–NIR database?, Geoderma, 213, 1-9, https://doi.org/10.1016/j.geoderma.2013.07.016, 2014.

Minasny, B., McBratney, A. B., Stockmann, U., & Hong, S. Y.: Cubist, a Regression Rule Approach for use in Calibration of NIR Spectra, Picking Up Good Vib, 630, 2013.

Peng, Y., Knadel, M., Gislum, R., Deng, F., Norgaard, T., de Jonge, L. W., ... & Greve, M. H.: Predicting soil organic carbon at field scale using a national soil spectral library, Journal of Near Infrared Spectroscopy, 21, 213-222, 2013.

Method: "Four replicates per sample were measured and an average of 32-co-added scans were used for each sample" why? Four replicates are enough for you? If yes, you need to explain the reasons from a statistical representative viewpoint;
This is information given from the AfSIS spectral library, which was previously measured using the standard operation procedure of the Soil-Plant Spectral Diagnostics Laboratory of the World Agroforestry Center. We found it important and therefore added it to the manuscript. The aggregation of 32-co-added internal measurements into one final spectrum per measured replicate in different wells is a strategy proposed by the OPUS BRUKER software (Bruker Optics GmbH, Germany), which is common on different IR spectrometers. Previous internal tests in our lab confirmed that there was no added benefit
doing more than four measurements on replicates in different wells, evaluated on the modeled outcome, which is the proper way of testing a measurement protocol. For example, Peng et al. (2014) report that no further prediction improvements were found by increasing replicates beyond 3 replicates, and some even show deleterious effects at excessive number of replicates likely due to higher chances causing excessive scattering. Our samples were finely powdered and have a relatively low spectral variability, and the scattering effects were alleviated by thoroughly testing single preprocessing methods and combinations
thereof.

Peng, Y., Knadel, M., Gislum, R., Schelde, K., Thomsen, A., Greve, M.H.: Quantification of SOC and Clay Content Using Visible Near-Infrared Reflectance–Mid-Infrared Reflectance Spectroscopy With Jack-Knifing Partial Least Squares Regression, Soil Science, 179, 325-332, https://doi.org/10.1097/SS.0000000000000074, 2014.

Results (general comment): very aseptic. It looks like a technical report totally detached from the context;
We appreciate this perspective; however, we were trying to adhere to the classical stylistic guidelines of SOIL in which results are presented in a "pure" form divorced from discussion and interpretation. We furthermore disagree with the opinion of the reviewer that the results were detached from the context. We clearly document that the central African MIR SSL adds complementary soil information with regard to what is already available in the library of the Africa Soil Information Service. The way we developed the estimation scenarios reflects one of the key practical issues that motivates doing spectral research, namely the fact that we use an existing library and predict understudied regions with it and therefore minimizing additional costs for new soil wet chemistry analyses. These analyses were done with statistically sound methods. The results section follows these strategies and presents our finding in a clear structure. We provide insights into patterns we found, what worked and what not, and above all, we round up our findings with a recipe.

Results (paragraph 3.1 and Fig. 3): I discover for the first time that the authors applied a multivariate approach too. In particular, they used a PCA. Unfortunately, they didn't explain to us anything about how it was implemented. This is really unusual for this reviewer. Indeed, when a multivariate tool is used, data-pretreatment represent a pivotal matter, but the authors didn't explain anything about this. Additionally, several authors, statisticians included, clearly demonstrated that PFA was better for variability interpretation in a soil dataset with soil data;

We explain the use of multivariate methods before section 3.1 The first reference to a multivariate approach (within our manuscript) is given in section 2.4 (Spectral resampling and pre-processing) of the materials and methods. Sections 2.5 (Modeling and prediction data) and 2.6 (Predictive modeling) also explain the use of multivariate methods.

Concerning the use of principal component analysis, unfortunately the reviewer does not provide any information, clue or references to scientific literature supporting the claims about "PFA" being "better" for "variability interpretation" than PCA. We assume that with "PFA" the reviewer refers to Principal Factor Analysis (as she/he does not provide the name of the method in full). Unfortunately, we did not find scientific references reporting the convenience of using PFA over PCA in the soil spectroscopy literature. Although we cannot claim what method is best (PFA or PCA) for infrared spectroscopy data (and it is not at all the purpose of our paper), we do know that PCA is a well suited method for the purpose of data visualization (which our only aim for using it). Whether PFA would add some benefit for our data visualization is then debatable.

Please also note that we do not use PCA as data pretreatment, therefore we do not explain PCA as such. Finally we used PCA, as it is the standard method for latent variable extraction and exploration in chemometrics (please see Cordella et al., 2012) and its use can be considered as standard in soil spectroscopy for exploratory analysis and visualization (e.g. Stenberg et al., 2010; Viscarra Rossel and Chen, 2011; Nocita et l., 2013; Sanderman et al., 2020).

We will be very grateful to the reviewer if he/she could share with us scientific literature about PFA in spectroscopy that we could use to consider the use of this method.

Finally, we agree with the reviewer that we could provide more details to the reader on "the implementation" of PCA and added this information accordingly.

Cordella, C. B.: PCA: the basic building block of chemometrics, Analytical chemistry, 47, http://dx.doi.org/10.5772/51429, 2012.

Nocita, M., Stevens, A., Noon, C., & van Wesemael, B.: Prediction of soil organic carbon for different levels of soil moisture using Vis-NIR spectroscopy, Geoderma, 199, 37-42, https://doi.org/10.1016/j.geoderma.2012.07.020, 2013.

Sanderman, J., Savage, K., & Dangal, S. R.: Mid-infrared spectroscopy for prediction of soil health indicators in the United States, Soil Science Society of America Journal, 84, 251-261, https://doi.org/10.1002/saj2.20009, 2020.

Stenberg, B., Rossel, R. A. V., Mouazen, A. M., & Wetterlind, J.: Visible and near infrared spectroscopy in soil science, 1005 Advances in agronomy, 107, 163-215. https://doi.org/10.1016/S0065-2113(10)07005-7, 2010.

Viscarra Rossel, R., & Chen, C.: Digitally mapping the information content of visible–near infrared spectra of surficial Australian soils, Remote Sensing of Environment, 115, 1443-1455, https://doi.org/10.1016/j.rse.2011.02.004, 2011.

Line 268: soils rather than "sols";
We thank the reviewer for spotting this typo.

Results (lines 267-268): "This was expected as the principal component analysis indicates that the sols of these regions might not be properly represented by the AfSIS library." Where? I don't see such an information from PCA;
Figure 3 shows the coverage of different PC spaces of the certain regions compared to the AfSIS SSL, which is coloured in black. The first three components explain more than 70 % of the variance in the spectra and therefore showing these three components is adequate to analyse differences between regions. Moreover, the distances in a score space provide a useful tool to analyze similarities/dissimilarities (see review 1 and comments/answers above). We agree the graph can be presented in a simpler and clearer way and changed to PC1-PC2, PC1-PC3, PC2-PC3 plots, as suggested by reviewer 1 (see above).
Moreover, we added more information on how we performed the principal component analysis in the methods section (see above).

Results (lines 276-279): I do not fully agree with the suggested reasons for the total C and N predictions underestimation trend in the six investigated regions. Indeed, several outliers occur in your dataset. This was typically due to an underestimation in 1025 investigated pedovariability (vide supra);

We thank the reviewer for this comment, but we respectfully disagree. Of course, there is a high pedogenic variability between the soils, however, using the similarity-based approach of memory-based learning we overcome this issue. Please find a detailed answer to a similar comment above. In these four lines 266-279 which the reviewer points out, we do not discuss outliers: there was a general trend of underestimation of the predictions (Haut-Katanga, South Kivu, Tshopo, Tshuapa for TC) and (Haut-Katanga, South Kivu, Tshopo, Tshuapa and Iburengerazuba for TN) for all predicted spectra (downwards shift from the 1:1 line in Figure 4). This overestimation was less pronounced in strategy 2 and strategy 3. Outliers, i.e. soil samples with large distances to the continental AfSIS SSL and therefore different in their chemico-physical properties, were removed from these analyses. These samples cannot be accurately predicted by the library and need therefore to be traditionally analysed. We emphasized this more in depth in the revised manuscript.

"Results" and "Discussion" (general comment): both these parts are full of "could", "may", "might", etc. I understand that caution is always required in a scientific text, but some more certainties should be given. So, I wonder: are the authors sure enough of the applied method and the validity of the obtained results or not? As a reviewer, the text has several methodological drawbacks, which bring me to hypothesize that all these doubts could be the demonstration of a low statistical robustness of obtained results;

The reviewer is correct, using these words too often leaves the impression of uncertainty. That was not our intention and we changed this accordingly. However, we are confident of the correctness and robustness of our methods and results.

Discussion (line 309): "We showed that TC and TN in six regions of our CSSL can be accurately predicted". Honestly, I am not agreed. In previous pages and Tables, total C and N prediction can be rarely defined as "accurate";

We kindly disagree with the reviewer. For this large-scale continental study, these results are accurate with reasonably low prediction errors, especially when comparing them to studies covering similar large geographical areas (see comment and references above).

Discussion (line 309): "The advantage of using MBL is that it finds spectrally similar observations for every new observation to fit specific models". This is an obvious observation that can be written for every prediction "model";

The reviewer might have misunderstood the methods of our modeling approach. General predictive models are trained with all available calibration data and the new observations are predicted by this "global" model, regardless of the similarity to the observations in the calibration set. As we described in the introduction, line 65, with memory-based learning, a predictive model is trained specifically for the prediction set using a subset of samples in a library based on their similarity/dissimilarity. Therefore, we don't see the problem with this sentence.

Discussion (line 312): ")"…?;

Thank you very much for spotting this typo.

Discussion (general comment): extremely redundant with the "Results" section. A combination of the "Results and Discussion" section it would have improved the paper in terms of overall quality, clarity, and readability;

We thank the reviewer for this suggestion; however, we followed the guidelines of SOIL. Please see above.

Discussion (general comment): readability is made really low due to the presence of too many acronyms. I understand that several acronyms characterize the whole paper, but some strategies would have improved readability (for instance, avoiding its use while preferring a "recall" of their original meaning);

We agree with the reviewer that in general, too many acronyms make it hard to follow a text. Nevertheless, we do not think
we used too many acronyms in this manuscript. We abbreviated the two spectral libraries (CSSL and AfSIS SSL), the modeling method (MBL, PLS, WA-PLS), statistics (RPIQpred, RMSEpred, PCA), the soil properties (TC, TN), a long country name (DRC), spectroscopic specific terms (IR, FT-IR, MIR), which are mostly very common in soil infrared spectroscopy publications. We improved the readability of the discussions by repeating some describing words (e.g. continental/large-scale AfSIS SSL). If the reviewer insists, we will add a table with explanations of the acronyms to the manuscript.

Discussion (line 319-323): another obvious observation that strongly affect your paper in terms of novelty;

We thank the reviewer for this comment, but again, we strongly disagree. The contrary is the case: exactly with these lines as the reviewer points out, we highlight the novelty and importance of our research and results. We establish a soil spectral library with soil samples from the humid central African tropics including forest soils with high organic carbon contents. This area
has not been covered by the previously established continental AfSIS infrared library yet (Figure A1) and is still highly understudied. With our proposed infrared library, we bring a new soil variability and improve predictions for soil TC and TN (as well as many other soil parameters) for central African regions (for more details please see comments above on a similar question). However, we added Figure A1 to the main text and rephrase these sentences to make this clearer.

Discussion (line 324-326): "We conclude that the particularly high soil diversity in these two regions in terms of soil biogeochemical properties introduces additional complexity in the soil spectral prediction workflow" this is the point! Even if, in my opinion, it would be better to use "soil bio-physical-chemical features" rather than "soil biogeochemical properties". However, this clearly confirm all my previous doubts, and I am astonished that the authors recognized such a big issue only at the end of their paper without additional insights about this;

We agree with the reviewer, that we should also include physical properties to the sentences and changed it as following:

*"We conclude that the particularly high soil diversity in these two regions in terms of soil biogeochemical and soil physical properties introduces additional complexity in the soil spectral prediction workflow."*

However, we kindly disagree with the reviewer about seeing an issue behind this sentence. As already answered in the comment above, this argument points out the importance of our study and our data we contribute to the scientific community. The complexity and differing chemical, biological and physical properties in soils from the Congo Basin will improve future soil analyses for these particular regions and bring new variability to already existing soil spectral libraries (see comments/answers above on a similar question). The positive impact of spiking (reducing RMSEpred, increasing RPIQpred values) underlines this argument. These regions have not been covered by the existing continental library, moreover they can also not be represented by the soils of the other central African regions. Adding the region-specific soil properties by spiking (Table 4, Figure 5) has shown to be effective and will also be effective and be improved by the future addition of new data. We acknowledge that this has not been discussed enough in the discussion and added this accordingly.

Discussion (line 324-326): "Regions that occupied the same score space of the first two principal components as the corresponding other regions and the AfSIS SSL (Figure 3) showed only a minimal effect from spiking (Figure 1)" where I can see such an outcome? It is not contained in Fig. 3 and 1 for sure;

We assume the reviewer addresses the lines 340-341 with this comment (instead of the indicated lines). Figure 3 presents the first three components of a principal component analysis of the pre-processed MIR spectra, which cover together more than 70 % of the variance. Therefore, we argue that the 3D visualization of these score spaces is a first indication of differences, in case of large (e.g. mahalanobis) distances. South Kivu (orange) clearly covers a large area differing from the AfSIS SSL and the other central African regions (and to some extent also Iburengerazuba and Tshuapa). In our opinion, these larger distances can be used to discuss the performances of the strategies. Spiking had a positive effect on all regions (Figure 5, will be corrected), which can be explained by the addition of closer and more similar samples to the prediction models. We agree that the 3D plot is not appropriate and changed it to PC1-PC2, PC1-PC3 and PC2-PC3 plots, as suggested by the reviewer 1 (see above).

Discussion (l. 348-250): "Even though spiking is described as particularly effective in improving performance of small sized models (Guerrero et al., 2010), spiking, in our study, did not have as strong of an effect as reported by earlier studies (e.g., Guerrero et al., 2014; Seidel et al., 2019; Barthès et al., 2020; Wetterlind and Stenberg, 2010)…and the reason is!?!;

We fully agree with the reviewer that the effect of spiking has to be discussed more in depth. The effect was actually pronounced for all regions. Spiking reduced the RMSEpred for all regions for TC and TN and increased RPIQpred values. The positive effect of spiking is due to the addition of local samples to the models and therefore adding information of the target region. We added more explanations to the results and discussion sections.

Discussion (l. 353-354): "The addition of geographically proximal regions to the large-scale library, which are included in our CSSL, improved prediction accuracy significantly". Sorry but once again, I disagree with the authors. From your reported results, it seems that accuracy improved but not in a so highly significant degree;

We thank the reviewer for this comment. We understand that this sentence is not clear, and we rephrased it as following:

*"Six central African regions were predicted for soil TC and TN with sufficient accuracy using the large-scale AfSIS soil spectral library only. The general positive effect of adding geographically closer samples to the AfSIS SSL (strategy 2)*
*underlines the usability of spectral libraries for new regions. The generally positive effect of strategy 3, spiking of all regional predictions for TC and TN with samples from the target area, encourages the future amendment of currently existing libraries to improve prediction accuracy. "*

We respectfully disagree with the reviewer, that prediction accuracy did not improve between the three strategies. For a study
in this scale the prediction errors were on one hand more than sufficient for most scientific and applied uses and on the other hand, they were considerably improved at least for strategy 3 compared to strategy 1. The accuracy gain is of course relative and there are different requirements on accuracy depending on the interests and the possibilities to invest in more expensive laboratory wet chemistry analyses. We described and discussed these trade-offs in more detail to emphasize this change.

References: Total nr. of references: 77…too much for an original article; Total nr. of references before 2011 > 20; Self-citations > 10

We thank the reviewer for checking our references attentively. We use citations to confirm our statements where required. We carefully went through all of them and re-evaluate them to see if we could reduce it to a smaller number. Indeed, there is a problematic tendency in modern scientific writing to only cite the most recent references that often make claims that were
established much earlier by original studies. We therefore kindly disagree with the reviewer that the older references would be problematic. Moreover, we only added self-citations that were absolutely necessary. The presented library stems from both soil archives and data collected within different projects, universities, and institutes. Most of the sample sets have already been published (Table 1), therefore we find it crucial to cite the original studies. Without these collaborative research and data collection efforts, we could not have created this library.

---

## Editor Decision (ED1)

**The Central African Soil Spectral Library: A new soil infrared repository and a geographical prediction analysis**

Laura Summerauer1, Philipp Baumann1, Leonardo Ramirez-Lopez2, Matti Barthel1, Marijn Bauters3,4, Benjamin Bukombe5, Mario Reichenbach5, Pascal Boeckx3, Elizabeth Kearsley4, Kristof Van Oost6, Bernard Vanlauwe7, Dieudonné Chiragaga7, Aimé Bisimwa Heri-Kazi6, Pieter Moonen8, Andrew Sila9, Keith Shepherd9, Basile Bazirake Mujinya10, Eric Van Ranst11, Geert Baert3, Sebastian Doetterl1,5, and Johan Six1

1Department of Environmental Systems Science, ETH Zurich, Switzerland
2Data Science Department, BUCHI Labortechnik AG, Flawil, Switzerland

[revised manuscript text omitted]

---

## Author Response (AR2)

**Response with revised manuscript**

"The Central African Soil Spectral Library: A new soil infrared repository and a geographical prediction analysis" by Summerauer et al. (SOIL-2020-99)

**Dear Editor,**

5    We thank for the helpful comments on our revised manuscript "The Central African Soil Spectral Library: A new soil infrared repository and a geographical prediction analysis" by Summerauer et al. (SOIL-2020-99). All grammatical errors have been fixed and suggestions have now been incorporated into the final manuscript. A track-changed version of the manuscript is also provided separately.

10    We hope we addressed all comments to your satisfaction.

On behalf of all co-authors,

Laura Summerauer